# PROVABLE ADVERSARIAL DETECTION: PRIME QUANTIZATION MEETS GROMOV–WASSERSTEIN

## ABSTRACT

Adversarial vulnerability persists across modern vision architectures from CNNs to vision language models (VLMs), yet existing detection methods rely on heuristics without theoretical guarantees. We address the fundamental question of when adversarial perturbations can be provably detected from a geometric perspective. Our key insight is that adversarial perturbations cannot simultaneously preserve geometric structure across spaces with fundamentally different properties. Accordingly, we construct two such complementary metric spaces. First, we use a standard CNN embedding space $Z$, where adversarial samples exhibit significant displacement patterns. Second, we build a novel prime-quantized space $P$, that absorbs small perturbations through number-theoretic discretization, resulting in minimal displacement, while preserving discriminability. We then leverage the geometric discrepancies across spaces $Z$ and $P$ to detect adversarial samples. To the best of our knowledge, we establish the first rigorous separation theory for adversarial detection, proving that adversarial samples create unavoidable geometric inconsistencies across both spaces. Our framework provides theoretical guarantees including pixel-level absorption bounds, neighborhood diameter concentration, Gromov-Wasserstein (GW) separation theorems, and practical risk control. Extensive experiments validate our theoretical predictions and achieve consistently strong detection performance across a wide range of attack types and model families.

## 1 INTRODUCTION

Vision systems have rapidly progressed from CNNs He et al. (2016) to Vision–Language Models (VLMs) Radford et al. (2021) and multimodal architectures OpenAI (2023), yet adversarial vulnerability persists across all these paradigms. As these increasingly capable models are deployed at scale, the consequences of undetected adversarial attacks also scale, making *detection* a core safety requirement.

Defenses fall into three main families. *Adversarial training* (Madry et al., 2018; Zhang et al., 2019; Gowal et al., 2021) augments models with adversarial examples. While effective in restricted scenarios, it requires expensive retraining and often fails to generalize across diverse attacks, including gradient-based (Goodfellow et al., 2015; Madry et al., 2018; Carlini & Wagner, 2017), physical (Brown et al., 2017), and natural corruptions (Hendrycks & Dietterich, 2019b; Engstrom et al., 2019). *Detection methods* (Metzen et al., 2017; Feinman et al., 2017; Ma et al., 2018; Lee et al., 2018; Ma et al., 2019; Meng & Chen, 2017; Mahmood et al., 2021) rely on auxiliary classifiers or statistical tests, but remain heuristic and easily broken by adaptive adversaries (Athalye et al., 2018). *Certifiable robustness* (Raghunathan et al., 2018; Cohen et al., 2019; Salman et al., 2019) provides provable invariance regions, but targets robust classification rather than detection, and is computationally intensive. Trade-off frameworks such as TRADES (Zhang et al., 2019) deepened our theoretical understanding, but left unanswered the key question: "*Can detection itself be endowed with guarantees, and what properties make adversarial examples inherently detectable?*"

**Our insight.** Clean and adversarial samples leave distinct geometric traces across two complementary spaces, namely, the CNN embedding space $Z$ and a prime-quantized space $P$. In $Z$, clean samples form tight neighborhoods, while adversarial ones exhibit characteristic displacements that disrupt local structure. In $P$, each pixel is discretized by rounding to nearby primes under a secret

bit mask, so small perturbations are either absorbed within prime gaps or forced into discrete jumps. This mechanism preserves overall discriminability yet creates systematic cross-space inconsistencies, making adversarial inputs detectable through geometric analysis.

To the best of our knowledge, we establish the first rigorous *separation theory for adversarial detection*, spanning four levels of guarantees: (i) pixel-level absorption bounds proving when perturbations vanish in $P$ or cross prime-gap boundaries, (ii) $K$-NN diameter envelopes showing clean and adversarial neighborhoods diverge differently in $Z$ and $P$, (iii) cross-space separation theorems based on Gromov–Wasserstein (GW) distances that yield a *non-vanishing gap* scaling with dimension and perturbation strength, and (iv) risk control guarantees establishing that simple thresholding achieves bounded misclassification rates. Together, these results provide a principled foundation for adversarial detection, addressing the open question of when and why detection must succeed.

**Contributions.** (i) We introduce *prime quantization*, a cryptographically inspired discretization that generalizes across CNN, VLM, and multimodal architectures, and could extend to other one-way transforms. (ii) We present a unified theoretical framework proving that adversarial perturbations necessarily induce cross-space inconsistencies, with guarantees from pixel absorption up through GW-based separation. (iii) We empirically validate our method on a broad suite of attacks, VLM zero-shot settings, and adaptive adversaries, demonstrating consistent and strong performance compared to state-of-the-art defenses.

## 2 RELATED WORK

**Adversarial attacks** are categorized by attacker knowledge into: (i) *white-box* (FGSM (Goodfellow et al., 2015), PGD (Madry et al., 2018), C&W (Carlini & Wagner, 2017)), (ii) *black-box* (ZOO (Chen et al., 2017), Square (Andriushchenko et al., 2020)), and (iii) *adaptive attacks* that exploit defense mechanisms (Athalye et al., 2018), often defeating methods that appear robust under non-adaptive evaluation.

**Detection methods** include: (i) *autoencoder-based reconstruction* (MagNet (Meng & Chen, 2017), PixelDefend (Song et al., 2018)), (ii) *distributional analysis* (Mahalanobis (Lee et al., 2018)), (iii) *prediction differences* (Feature Squeezing (Xu et al., 2018)), and (iv) *learned classifiers* (MetaAdvDet (Ma et al., 2019)). These approaches remain heuristic and are routinely bypassed by adaptive adversaries, with no guarantees on when detection must succeed.

**Robust training and certification** methods such as adversarial training (Madry et al., 2018), TRADES (Zhang et al., 2019), and certified defenses based on randomized smoothing (Cohen et al., 2019) or patch-based strategies (Xiang et al., 2022) aim at robust classification rather than detection, often requiring retraining and incurring accuracy trade-offs.

*In contrast, our work provides the first theoretical guarantees that adversarial perturbations create unavoidable cross-space inconsistencies, yielding a principled basis for detection with quantifiable confidence. Unlike most prior defenses, we further evaluate on vision–language models in zero-shot settings, highlighting robustness beyond CNN benchmarks.*

## 3 PRELIMINARIES

We introduce notation and the prime-quantized space used by our detector.

Let $X \in [0,1]^{N \times d}$ be $N$ images, each $x_i \in [0,1]^d$ a flattened vector of $d$ normalized pixels with label $y_i \in Y$ ($|Y| = C$). A classifier $f_\theta = g_\theta \circ h_\theta$ (where $h_\theta$ is the *feature extractor* and $g_\theta$ the *classification head*) has embedding $Z_i = h_\theta(x_i) \in \mathbb{R}^m$, with $Z = [Z_1, \ldots, Z_N]^T$. With a slight abuse of notation, we also use $Z$ to denote the ambient embedding space $\mathbb{R}^m$ equipped with the Euclidean metric $d_Z(z_i, z_j) = \|z_i - z_j\|_2$. Thus, each $Z_i$ is both a row of the embedding matrix and a point in metric space $(Z, d_Z)$.

**Definition 1** (Adversarial perturbation). *Given $x_i$, a perturbation $\eta \in \mathbb{R}^d$ yields $\tilde{x}_i = x_i + \eta$. It is $\epsilon$-bounded if $\|\eta\|_\infty \leq \epsilon$.*

To defend against such perturbations, we transform images into a discrete prime space via three steps: (i) scale to integers, (ii) round to primes, and (iii) rescale.

**Definition 2** (Scaling). *For $k \in \mathbb{N}$, let $\mathbb{P}_k = \{p \leq 10^k : p \text{ prime}\}$. Define $S_k(x_{i,j}) = \lfloor x_{i,j} 10^k \rfloor$ and $S_k^{-1}(n) = n \cdot 10^{-k}$.*

**Definition 3** (Prime rounding). *Given secret bit $b_j \in \{0, 1\}$, map integer $n$ to the nearest prime in $\mathbb{P}_k$. If $p_\ell < n < p_{\ell+1}$, where $p_\ell, p_{+1} \in \mathbb{P}_k$, then $R_k^{(0)}(n) = p_\ell$, $R_k^{(1)}(n) = p_{\ell+1}$.*

**Definition 4** (Prime quantization). *The pixel transform is $T_k^{(b_j)}(x_{i,j}) = S_k^{-1}(R_k^{(b_j)}(S_k(x_{i,j})))$. Extending component-wise yields $T_k^{(b)} : [0, 1]^d \to [0, 1]^d$.*

**Example 1.** *For $k = 2$, $x_{i,j} = 0.38$ gives $S_2 = 38$, between $37$ and $41$. Then $T_2^{(0)} = 0.37$, $T_2^{(1)} = 0.41$. A perturbation $0.385$ still maps to $38$, hence quantization is unchanged.*

**Space transformation.** $T_k^{(b)}$ maps images into discrete $P$, where prime gaps and secret $(b, k)$ yield irregular, attacker-unpredictable rounding. Unlike uniform quantization, prime rounding introduces structured but unpredictable discretization.

**Problem statement.** Given $x \in [0, 1]^d$, construct a detector $D(x) \in \{\text{clean}, \text{adv}\}$ by comparing $h_\theta(x) \in Z$ and $T_k^{(b)}(x) \in P$, ensuring w.h.p. that clean inputs agree across spaces while adversarial ones create detectable discrepancies.

# 4 MULTISCALE GROMOV-WASSERSTEIN (GW) ADVERSARIAL DETECTOR

Our method compares neighborhood behaviors of samples across the embedding space $Z$ and the prime-quantized space $P$, exploiting their complementary geometries.

**Clean neighborhoods.** In $Z$, clean samples cluster by class, so a sample's local neighborhood is dominated by its true label and its global neighborhood aligns with class centroids. In contrast, prime quantization scatters samples uniformly in $P$, destroying spatial coherence, resulting in neighborhoods that show nearly uniform label distributions without clustering.

**Adversarial neighborhoods.** In $Z$, adversarial samples *jump* from their true cluster toward a wrong class, shifting both local and global label distributions. In $P$, prime quantization often reduces the impact of small perturbations, depending on the quantization gaps and perturbation strength. Therefore, adversarial neighborhoods tend to resemble their clean counterparts more closely. This creates systematic cross-space discrepancies between $Z$ and $P$.

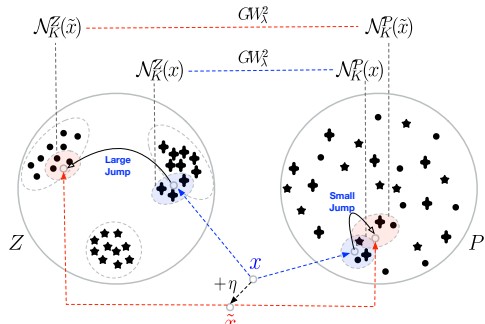

Figure 1: Geometry in $Z$ (clusters) vs. $P$ (dispersed).

We next formalize these ideas via *neighborhood maps* and quantify their mismatches using *Gromov-Wasserstein (GW) distances*.

**Definition 5** (Neighborhood map and induced distributions). *Let $(\mathsf{M}, d_\mathsf{M})$ be a metric space. For a query $q \in \mathsf{M}$ and integer $K \geq 1$, a neighborhood map $\mathcal{N}_K^\mathsf{M} : \mathsf{M} \to 2^\mathsf{M}$ returns a set of $K$ reference points, defining local neighborhoods when $K$ corresponds to nearest neighbors or global neighborhoods when $K$ corresponds to class centroids. The corresponding spatial distribution is $\mu_K^\mathsf{M}(q) = \frac{1}{K} \sum_{z \in \mathcal{N}_K^\mathsf{M}(q)} \delta_z \in \mathcal{P}(\mathsf{M})$ (i.e., the space of probability measures over $\mathsf{M}$). If $c : \mathsf{M} \to \{1, \ldots, C\}$ is a class-label map, the corresponding semantic distribution is defined by the pushforward $\psi_K^\mathsf{M}(q) = c_\# \mu_K^\mathsf{M}(q) = \frac{1}{K} \sum_{z \in \mathcal{N}_K^\mathsf{M}(q)} \delta_{c(z)} \in \mathcal{P}(\{1, \ldots, C\})$.*

The GW distance compares probability distributions that are supported on possibly distinct metric spaces via alignment.

**Definition 6** (Gromov-Wasserstein Distance). *Consider two metric measure (mm) spaces $(X, d_X, \mu_X)$ and $(Y, d_Y, \nu_X)$ along with a loss function $L^2(x, x', y, y') := |d_X(x, x') - d_Y(y, y')|^2$,*

*the squared GW distance between them is*

$$GW^2(\mu_X, \nu_Y) := \inf_{\gamma \in \Pi(\mu_X, \nu_Y)} \int_{X \times Y} \int_{X \times Y} L^2(x, x', y, y') \gamma(dx \times dy) \gamma(dx' \times dy')$$

*, where $\Pi(\mu_X, \nu_Y)$ denotes the set of* couplings *between measures $\mu_X$ and $\mu_Y$. Additionally, $\gamma(dx \times dy)\gamma(dx' \times dy')$ represent integration w.r.t. the product coupling $\gamma \otimes \gamma$.*

As exact computation of $GW^2$ uses a *quadratic assignment problem* (QAP), known to be NP-hard Abdel Nasser H. Zaied (2014), various approximate reformulations that are computationally tractable have been proposed. We focus on the *entropic GW* distance proposed by Peyré et al. (2016)

$$GW_\lambda^2(\mu_X, \nu_Y) := \inf_{\gamma \in \Pi(\mu_X, \nu_Y)} \int \int L^2(x, x', y, y') \gamma(dx \times dy) \gamma(dx' \times dy') + \lambda KL(\gamma \| \mu_X \otimes \nu_Y)$$

, where $KL(\cdot, \cdot)$ is the Kullback-Liebler divergence between coupling $\gamma$ and the product measure $\mu_X \otimes \nu_Y$, and $\lambda > 0$ is a regularization parameter.

**Choice of scales (lo, gl).** The parameter `lo` denotes the local neighborhood size, i.e., the $k$ in the local $k$-NN graph used to enforce within-space consistency in both $Z$ and $P$ spaces. The parameter `gl` denotes the number of $k$-means centroids used to construct the global support for the cross-space GW coupling. This two-scale local/global structure follows the standard decomposition in GW geometry.

**Algorithm.** Our detector takes an image $x$, extracts its CNN embedding $z = h_\theta(x) \in Z$ and prime-quantized version $p = T_k^{(b)}(x) \in P$, and compares neighborhoods at two scales ($s \in \{\text{lo, gl}\}$). For each scale, we compute (i) spatial distributions $\mu_s^Z, \mu_s^P$, (ii) semantic distributions $\psi_s^Z, \psi_s^P$, and derive $g_1 = \text{GW}_\lambda^2(\mu_s^Z, \mu_s^P)$, $g_2 = \text{GW}_\lambda^2(\psi_s^Z, \psi_s^P)$, and entropy $h = \text{ENTROPY}(\psi_s^Z, \psi_s^P)$. The resulting six-dimensional feature vector $\mathbf{f}(x) = [g_{1,\text{lo}}, g_{2,\text{lo}}, h_{\text{lo}}, g_{1,\text{gl}}, g_{2,\text{gl}}, h_{\text{gl}}]$ encodes cross-space discrepancies, which are classified by an SVM. Full pseudocode is provided in Algorithm 1 in Appendix K.1.

## 5 GEOMETRIC FOUNDATIONS AND THEORETICAL GUARANTEES

We begin by analyzing the stability of the prime quantization map $T_k^{(b)}$, which is central to our cross-space detector. The key question is, *"when does a perturbation vanish into quantization noise, and when does it inevitably alter the output?"* Our results formalize two complementary phenomena: *pixel-level local stability* and *image-level injectivity*. Prime quantization related proofs are deferred to Appendix C. All our formal results are stated for *local neighborhoods*. While the framework naturally extends to global neighborhoods, we leave the full theoretical treatment of that case to future work.

### 5.1 PIXEL-LEVEL ABSORPTION

The *absorption radius* (Definition 7) captures the largest perturbation at a pixel that leaves its quantized value unchanged.

**Definition 7** (Absorption radius). *For $x_j \in [0, 1]$ with $S_k(x_j) = n \in (p_\ell, p_{\ell+1})$, the absorption radius is $r_{\text{abs}}(x_j, k) = \frac{\min\{n - p_\ell, p_{\ell+1} - n\}}{10^k}$.*

**Proposition 1** (Absorption guarantee). *If $|\eta_j| \leq r_{\text{abs}}(x_j, k)$, then $T_k^{(b_j)}(x_j + \eta_j) = T_k^{(b_j)}(x_j)$ for all bits $b_j$.*

**Lemma 1** (Absorption bounds). *For any $x_j \in [0, 1]$, $\frac{1}{2 \cdot 10^k} \leq r_{\text{abs}}(x_j, k) \leq \frac{1}{2}$.*

**Remarks.** These results formalize *pixel-level stability*: perturbations smaller than $r_{\text{abs}}$ vanish under prime quantization, while larger ones necessarily cause a quantization change. Although Lemma 1 permits $r_{\text{abs}} \leq 1/2$, practical values are tiny (e.g., $\leq 1.8 \times 10^{-3}$ for $k = 4$). Since adversarial budgets in vision ($\epsilon \geq 1/255 \approx 3.9 \times 10^{-3}$) typically exceed these radii, most attacks

cross prime boundaries and induce detectable discrepancies between $Z$ and $P$. Even when perturbations lie near or below $r_{abs}$, clean and adversarial samples seldom quantize identically. Because prime gaps are irregular, even a $1/255$ change can cross a prime-interval midpoint under the same bit-vector $b$, yielding different prime assignments. And when some coordinates do round identically, the $Z$-space embedding remains sensitive while $P$ stays piecewise constant, producing a measurable $Z$–$P$ mismatch. Larger prime resolution $k$ further shrinks $r_{abs}$ and increases the likelihood of such discrepancies.

## 5.2 Image-Level Injectivity

While prime quantization is many-to-one *per pixel*, we must ensure it does not collapse distinct images globally. Lemma 2 shows that such collisions are exponentially unlikely.

**Lemma 2** (Collision probability). *Fix $k \geq 2$ and let $N := 10^k$. Let $x, x' \in [0,1]^d$ be two independent random images with i.i.d. pixel marginals whose densities are bounded by $\Lambda$ on $[0,1]$ (in particular, $\Lambda = 1$ for the uniform distribution). For a fixed secret bit vector $b \in \{0,1\}^d$, let $T_k^{(b)}$ be the prime–quantization transform (Definition 4). Assume the prime–gap envelope $G_k$ from Assumption 1. Then, $\Pr\big[T_k^{(b)}(x) = T_k^{(b)}(x')\big] \leq \left( \Lambda^2 \frac{G_k}{N} \right)^d$. In particular, for $\Lambda = 1$, the collision probability decays as $\left( G_k/10^k \right)^d$ in the number of pixels.*

**Remarks.** Lemma 2 establishes that global collisions are vanishingly rare. Even with pixel-level absorption, distinct images remain separable: e.g., for CIFAR-10 ($d = 3072$) and $k = 4$, $(G_k/10^k)^d$ is effectively zero. Thus, quantization is locally many-to-one but globally almost injective, ensuring discriminability while dampening small perturbations.

## 5.3 Bridge to GW Separation

Pixel-level absorption (Def. 7, Prop. 1) and image-level injectivity (Lemma 2) set the boundary conditions: if $\epsilon \leq r_{abs}$, perturbations vanish in $P$ while $Z$ still moves; if $\epsilon > r_{abs}$, quantization shifts and $P$ changes—so in both regimes $Z$ and $P$ neighborhoods diverge. To formalize these divergences, we embed samples into spatial–semantic product spaces $\mathcal{C} = (Z \times Y, d_{\mathcal{C}})$ and $\mathcal{K} = (P \times Y, d_{\mathcal{K}})$, representing each image as $(h_\theta(x), y)$ and $(T_k^{(b)}(x), y)$. By Theorem 5 and Corollary 2, the $\ell_\infty$ product is the tightest among admissible component metrics, so any discrepancy in geometry or label mass yields separation in $\mathcal{C}$ and $\mathcal{K}$. This construction underlies the GW envelope and gap theorems that follow.

## 5.4 Diameter Envelopes in $\mathcal{C}$ and $\mathcal{K}$

We now summarize the behavior of $k$-nearest neighbor diameters in the two product spaces $\mathcal{C}$ (CNN-based) and $\mathcal{K}$ (prime-quantized). For clean samples, diameters concentrate tightly around a median distance; for adversarially perturbed samples, explicit additive expansion terms appear. The full technical statements and proofs are deferred to Appendix E.

**Lemma 3** (Unified local diameter envelopes in $\mathcal{C}$ and $\mathcal{K}$). *For any confidence $\delta \in (0,1)$ and local neighborhood size $K_{lo} \geq 2$, and under the variance proxy (Assumption 2) and prime-gap sensitivity (Assumption 3) conditions (see Appendix E), the following bounds hold for clean queries $q$ and adversarial queries $\tilde{q} = q + \eta$:*

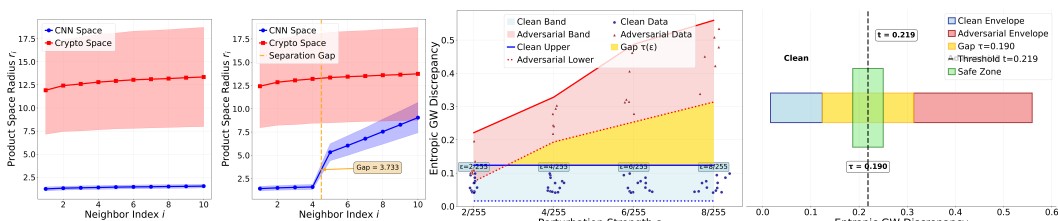

Figure 2: Geometric inconsistency detection and gap theorem validation. 1st panel: clean neighborhoods; 2nd panel: adversarial separation ($\gamma_{\mathcal{C}} = 3.567$); 3rd panel: gap theorem across $\varepsilon \in \{2/255, 4/255, 6/255, 8/255\}$; 4th panel: risk control thresholding (left:clean; right:adversarial).

$$\operatorname{diam}\left(\mathcal{N}_{K_{\text{lo}}}^{\mathcal{C}}(q)\right) \leq \underbrace{2\,\mu_{\mathcal{C}}\left(1 + \sqrt{\frac{2\log K_{\text{lo}}}{d}} + \sqrt{\frac{2\log(2/\delta)}{d}}\right)}_{=:U_{\text{clean}}^{\mathcal{C}}}, \tag{1}$$

$$\operatorname{diam}\left(\mathcal{N}_{K_{\text{lo}}}^{\mathcal{C}}(\tilde{q})\right) \leq U_{\text{clean}}^{\mathcal{C}} + \underbrace{\frac{2\sqrt{d}\,\sigma}{\sqrt{\delta}}\,\|\eta\|_{\infty}}_{\textit{Jacobian drift}} + 2\,\mathbf{1}_{\{y \neq \hat{y}\}}, \tag{2}$$

$$\operatorname{diam}\left(\mathcal{N}_{K_{\text{lo}}}^{\mathcal{K}}(q)\right) \leq \underbrace{2\,\mu_{\mathcal{K}}^{\max}(q)\left(1 + \frac{C_k}{\mu_{\mathcal{K}}^{\max}(q)}\sqrt{\frac{2\log K_{\text{lo}}}{d}} + \frac{C_k}{\mu_{\mathcal{K}}^{\max}(q)}\sqrt{\frac{2\log(2/\delta)}{d}}\right)}_{=:U_{\text{clean}}^{\mathcal{K}}}, \tag{3}$$

$$\operatorname{diam}\left(\mathcal{N}_{K_{\text{lo}}}^{\mathcal{K}}(\tilde{q})\right) \leq U_{\text{clean}}^{\mathcal{K}} + \underbrace{C_k\,\sqrt{d}\left(\sqrt{2\log K_{\text{lo}}} + \sqrt{2\log(2/\delta)}\right)}_{\textit{key sensitivity}} + \frac{2\sqrt{d}\,\sigma}{\sqrt{\delta}}\,\|\eta\|_{\infty} + 2\,\mathbf{1}_{\{y \neq \hat{y}\}}. \tag{4}$$

*Here $\mu_{\mathcal{C}}$ is the clean median pairwise distance in $\mathcal{C}$, $\mu_{\mathcal{K}}^{\max}(q)$ is the maximum key-annealed median distance in $\mathcal{K}$, $\sigma$ is the variance proxy from Assumption 2, $C_k = 2G_k$ is the prime-gap sensitivity constant from Assumption 3, and $\eta$ is the adversarial perturbation.*

**Geometric Insight.** In $\mathcal{C}$, clean neighborhood diameters concentrate around a median distance (App. Lemma 6), while adversarial perturbations add a Jacobian-driven drift and a possible label-flip penalty (App. Thm. 7; cf. equation 1–equation 2). In $\mathcal{K}$, clean neighborhoods are stabilized by prime-gap sensitivity (App. Thm. 8), whereas adversarial perturbations add terms from key sensitivity, perturbation norms, and label flips (App. Thm. 9; cf. equation 3–equation 4). Together these define the *diameter envelopes* that underlie the cross-space GW theorems. Fig. 2 (1st–2nd panels) empirically confirms this: clean samples form compact clusters consistent with Thms. 6, 8, while adversarial queries induce the predicted separation gap $\gamma_{\mathcal{C}} = 3.567$ (App. Thm. 2).

## 5.5 GROMOV–WASSERSTEIN BOUNDS: CLEAN VS. ADVERSARIAL

We now move from *local geometry* ($K$-NN diameter bounds in $\mathcal{C}$ and $\mathcal{K}$, Lemma 3) to a *distributional geometry* comparison across spaces. The Gromov–Wasserstein (GW) distance aligns pairwise distance structures, allowing us to bound: (i) in the *clean case*, similarity of $\mathcal{C}$ and $\mathcal{K}$, and (ii) in the *adversarial case*, a provable increase when perturbations inflate diameters differently across spaces. This separation underlies our detection framework.

**Theorem 1** (Clean cross–space GW upper bound via $K$-NN star radii). *Fix a clean query $x$ and consider its local neighborhoods $\mathcal{N}_K^{\mathcal{C}}(x) \subset \mathcal{C}$ and $\mathcal{N}_K^{\mathcal{K}}(x) \subset \mathcal{K}$, each endowed with the uniform probability measure on $K$ points.*

*Let $R_{\mathcal{C}}$ and $R_{\mathcal{K}}$ denote the corresponding $K$-NN radii (the $K$-th star distances from $x$) in $\mathcal{C}$ and $\mathcal{K}$ respectively. Then, for any confidence levels $\delta_{\mathcal{C}}, \delta_{\mathcal{K}} \in (0,1)$, the following high-probability envelopes hold:*

$$R_{\mathcal{C}} \;\leq\; \mu_{\mathcal{C}}\left(1 + \sqrt{\tfrac{2\log K}{d}} + \sqrt{\tfrac{2\log(2/\delta_{\mathcal{C}})}{d}}\right) \quad \text{with probability} \geq 1 - \delta_{\mathcal{C}}, \tag{5}$$

$$R_{\mathcal{K}} \;\leq\; \mu_{\mathcal{K}}\left(1 + \tfrac{C_k}{\mu_{\mathcal{K}}}\sqrt{d}\left(\sqrt{2\log K} + \sqrt{2\log(2/\delta_{\mathcal{K}})}\right)\right) \quad \text{with probability} \geq 1 - \delta_{\mathcal{K}}. \tag{6}$$

*Consequently, with probability at least $1 - (\delta_{\mathcal{C}} + \delta_{\mathcal{K}})$,*

$$\mathrm{GW}^2\!\left(\mathcal{N}_K^{\mathcal{C}}(x), \mathcal{N}_K^{\mathcal{K}}(x)\right) \;\leq\; 4\!\left(1 - \tfrac{1}{K}\right)\left(R_{\mathcal{C}} + R_{\mathcal{K}}\right)^2. \tag{7}$$

Full proof and derivation of radius envelopes equation 5–equation 6 are given in Appendix G.

**Notation (adversarial queries, radii, and gap).** For a clean input $x$ and perturbation $\eta$, the adversarial query in $\mathsf{M} \in \{\mathcal{C}, \mathcal{K}\}$ is $\tilde{q}_{\mathsf{M}} = z^{\mathsf{M}}(x + \eta)$. Its $K$ nearest neighbors form $\widetilde{\mathcal{N}}_K^{\mathsf{M}}(x + \eta) = \{z_1^{\mathsf{M}}, \ldots, z_K^{\mathsf{M}}\}$ with radii $r_i^{\mathsf{M}} = d_{\mathsf{M}}(z_i^{\mathsf{M}}, \tilde{q}_{\mathsf{M}})$ and maximum $R_{\mathsf{M}}^{\mathrm{adv}} = \max_i r_i^{\mathsf{M}}$. We partition the $K$ neighbors into an inner set $L$ of size $(1 - \theta)K$ and outer set $H$ of size $\theta K$, and define the separation gap as $\gamma_{\mathsf{M}} = \min_{i \in H} r_i^{\mathsf{M}} - \max_{j \in L} r_j^{\mathsf{M}}$.

**Theorem 2** (Adversarial cross–space GW lower bound). *Fix a query $x$ and perturbation $\eta$, and consider the adversarial neighborhoods $\widetilde{\mathcal{N}}_K^{\mathcal{C}}(x + \eta)$ and $\widetilde{\mathcal{N}}_K^{\mathcal{K}}(x + \eta)$, each with uniform measure on $K$ points. Let $\gamma_{\mathcal{C}}$ be the separation gap and let $R_{\mathcal{K}}^{\mathrm{adv}}$ denote the adversarial $K$–NN radius in $\mathcal{K}$, bounded as in Theorem 8. Then, with probability at least $1 - \delta_{\mathcal{K}}^{\mathrm{env}}$, $\mathrm{GW}^2\!\left(\widetilde{\mathcal{N}}_K^{\mathcal{C}}(x + \eta), \widetilde{\mathcal{N}}_K^{\mathcal{K}}(x + \eta)\right) \geq 2\,\theta^2\left(\gamma_{\mathcal{C}} - 2R_{\mathcal{K}}^{\mathrm{adv}}\right)_+^2.$*

Full proof is provided in Appendix G.

**Mirror results.** For brevity, we omit the symmetric (i) *clean lower bounds* and (ii) *adversarial upper bounds* on GW, but detailed proofs are provided in Theorem 10 and Theorem 11 in Appendix G.

**Remarks.** Together with Theorem 1, these results establish a clear separation: GW distance is tightly bounded for clean neighborhoods but grows under adversarial perturbations whenever $\gamma_{\mathcal{C}}$ dominates $R_{\mathcal{K}}^{\mathrm{adv}}$. Empirical results in Fig. 2 (panel 3) confirm this gap theorem: the cross-space GW discrepancy increases monotonically with perturbation strength $\varepsilon$. The growth matches the theoretical scaling $\Omega(d^2\sigma^2\varepsilon^2) - O(\frac{\log K}{d})$ derived by combining our adversarial lower bound (Theorem 2) with clean concentration envelopes (Lemma 3; see Appendix G). This demonstrates that stronger perturbations amplify cross-space inconsistencies, making detection increasingly reliable.

## 5.6 GW Gap and Risk Control

**Theorem 3** (Cross–space GW gap). *With probability at least $1 - (\delta_{\mathcal{C}} + \delta_{\mathcal{K}} + \delta_{\mathcal{K}}^{\mathrm{env}} + \delta_{\mathrm{aux}})$, the clean and adversarial GW discrepancies satisfy $|\mathrm{GW}_{\mathrm{adv}}^2 - \mathrm{GW}_{\mathrm{clean}}^2| \geq \tau := \max\{\tau_{\mathrm{adv}}, \tau_{\mathrm{clean}}, 0\}$, where $\tau_{\mathrm{adv}} = L_{\mathrm{adv}} - U_{\mathrm{clean}}$ and $\tau_{\mathrm{clean}} = L_{\mathrm{clean}} - U_{\mathrm{adv}}$. Under Assumption 2, for fixed $K$ and perturbation $\|\eta\|_\infty = \varepsilon$, we obtain $\tau = \Omega(d^2\sigma^2\varepsilon^2) - O(\frac{\log K}{d})$.*

**Lemma 4** (Risk control via GW margin). *If the gap event holds with margin $\tau > 0$ and an estimator $\widehat{\mathrm{GW}_\lambda^2}$ satisfies $\Pr\!\left(|\widehat{\mathrm{GW}_\lambda^2} - \mathrm{GW}^2| \leq \tau/3\right) \geq 1 - \delta_{\mathrm{est}}$, then thresholding $\widehat{\mathrm{GW}_\lambda^2}$ at the midpoint between clean and adversarial envelopes makes no error on this event. Thus $\Pr(\text{misclassification}) \leq \Pr(E_{\mathrm{gap}}^c) + \delta_{\mathrm{est}}$.*

**From GW to entropic GW.** All bounds above were stated for quadratic $\mathrm{GW}^2$. For entropic GW $\mathrm{GW}_\lambda^2$ with $\lambda > 0$, the lower bounds remain unchanged, while the upper bounds incur only an additive $2\lambda \log K$ (Corollaries 7–8). Hence, the clean/adversarial separation guarantees extend seamlessly to the entropic case used in practice.

**Remarks.** Theorem 3 certifies a provable margin: clean neighborhoods in $\mathcal{C}, \mathcal{K}$ contract to $O(\frac{\log K}{d})$, while adversarial perturbations inflate by $\Omega(d^2\sigma^2\varepsilon^2)$. Lemma 4 translates this into a statistical guarantee: once $\widehat{\mathrm{GW}_\lambda^2}$ concentrates within $\tau/3$, thresholding achieves negligible error. Empirical evidence (Fig. 2, panel 4) confirms the theory: clean and adversarial discrepancy distributions separate cleanly, validating the predicted risk bound. Proofs and full derivations are in App. G.

## 6 EMPIRICAL ANALYSIS

### 6.1 EXPERIMENTAL SETUP

**Adversarial Attacks.** We evaluate a broad suite spanning gradient-based, optimization, spatial, and perceptual perturbations: Auto-Attack (**AA**) (Croce & Hein, 2020), Carlini–Wagner (**CW**) (Carlini & Wagner, 2017), Patch (**PT**) (Brown et al., 2017), Projected Gradient Descent (**PGD**) (Madry et al., 2018), Spatial (**SA**) (Engstrom et al., 2019), Square (**SQ**) (Andriushchenko et al., 2020), Universal Perturbations (**UP**) (Moosavi-Dezfooli et al., 2017), Auto-PGD (**AP**) (Croce & Hein, 2020), Fast Gradient Sign (**FG**) (Goodfellow et al., 2015), Frequency (**FA**) (Yin et al., 2019), Gaussian Blur (**GB**) (Zhang et al., 2022), Pixel Flip (**PF**) (Su et al., 2019), Semantic Rotation (**SR**) (Hosseini & Poovendran, 2018), AdvAD (**AAD**) (Li et al., 2024), Penalizing Gradient Norm (**PGN**) (Ge et al., 2023), and Block Shuffle and Rotation (**BSR**) (Wang et al., 2024). We use these boldface abbreviations throughout tables and figures for brevity. Refer to Table 6 in Appendix H.5 for attack hyperparameter settings and defaults.

**Baseline Defenses.** We benchmark against representative detection methods: Mahalanobis Detector (**MD**) (Lee et al., 2018), Feature Squeezing (**FS**) (Xu et al., 2018), Meta-Adversarial-Detect (**MAD**) (Ma et al., 2019), MagNet (**MN**) (Meng & Chen, 2017), Multiple Perturbation Detector (**EA**) (Zhang et al., 2023), and Be Your Own Neighborhood (**BY**) (He et al., 2022).

**Evaluation Metrics.** We report the following complementary metrics. (1) *Binary detection accuracy*: overall accuracy of classifying inputs as clean or adversarial. (2) *True Positive Rate (TPR)*: fraction of adversarial samples correctly flagged as adversarial, i.e., $\mathrm{TPR} = \frac{\text{detected adversarial}}{\text{all adversarial}}$. (3) *End-to-End accuracy*: proportion of clean samples correctly classified and passed by the detector, plus adversarial samples correctly blocked; this reflects system-level robustness under attack. (4) *Precision*: fraction of samples flagged as adversarial that are truly adversarial. (5) *Recall*: identical to TPR—the fraction of adversarial samples correctly detected. (6) *F1-score*: harmonic mean of precision and recall, summarizing detection quality under imbalance. (7) *AUC-ROC*: area under the ROC curve, measuring threshold-independent separability between clean and adversarial distributions.

### 6.2 ADVERSARIAL DETECTION ACCURACY

**Setup.** We evaluate detection on the datasets CIFAR-10 Krizhevsky (2009), FMNIST Xiao et al. (2017), KMNIST Clanuwat et al. (2018), and ImageNet Deng et al. (2009) using models ResNet18 He et al. (2016) and ViT Dosovitskiy (2020), with adversarial datasets generated from the attack suite in Sec. 6.1. Detector/classifier hyperparameters and attack configurations appear in Appendix I and Appendix H.5, respectively. We note that ResNet18 on CIFAR-10 is employed as our default configuration.

**Results and analysis.**

Table 1 shows that our detector **achieves ≥ 95% binary detection** on **12 of 13 attacks**, with Gaussian blur (85.7%) as the only exception. Accuracy remains consistently high across attack families:

| Attack | Ours | MD | FS | MAD | MN |
|--------|------|------|------|------|------|
| AA | **97.9** | 68.9 | 82.7 | 52.0 | 74.1 |
| CW | **97.0** | 73.6 | 86.0 | 51.4 | 56.7 |
| PT | **98.0** | 86.4 | 67.8 | 50.7 | 57.3 |
| PGD | **97.8** | 91.3 | 74.4 | 51.1 | 81.0 |
| SA | **96.8** | 78.1 | 74.4 | 41.1 | 54.9 |
| SQ | **97.6** | 89.2 | 88.5 | 51.0 | 44.4 |
| UP | **97.8** | 66.4 | 53.7 | 50.7 | 47.9 |
| AP | **97.6** | 68.3 | 81.4 | 50.2 | 73.6 |
| FG | **98.0** | 73.8 | 60.9 | 49.6 | 44.7 |
| FA | **95.1** | 49.9 | 50.0 | 49.7 | 49.8 |
| GB | **85.7** | 49.8 | 51.7 | 48.2 | 48.6 |
| PF | **97.0** | 51.7 | 51.4 | 49.5 | 49.4 |
| SR | **95.9** | 50.3 | 52.7 | 50.1 | 49.1 |

Table 1: Binary detection accuracy (%). Best results are in **bold** and second best are underlined.

ilies: (i) *Gradient/optimization* (AA, CW, PGD, AP): **96–98%**, with margins of **+10–20** points, since

| Attack | Model | Ours | MD | FS | MAD | MN | EA | BY |
|--------|-------|------|-----|-----|-----|-----|-----|-----|
| **PGD** | ResNet-18 | **0.97** | 0.91 | 0.71 | 0.25 | 0.78 | 0.96 | 0.70 |
|         | ViT | **0.95** | 0.65 | 0.75 | 0.57 | 0.01 | **0.95** | 0.80 |
| **SQ** | ResNet-18 | **0.96** | 0.89 | 0.85 | 0.24 | 0.01 | 0.90 | 0.57 |
|        | ViT | **0.95** | 0.66 | 0.86 | 0.56 | 0.02 | 0.89 | 0.61 |
| **PT** | ResNet-18 | **0.98** | 0.86 | 0.54 | 0.26 | 0.03 | 0.89 | 0.80 |
|        | ViT | **0.95** | 0.66 | 0.67 | 0.54 | 0.01 | 0.89 | 0.81 |
| **AAD** | ResNet-18 | **0.95** | 0.31 | 0.46 | 0.52 | 0.13 | 0.91 | 0.59 |
|         | ViT | **0.93** | 0.67 | 0.57 | 0.64 | 0.01 | 0.92 | 0.82 |
| **PGN** | ResNet-18 | **0.95** | 0.63 | 0.62 | 0.24 | 0.01 | 0.90 | 0.77 |
|         | ViT | **0.96** | 0.67 | 0.62 | 0.64 | 0.02 | 0.93 | 0.77 |
| **BSR** | ResNet-18 | **0.95** | 0.67 | 0.52 | 0.23 | 0.40 | 0.92 | 0.81 |
|         | ViT | **0.98** | 0.67 | 0.56 | 0.64 | 0.02 | 0.92 | 0.78 |

Table 2: F1-score comparison on CIFAR-10 across multiple attacks using ResNet-18 and ViT. Best results are in **bold**, second best are underlined.

small-norm shifts in $Z$ are often absorbed in $P$, producing sharp cross-space mismatches; (ii) *Spatial/patch* (SA, PT): **97–98%**, where local structural changes disrupt geometry differently in each space; (iii) *Transfer/decision-based* (SQ, UP): **97–98%**, where transfer-induced distortions misalign $Z$ and $P$ far more than gradient-based attacks, yielding especially large gains (**+31 points** on UP); and (iv) *Perceptual/frequency* (FA, PF, SR): **95–97%**, where frequency and semantic shifts perturb $P$'s discrete neighborhoods and $Z$'s embeddings in complementary ways, creating highly detectable discrepancies. Gaussian blur is the hardest case because it averages neighboring pixels, suppressing edges and textures, inducing similar distortions in both $Z$ and $P$. This reduces the cross-space discrepancy that our detector exploits. Nevertheless, uneven quantization in $P$ ensures residual separation, and we still outperform all baselines on blur. Full per-dataset results and additional metrics, including TPR/FPR heatmaps and end-to-end-accuracy, are reported in Table 13(Appendix L) and Appendix M respectively. Across all six attacks and both architectures (ResNet-18 and ViT), our method consistently achieves the highest F1-scores, typically exceeding **0.95**. EA generally emerges as the strongest baseline yet remains noticeably weaker than our detector, especially under patch-based and structure-altering attacks such as SQ and PT. Modern attacks such as AAD, PGN, and BSR also show the same trend: while EA or BY occasionally achieve strong second-best performance, our approach maintains a clear advantage across architectures. These results highlight the robustness and model-agnostic behavior of the proposed Z–P discrepancy framework.

### 6.3 Adaptive Attack Resistance and Ablation

**Adaptive attacks.** We test two white-box adaptive formulations: (i) cross-space ($C_{cross}$) and (ii) multi-scale ($C_{ms}$), where the adversary knows the architecture but not the secret bit vector $b^\star$. The complete formulation of the adversary's objective, the prior distribution over unknown secret bits, the consistency penalties, and the optimization procedure are provided in Appendix J. As shown in Table 3a, our method maintains strong detection (**84–90%**) across CIFAR-10, FMNIST, and KMNIST, demonstrating robustness even when defenses are explicitly targeted. This accuracy drop relative to non-adaptive attacks arises because the adversary now explicitly optimizes to *minimize cross-space discrepancies* (CNN vs. crypto features). By enforcing feature consistency under a prior over $b$, they can partially reduce the mismatches our detector relies on.

**Ablation study.** To quantify feature contributions, we compare detectors using only local GW features, only global GW features, or both. Table 3b shows that while local or global features alone yield moderate performance (**65–83%**), combining them achieves **97–98%** across all attacks. This confirms that local fine-grained cues and global structural signals are complementary.

### 6.4 Zero-Shot Setting

Adversarial robustness in large-scale Vision–Language Models (VLMs) remains relatively underexplored, especially in the *zero-shot* regime where models are accessed only through APIs and

| Dataset | $C_{cross}$ | $C_{ms}$ |
|---|---|---|
| CIFAR-10 | 86.7 | 84.5 |
| FMNIST | 89.6 | 87.9 |
| KMNIST | 88.2 | 86.8 |

(a) **Adaptive attack detection (%).**

| Features | AA | CW | PT | PGD | FG |
|---|---|---|---|---|---|
| Local only | 67.7 | 76.2 | 73.2 | 62.5 | 79.1 |
| Global only | 83.5 | 66.5 | 75.6 | 65.0 | 71.3 |
| Both | **97.9** | **97.0** | **98.0** | **97.8** | **98.0** |

(b) **Ablation on CIFAR-10 (%).**

Table 3: **Adaptive robustness and feature ablation.** (a) Our method resists adaptive white-box attacks despite defense-aware optimization. (b) Combining local and global GW features yields the strongest detection across attacks.

| Attack | Dataset | Detection Accuracy / AUC | | Precision / Recall / F1 | |
|---|---|---|---|---|---|
| | | LLaVA-1.5 | Qwen-2.7B-VL | LLaVA-1.5 | Qwen-2.7B-VL |
| APGD | CalTech-101 | 89.50 / 0.99 | 89.63 / 0.99 | 0.89 / 0.89 / 0.89 | 0.89 / 0.89 / 0.89 |
| | Food-101 | 90.44 / 0.99 | 87.72 / 0.99 | 0.90 / 0.90 / 0.90 | 0.87 / 0.87 / 0.87 |
| | CalTech-256 | 88.60 / 0.95 | 87.83 / 0.99 | 0.88 / 0.88 / 0.88 | 0.87 / 0.87 / 0.87 |
| PGD | CalTech-101 | 89.50 / 0.99 | 87.13 / 0.99 | 0.89 / 0.89 / 0.89 | 0.87 / 0.87 / 0.86 |
| | Food-101 | 87.13 / 0.99 | 88.42 / 0.99 | 0.87 / 0.87 / 0.87 | 0.88 / 0.88 / 0.88 |
| | CalTech-256 | 83.00 / 0.91 | 88.76 / 0.99 | 0.83 / 0.83 / 0.82 | 0.88 / 0.88 / 0.88 |
| FGSM | CalTech-101 | 90.80 / 0.99 | 89.20 / 0.99 | 0.90 / 0.90 / 0.90 | 0.89 / 0.89 / 0.89 |
| | Food-101 | 87.08 / 0.94 | 90.74 / 0.99 | 0.84 / 0.81 / 0.81 | 0.90 / 0.90 / 0.90 |
| | CalTech-256 | 85.50 / 0.92 | 88.07 / 0.99 | 0.85 / 0.85 / 0.85 | 0.88 / 0.88 / 0.88 |

Table 4: Zero-shot adversarial detection performance on LLaVA-1.5 and Qwen-2.7B-VL across multiple datasets and attacks.

adversaries rely on transfer attacks. This provides a natural testbed for evaluating cross-model generalization, since neither gradients nor model parameters are available.

We evaluate the zero-shot transferability of our detector across two recent VLMs—LLaVA-1.5-7B (Liu et al., 2023) and Qwen-2.7B-VL qwe (2024)—on three diverse datasets: **CalTech-101** Fei-Fei et al. (2004), **Food-101** Bossard et al. (2014), and **CalTech-256** Griffin et al. (2007). Adversarial examples are generated using PGD, APGD, and FGSM following (Cui et al., 2024), and CLIP (Radford et al., 2021) embeddings define the $Z$-space. Table 4 reports detection accuracy, AUROC, and precision/recall/F1 metrics for all model–dataset combinations.

Across all attacks and datasets, the detector achieves strong zero-shot transferability: detection accuracies are consistently $\geq 83\%$, AUROC values $\geq 0.94$ (often $\geq 0.99$), and precision/recall/F1 scores typically remain $\geq 0.87$. These results indicate robust generalization across VLM architectures without requiring access to model internals.

The robustness stems from adversarial perturbations disrupting semantic alignment in $Z$ while being unevenly absorbed in $P$, yielding a persistent cross-space discrepancy detectable even under transfer. Additional robustness and generalization results—including TPR/FPR heatmaps, cross-attack transfer, and cross-model generalization—are provided in Appendix L.

# 7 CONCLUSION

We introduced a principled framework for adversarial detection based on geometric inconsistencies between the embedding space $Z$ and a prime-quantized space $P$. Our theory shows that adversarial perturbations inevitably create detectable cross-space discrepancies, providing the first guarantees for when detection must succeed. Experiments confirm consistently high detection accuracy across diverse attacks, strong generalization to zero-shot VLMs, and robustness to adaptive adversaries. These results demonstrate that geometric reasoning offers a solid foundation for adversarial robustness. An immediate direction is to adapt our framework to multimodal models, where both adversarial pressure and generalization demands are higher. Extending our theoretical guarantess from local to global neighborhoods also forms an interesting direction for future work.

## 8 REPRODUCIBILITY STATEMENT

In accordance with the guidelines, we present all assumptions, definitions, and proofs underlying the theoretical results in Appendix C–G. Implementation details, training setups, and hyperparameters of our method are provided in Appendix K and Appendix I, enabling independent reproduction of results. Due to institutional clearance requirements, we cannot release source code at submission time, but the algorithmic descriptions and parameter specifications are sufficient to reimplement our method. We will make code available once internal review permits.

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

# APPENDIX

## A  LLM USAGE

In this work, we employed large language models (LLMs) as auxiliary tools for: (1) polishing and refining text, (2) assisting in literature search and related work, (3) formatting tables, and (4) providing coding support (e.g., debugging and boilerplate generation).

## B  NOTATION

For ease of reference, we summarize in Table 5 the main symbols and spaces used throughout the paper. Unless otherwise noted, all notation is consistent across sections.

| Symbol | Meaning |
|---|---|
| **Data & embeddings** | |
| $X = \{x_i\}_{i=1}^N$, $x_i \in [0,1]^d$ | Dataset of $N$ images (dimension $d$). |
| $Y = \{1, \ldots, C\}$, $y_i \in Y$ | Label set and label of $x_i$. |
| $h_\theta : [0,1]^d \to \mathbb{R}^m$ | CNN feature extractor. |
| $z_i = h_\theta(x_i) \in \mathbb{R}^m$ | Embedding of $x_i$. |
| $T_k^{(b)}$ | Prime quantization map with key $b$ and resolution $k$. |
| $Q^{k,b}(x)$ | Prime–quantized embedding of $x$. |
| $P$ | Prime–quantized space (Euclidean metric). |
| **Product spaces & metrics** | |
| $\mathcal{C} = (Z \times Y, d_\mathcal{C})$ | CNN–label product space; $d_\mathcal{C}((z,y),(z',y')) = \max\{\|z - z'\|_2, \mathbf{1}[y \neq y']\}$. |
| $\mathcal{K} = (P \times Y, d_\mathcal{K})$ | Prime–label product space (analogous metric). |
| **Neighborhoods & radii** | |
| $\mathcal{N}_K^\mathsf{M}(q)$ | $K$–nearest neighbors of $q$ in $\mathsf{M} \in \{\mathcal{C}, \mathcal{K}\}$. |
| $R_\mathsf{M}$, $R_\mathsf{M}^{\mathrm{adv}}$ | Clean / adversarial $K$–NN radii in $\mathsf{M}$. |
| $r_i^\mathsf{M}$ | Distance of $i$-th neighbor to $q$ in $\mathsf{M}$. |
| $\gamma_\mathsf{M}$ | Separation gap between outer and inner neighbor groups. |
| **GW quantities & envelopes** | |
| $\mathrm{GW}^2$, $\mathrm{GW}_\lambda^2$ | Quadratic and entropic GW discrepancies. |
| $D_\mathsf{M}$ | Distance matrix in space $\mathsf{M}$. |
| $\pi$ | Coupling (transport plan) in GW. |
| $U_\cdot$, $L_\cdot$ | GW upper / lower envelopes. |
| $\tau$ | Two–sided GW margin. |
| **Perturbations & constants** | |
| $\eta$, $\|\eta\|_\infty \leq \epsilon$ | Adversarial perturbation and budget. |
| $\mu_\mathcal{C}$ | Median pairwise distance in $\mathcal{C}$. |
| $\mu_\mathcal{K}^{\mathrm{key}}$, $\mu_\mathcal{K}^{\mathrm{max}}$ | Key–annealed median; maximum over dataset. |
| $\sigma$ | Variance proxy (Assumption A1). |
| $G_k$, $C_k = 2G_k$ | Prime gap bound (Dusart) and sensitivity constant. |
| $\delta$ | Confidence parameters (clean/env/grad/est/gap). |
| $d$, $m$, $C$, $N$, $K$ | Input dim, embedding dim, #classes, #samples, #neighbors. |
| $b \in \{0,1\}^d$, $k$ | Secret key bits; quantization resolution. |

Table 5: Notation summary used throughout the paper.

## C PROOFS FOR PRIME QUANTIZATION RESULTS

**Proof Roadmap.** The auxiliary results in this section establish the robustness of the prime quantization transform under bounded perturbations. We begin with Lemma 5, which shows how a perturbation of size $\epsilon$ translates into an integer drift in the scaled domain. This feeds directly into Theorem 4, which proves that whenever the perturbation budget exceeds the distance to a prime boundary, one can construct a feasible perturbation that crosses the gap, thereby changing the quantized value. To complement this, we define the absorption radius (Definition 7), derive the guarantee that perturbations below this radius are absorbed (Proposition 1), and bound the possible size of this radius in Lemma 1. Finally, Corollary 1 ties these ingredients together, yielding a crisp detection condition: perturbations below the absorption radius leave quantization unchanged, while those above it necessarily induce a detectable change.

**Lemma 5** (Perturbation budget constraint on scaled pixel values). *Let $x \in [0,1]^d$ be an image and $x_j \in [0,1]$ its $j$-th pixel. Consider a perturbation vector $\eta \in \mathbb{R}^d$ with $\|\eta\|_\infty \leq \epsilon$. Then for each pixel $x_j$,*

$$\left| S_k(x_j + \eta_j) - S_k(x_j) \right| \leq \lfloor \epsilon \cdot 10^k \rfloor + 1.$$

*Proof.* We have $S_k(x_j + \eta_j) = \lfloor (x_j + \eta_j) \cdot 10^k \rfloor = \lfloor x_j \cdot 10^k + \eta_j \cdot 10^k \rfloor$ and $S_k(x_j) = \lfloor x_j \cdot 10^k \rfloor$. Since $\|\eta\|_\infty \leq \epsilon$, it follows that $|\eta_j| \leq \epsilon$ and hence $|\eta_j \cdot 10^k| \leq \epsilon \cdot 10^k$. By the floor inequality $|\lfloor a+b \rfloor - \lfloor a \rfloor| \leq \lceil |b| \rceil$, it follows that,

$$\left| S_k(x_j + \eta_j) - S_k(x_j) \right| = \left| \lfloor x_j \cdot 10^k + \eta_j \cdot 10^k \rfloor - \lfloor x_j \cdot 10^k \rfloor \right|$$
$$\leq |\eta_j \cdot 10^k| + 1$$
$$\leq \lfloor \epsilon \cdot 10^k \rfloor + 1.$$

$\square$

**Theorem 4** ($\epsilon$-Dependent Gap-Crossing Detection). *Let $x_j \in [0,1]$ be the $j$-th pixel of an image, with $S_k(x_j) = n \in (p_l, p_{l+1})$, and let $\epsilon > 0$ be a perturbation budget such that*

$$\epsilon \cdot 10^k > \min\{\, n - p_l, \; p_{l+1} - n \,\}.$$

*Then there exists a perturbation $\eta_j$ with $|\eta_j| \leq \epsilon$ such that $S_k(x_j)$ and $S_k(x_j + \eta_j)$ lie in different prime gap intervals, and hence*

$$T_k^{(b_j)}(x_j + \eta_j) \neq T_k^{(b_j)}(x_j)$$

*for any secret bit $b_j \in \{0,1\}$.*

*Proof.* From Lemma 5, any perturbation $|\eta_j| \leq \epsilon$ induces an integer drift in the scaled domain of at most $\lfloor \epsilon \cdot 10^k \rfloor + 1$. Thus, whenever $\epsilon \cdot 10^k$ exceeds the distance from $n$ to the nearest prime boundary, some perturbation $\eta_j$ exists that pushes $S_k(x_j)$ across that boundary.

Since $S_k(x_j) = n \in (p_l, p_{l+1})$, two cases arise:

(*i*) Closer to $p_l$. If $n - p_l \leq p_{l+1} - n$ and $\epsilon \cdot 10^k > n - p_l$, choose $\eta_j < 0$ with $-\epsilon \leq \eta_j < -(n-p_l)/10^k$. Then $S_k(x_j + \eta_j) \leq \lfloor n + \eta_j \cdot 10^k \rfloor < p_l$, placing the perturbed value in $(p_{j-1}, p_l)$.

(*ii*) Closer to $p_{l+1}$. If $p_{l+1} - n < n - p_l$ and $\epsilon \cdot 10^k > p_{l+1} - n$, choose $\eta_j > 0$ with $(p_{l+1}-n)/10^k < \eta_j \leq \epsilon$. Then $S_k(x_j + \eta_j) \geq \lfloor n + \eta_j \cdot 10^k \rfloor > p_{l+1}$, placing the perturbed value in $(p_{l+1}, p_{j+2})$.

In both cases, $S_k(x_j + \eta_j)$ and $S_k(x_j)$ lie in different prime-gap intervals. Since $R_k^{(b_j)}$ rounds each integer to one of the two primes bracketing its interval, the images

$$R_k^{(b_j)}(S_k(x_j)) \in \{p_l, p_{l+1}\}, \qquad R_k^{(b_j)}(S_k(x_j + \eta_j)) \in \{p_\ell, p_{\ell+1}\}, \; \ell \neq j$$

must map to disjoint prime sets. Hence

$$T_k^{(b_j)}(x_j) = S_k^{-1}(R_k^{(b_j)}(n)) \;\neq\; S_k^{-1}(R_k^{(b_j)}(m)) = T_k^{(b_j)}(x_j + \eta_j).$$

Therefore, if $\epsilon \cdot 10^k > \min\{n - p_l, \, p_{l+1} - n\}$, some perturbation $|\eta_j| \leq \epsilon$ necessarily changes the prime quantization output, regardless of the secret bit $b_j$. $\square$

**Remark 1.** *The condition in Theorem 4 is sufficient: it ensures that some perturbation of size $\leq \epsilon$ crosses a prime boundary, though not every direction must. This simplification is enough for our later GW separation results.*

**Definition 7** (Absorption radius). *For $x_j \in [0,1]$ with $S_k(x_j) = n \in (p_\ell, p_{\ell+1})$, the absorption radius is $r_{\mathrm{abs}}(x_j, k) = \frac{\min\{n - p_\ell, \, p_{\ell+1} - n\}}{10^k}$.*

**Proposition 1** (Absorption guarantee). *If $|\eta_j| \leq r_{\mathrm{abs}}(x_j, k)$, then $T_k^{(b_j)}(x_j + \eta_j) = T_k^{(b_j)}(x_j)$ for all bits $b_j$.*

*Proof.* By Definition 7 of absorption radius $r_{\text{abs}}(x_j, k)$, we have $S_k(x_j), S_k(x_j + \eta_j) \in (p_\ell, p_{\ell+1})$ for the same prime gap interval. Since both $S_k(x_j)$ and $S_k(x_j + \eta_j)$ lie in the same prime gap $(p_\ell, p_{\ell+1})$, the prime rounding operator $R_k^{(b_j)}$ maps both to the same prime: $R_k^{(b_j)}(S_k(x_j + \eta_j)) = R_k^{(b_j)}(S_k(x_j))$.

Applying $S_k^{-1}$ to both sides:

$$T_k^{(b_j)}(x_j + \eta_j) = S_k^{-1}(R_k^{(b_j)}(S_k(x_j + \eta_j))) = S_k^{-1}(R_k^{(b_j)}(S_k(x_j))) = T_k^{(b_j)}(x_j).$$

$\square$

**Lemma 6** (Elementary Prime Gap Bound Hardy & Wright (2008)). *For any two consecutive primes $p_l < p_{l+1}$, we have the prime gap as $p_{l+1} - p_l \leq p_l$.*

**Assumption 1** (Prime-gap envelope for all $k \geq 2$). *Let $N = 10^k$. There exists an absolute constant $C_0 > 0$ covering $N < x_0 = 396{,}738$ such that*

$$G_k \;:=\; C_0 + \frac{N}{25\,(\ln N)^2}$$

*satisfies $p_{\ell+1} - p_\ell \leq G_k$ for all consecutive primes $p_\ell < p_{\ell+1} \leq N$. This is a direct consequence of Proposition 6.8 in Dusart Dusart (2010).*

**Lemma 1** (Absorption bounds). *For any $x_j \in [0, 1]$, $\frac{1}{2 \cdot 10^k} \leq r_{\text{abs}}(x_j, k) \leq \frac{1}{2}$.*

*Proof.* By Definition 7, for the $j$-th pixel $x_j \in [0, 1]$ with $S_k(x_j) = n \in (p_\ell, p_{\ell+1})$, the absorption radius is

$$r_{\text{abs}}(x_j, k) \;=\; \frac{\min\{\, n - p_\ell,\ p_{\ell+1} - n\,\}}{10^k}.$$

We will now proceed to prove each bound separately.

*(i) Lower bound.* The minimum prime gap is 1 (between 2 and 3). Hence for any $n \in (p_\ell, p_{\ell+1})$, at least one of $(n - p_\ell)$ or $(p_{\ell+1} - n)$ is at least $1/2$. Thus, $\min\{\, n - p_\ell,\ p_{\ell+1} - n\,\} \geq \frac{1}{2}$, which implies $r_{\text{abs}}(x_j, k) \geq \frac{1}{2 \cdot 10^k}$.

*(ii) Upper bound.* The maximum of $\min\{\, n - p_\ell,\ p_{\ell+1} - n\,\}$ occurs when $n$ is at the midpoint of the prime gap, i.e., $\min\{\, n - p_\ell,\ p_{\ell+1} - n\,\} \leq \frac{p_{\ell+1} - p_\ell}{2}$. Since $p_{\ell+1} \leq 10^k$ by construction, it follows that $r_{\text{abs}}(x_j, k) \leq \frac{10^k/2}{10^k} = \frac{1}{2}$. Hence, we obtain $\frac{1}{2 \cdot 10^k} \leq r_{\text{abs}}(x_j, k) \leq \frac{1}{2}$, which completes the proof. $\square$

**Corollary 1** (Absorption vs. Gap-Crossing Condition). *Let $x_j \in [0, 1]$ be a pixel and $\epsilon > 0$ a perturbation budget. Then:*

*(i) Absorption. If $\epsilon \leq r_{\text{abs}}(x_j, k)$, every $|\eta_j| \leq \epsilon$ is absorbed, i.e. $T_k^{(b_j)}(x_j + \eta_j) = T_k^{(b_j)}(x_j)$.*

*(i) Gap crossing. If $\epsilon > r_{\text{abs}}(x_j, k)$, there exists some $|\eta_j| \leq \epsilon$ for which $T_k^{(b_j)}(x_j + \eta_j) \neq T_k^{(b_j)}(x_j)$.*

*Proof.* Part *(i)* is an immediate consequence of Proposition 1. For part *(ii)*, Theorem 4 ensures that whenever $\epsilon \cdot 10^k > \min\{n - p_\ell, p_{\ell+1} - n\}$, equivalently $\epsilon > r_{\text{abs}}(x_j, k)$, one can construct a perturbation $\eta_j$ that shifts $S_k(x_j)$ into a different prime-gap interval, thereby altering the prime quantization output. $\square$

**Lemma 2** (Collision probability). *Fix $k \geq 2$ and let $N := 10^k$. Let $x, x' \in [0, 1]^d$ be two independent random images with i.i.d. pixel marginals whose densities are bounded by $\Lambda$ on $[0, 1]$ (in particular, $\Lambda = 1$ for the uniform distribution). For a fixed secret bit vector $b \in \{0, 1\}^d$, let $T_k^{(b)}$ be the prime–quantization transform (Definition 4). Assume the prime–gap envelope $G_k$ from Assumption 1. Then, $\Pr[T_k^{(b)}(x) = T_k^{(b)}(x')] \leq \left(\Lambda^2 \frac{G_k}{N}\right)^d$. In particular, for $\Lambda = 1$, the collision probability decays as $(G_k/10^k)^d$ in the number of pixels.*

*Proof.* Fix two independent images $x, x' \in [0,1]^d$ with i.i.d. pixel marginals of density at most $\Lambda$ on $[0,1]$. Let $N = 10^k$, and for each pixel index $i \in \{1, \ldots, d\}$ define the scaled integers

$$U_i := S_k(x_i) = \lfloor N x_i \rfloor, \qquad U_i' := S_k(x_i') = \lfloor N x_i' \rfloor.$$

Partition $\{0, 1, \ldots, N-1\}$ into prime-gap intervals $I_j = (p_j, p_{j+1}) \cap \{0, \ldots, N-1\}$ with lengths $g_j = |I_j|$. Since each pixel marginal has density $\leq \Lambda$, the probability of landing in any integer bin is $\leq \Lambda/N$. Therefore, for any gap $I_j$,

$$\Pr[U_i \in I_j] \leq \sum_{u \in I_j} \frac{\Lambda}{N} = \Lambda \frac{g_j}{N}, \qquad \Pr[U_i' \in I_j] \leq \Lambda \frac{g_j}{N}.$$

For a fixed secret bit $b_i$, collision occurs at pixel $i$ if both $U_i$ and $U_i'$ fall in the same gap $I_j$, since then $R_k^{(b_i)}$ maps both to the same prime. By independence of $U_i$ and $U_i'$,

$$\Pr[\text{collision at pixel } i] = \sum_j \Pr[U_i \in I_j] \Pr[U_i' \in I_j] \leq \sum_j \left( \Lambda \frac{g_j}{N} \right)^2.$$

Pixels are i.i.d. across $i$, so collisions at all $d$ coordinates occur with probability

$$\Pr\left[ T_k^{(b)}(x) = T_k^{(b)}(x') \right] \leq \left( \Lambda^2 \sum_j (g_j/N)^2 \right)^d.$$

Finally, note that $\sum_j (g_j/N)^2 \leq (\max_j g_j/N) \cdot \sum_j g_j/N \leq G_k/N$, where $G_k$ is the prime-gap envelope from Assumption 1. Hence, $\Pr\left[ T_k^{(b)}(x) = T_k^{(b)}(x') \right] \leq \left( \Lambda^2 \frac{G_k}{N} \right)^d$, yielding the stated bound. $\qquad \square$

**Remark 2** (Numerics and scope). *For $k = 3$ ($N = 10^3$) with empirical maximum gap $G_k = 36$, the per-pixel factor is $36/1000 = 0.036$, so for CIFAR-10 ($d = 3072$) the bound is at most $(0.036)^{3072} \approx 10^{-4.4 \times 10^3}$. For $k = 4$ ($N = 10^4$, $G_k = 36$), the per-pixel factor is $3.6 \times 10^{-3}$ and the overall bound is even smaller. This result is* distributional, *i.e., it certifies that collisions are exponentially unlikely for two independent draws with bounded pixel densities. It does not claim that $T_k^{(b)}$ is injective on $[0,1]^d$ (the map is many-to-one by construction). Rather, it quantifies that image-level collisions are negligible under natural sampling.*

## D  PROOFS FOR BOUNDS IN $\ell_p$ PRODUCT METRIC SPACES

**Roadmap.** In Section 5, we introduced the spatial–semantic product spaces $\mathcal{C}$ and $\mathcal{K}$, both endowed with the $\ell_\infty$ metric. The purpose of this appendix is to justify that choice. We first establish in Theorem 5 that upper bounds in an $\ell_p$ product space always imply corresponding bounds in the component spaces. We then prove in Corollary 2 that among all $\ell_p$ metrics, $\ell_\infty$ achieves the tightest possible uniform upper bound. Together these results explain why $\ell_\infty$ is the natural metric for $\mathcal{C}$ and $\mathcal{K}$, ensuring that perturbations in either spatial geometry or class distribution immediately translate into separation in the product space.

Let $(X, d_X)$ and $(Y, d_Y)$ be metric spaces. We consider their product space $W = X \times Y$ endowed with a standard $\ell_p$ product metric. For $p \in [1, \infty)$, this metric is defined by

$$d_{W,p}((x_1, y_1), (x_2, y_2)) := \left( d_X(x_1, x_2)^p + d_Y(y_1, y_2)^p \right)^{1/p}, \tag{8}$$

and for $p = \infty$ by

$$d_{W,\infty}((x_1, y_1), (x_2, y_2)) := \max \left\{ d_X(x_1, x_2), d_Y(y_1, y_2) \right\}. \tag{9}$$

We now establish a general theorem relating upper bounds in the product space $W$ to upper bounds in the component spaces $X$ and $Y$.

**Theorem 5** (Component-wise Upper Bounds from $\ell_p$ Product Metrics). *Let $(X, d_X)$ and $(Y, d_Y)$ be metric spaces, and let $W = X \times Y$ with the $\ell_p$ metric $d_{W,p}$ for some $p \in [1, \infty]$. Suppose there exists a constant $M \geq 0$ such that*

$$d_{W,p}((x_1, y_1), (x_2, y_2)) \leq M, \quad \forall (x_1, y_1), (x_2, y_2) \in W. \tag{10}$$

*Then the following component-wise bounds hold:*

$$d_X(x_1, x_2) \leq M, \quad d_Y(y_1, y_2) \leq M, \quad \forall x_1, x_2 \in X, y_1, y_2 \in Y. \tag{11}$$

*Proof.* We consider two cases:

*(i) $1 \leq p < \infty$.* By definition, for any $(x_1, y_1), (x_2, y_2) \in W$,

$$d_{W,p}((x_1, y_1), (x_2, y_2)) = \left(d_X(x_1, x_2)^p + d_Y(y_1, y_2)^p\right)^{1/p}.$$

Since $d_X(x_1, x_2)^p \geq 0$ and $d_Y(y_1, y_2)^p \geq 0$, it immediately follows that

$$d_X(x_1, x_2)^p \leq d_X(x_1, x_2)^p + d_Y(y_1, y_2)^p = d_{W,p}((x_1, y_1), (x_2, y_2))^p.$$

Taking the $p$-th root on both sides gives $d_X(x_1, x_2) \leq d_{W,p}((x_1, y_1), (x_2, y_2)) \leq M$. An identical argument applies to $d_Y(y_1, y_2)$.

*(ii) $p = \infty$.* By definition,

$$d_{W,\infty}((x_1, y_1), (x_2, y_2)) = \max\{d_X(x_1, x_2), d_Y(y_1, y_2)\}.$$

Hence, by properties of the maximum function,

$$d_X(x_1, x_2) \leq d_{W,\infty}((x_1, y_1), (x_2, y_2)) \leq M, \quad d_Y(y_1, y_2) \leq d_{W,\infty}((x_1, y_1), (x_2, y_2)) \leq M.$$

Combining the two cases, the theorem follows. $\square$

**Corollary 2** (Tightest Upper Bound in $\ell_p$ Product Spaces). *Let $(X, d_X)$ and $(Y, d_Y)$ have known upper bounds $M_X$ and $M_Y$ respectively, i.e.,*

$$d_X(x_1, x_2) \leq M_X, \quad d_Y(y_1, y_2) \leq M_Y, \quad \forall x_1, x_2 \in X, y_1, y_2 \in Y.$$

*Then the corresponding upper bound for the product space $(W, d_{W,p})$ is*

$$d_{W,p}((x_1, y_1), (x_2, y_2)) \leq \begin{cases} (M_X^p + M_Y^p)^{1/p}, & 1 \leq p < \infty, \\ \max\{M_X, M_Y\}, & p = \infty. \end{cases}$$

*Moreover, among all $\ell_p$ product metrics, the $\ell_\infty$ metric achieves the* tightest *upper bound, i.e.,*

$$\max\{M_X, M_Y\} \leq (M_X^p + M_Y^p)^{1/p}, \quad \forall p \in [1, \infty),$$

*and is therefore optimal when minimizing the guaranteed upper bound in the product space.*

*Proof.* The bound for $1 \leq p < \infty$ follows directly from the monotonicity of the $\ell_p$ norm:

$$d_{W,p}((x_1, y_1), (x_2, y_2)) \leq (M_X^p + M_Y^p)^{1/p}.$$

For $p = \infty$, by definition $d_{W,\infty} = \max\{d_X, d_Y\} \leq \max\{M_X, M_Y\}$. To see that $\ell_\infty$ is the tightest, observe that for any $p < \infty$, $(M_X^p + M_Y^p)^{1/p} \geq \max\{M_X, M_Y\}$. Equality occurs only if one of $M_X$ or $M_Y$ is zero. Hence, $\ell_\infty$ gives the smallest guaranteed upper bound over all $\ell_p$ norms. $\square$

**Definition 8** (Spatial, semantic, and product metric spaces). *Let $X \subset [0, 1]^d$ be the image space and $Y = \{1, \ldots, C\}$ the label set. For a representation map $f : X \to \mathbb{R}^m$, define the spatial metric space $(M^X, d_X)$ with $M^X = \{f(x) : x \in X\}$ and $d_X(x_1, x_2) = \|x_1 - x_2\|_2$. The semantic metric space is $(M^Y, d_Y)$ with $M^Y = Y$ and $d_Y(y_1, y_2) = \mathbf{1}[y_1 \neq y_2]$.*

*Their $\ell_\infty$ product is the metric space*

$$M^{XY} = (M^X \times M^Y, d_{XY}), \qquad d_{XY}((x_1, y_1), (x_2, y_2)) = \max\{d_X(x_1, x_2), d_Y(y_1, y_2)\}.$$

*Each image $x \in X$ embeds as $(f(x), y)$ where $y \in Y$ is its class label. Projections are defined by $\Pi_X(x, y) = x$ and $\Pi_Y(x, y) = y$. Instantiating $f = h_\theta$ yields the space $\mathcal{C}$ with spatial component $Z$, and instantiating $f = T_k^{(b)}$ yields $\mathcal{K}$ with spatial component $P$.*

Instantiating $f(x) = h_\theta(x)$ or $f(x) = T_k^{(b)}(x)$ yields the product spaces $\mathcal{C}$ and $\mathcal{K}$, respectively. These will serve as the foundation for the GW bounds in Section 5.

## E  PROOFS FOR DIAMETER BOUNDS IN $\mathcal{C}$ AND $\mathcal{K}$

**Proposition 2** (Concentration in $\ell_\infty$ product spaces). *Let $\{(X_i, d_i)\}_{i=1}^n$ be metric spaces and let $W = \prod_{i=1}^n X_i$ be endowed with the $\ell_\infty$ product metric $d_\infty\big((x_i)_{i=1}^n, (y_i)_{i=1}^n\big) := \max_{1 \le i \le n} d_i(x_i, y_i)$. Let $\mathbf{X} = (X_1, \ldots, X_n)$ be a random element of $W$ and fix reference points $m_i \in X_i$ (e.g., means or Fréchet means), writing $\mathbf{m} = (m_1, \ldots, m_n)$.*

*Assume that each coordinate concentrates around its reference point, i.e., there exist tail functions $\psi_i : (0, \infty) \to [0, 1]$ such that for all $t > 0$,*

$$\Pr\big\{d_i(X_i, m_i) \ge t\big\} \le \psi_i(t) \qquad (i = 1, \ldots, n).$$

*Then the product random element concentrates around $\mathbf{m}$ in $(W, d_\infty)$: for all $t > 0$,*

$$\Pr\Big\{d_\infty(\mathbf{X}, \mathbf{m}) \ge t\Big\} = \Pr\Big\{\max_{1 \le i \le n} d_i(X_i, m_i) \ge t\Big\} \le \sum_{i=1}^n \psi_i(t).$$

*Proof.* The event $\{d_\infty(\mathbf{X}, \mathbf{m}) \ge t\}$ equals $\{\max_i d_i(X_i, m_i) \ge t\}$, which is contained in the union $\bigcup_i \{d_i(X_i, m_i) \ge t\}$. Apply the union bound and the assumed coordinate-wise tail bounds. □

**Corollary 3.** *(1) For $n = 2$ and real-valued coordinates with $d_i(x, m) = |x - m|$, letting $M = \max\{X, Y\}$ and $m = \max\{\mathbb{E}X, \mathbb{E}Y\}$ gives*

$$\Pr\{|M - m| \ge t\} \le \Pr\{|X - \mathbb{E}X| \ge t\} + \Pr\{|Y - \mathbb{E}Y| \ge t\}.$$

**Theorem 6** (CNN Product Space Clean Diameter Bounds). *Let $\mathcal{C} = (Z \times Y, d_\mathcal{C})$ be the CNN product space. For clean images, the $K$-nearest neighbor diameter satisfies, for any $\delta \in (0, 1)$,*

$$\mathbb{P}\left[\mathrm{diam}\left(\mathcal{N}_k^\mathcal{C}(x_{\mathrm{clean}})\right) \le 2\mu_\mathcal{C}\left(1 + \sqrt{\tfrac{2\log K}{d}} + \sqrt{\tfrac{2\log(2/\delta)}{d}}\right)\right] \ge 1 - \delta, \qquad (12)$$

*where $\mu_\mathcal{C}$ is the clean median pairwise distance in $\mathcal{C}$ and $d$ is the spatial feature dimension of $Z$.*

*Proof.* We first establish that distances between clean embeddings in the product space $\mathcal{C}$ satisfy sub-Gaussian concentration properties. This will serve as the foundation for bounding $K$-NN diameters. Recall the definition of a sub-Gaussian random variable.

**Definition (Sub-Gaussian random variable).** A real random variable $X$ is called *sub-Gaussian* with parameter $\sigma^2$ if for all $t \in \mathbb{R}$,

$$\mathbb{E}[e^{tX}] \le \exp\Big(\tfrac{\sigma^2 t^2}{2}\Big).$$

Equivalently, its tail probabilities satisfy $\mathbb{P}[\,|X - \mathbb{E}[X]| \ge t\,] \le 2\exp\Big(-\tfrac{t^2}{2\sigma^2}\Big)$.

For a random vector $Z = (Z_1, \ldots, Z_d) \in \mathbb{R}^d$, we say $Z$ is sub-Gaussian if every linear functional is sub-Gaussian:

$$\|Z\|_{\psi_2} = \sup_{u \in S^{d-1}} \|\langle Z, u\rangle\|_{\psi_2} < \infty,$$

where for a random variable $Y$, the sub-Gaussian norm is $\|Y\|_{\psi_2} = \inf\{t > 0 : \mathbb{E}[e^{Y^2/t^2}] \le 2\}$.

Each image $x$ maps to the product space via $z = (h_\theta(x), y) \in \mathcal{C}$, where $h_\theta(x) \in Z$ is its CNN embedding and $y \in Y$ its class label. For two clean images $x_i, x_j$, we denote their embeddings by $z_i, z_j$ and define the product space distance as $D_{ij} = d_\mathcal{C}(z_i, z_j)$.

We now analyze the *spatial* and *semantic* components of $D_{ij}$.

**Spatial component.** The embedding $\Pi_Z(z) \in \mathbb{R}^d$ has sub-Gaussian coordinates due to several architectural and statistical effects. Namely, batch normalization enforces near unit variance and zero mean across feature activations Ioffe & Szegedy (2015); Santurkar et al. (2018). In Poole et al. (2016); Schoenholz et al. (2017), the authors demonstrate CLT effects arise from weighted sums of many independent activations, yielding approximately Gaussian tails. Moreover, Regularization

techniques (e.g., weight decay Krogh & Hertz (1992), dropout Srivastava et al. (2014)) further constrain magnitudes, supporting sub-Gaussian tails Wager et al. (2013).

Formally, if $\sigma_Z^2$ is the empirical variance of a coordinate in $Z$, then for all $t > 0$,

$$\mathbb{P}(\,|[\Pi_Z(z)]_\ell - \mathbb{E}[\Pi_Z(z)_\ell]| \geq t\,) \;\leq\; 2\exp\left(-\frac{t^2}{2\sigma_Z^2}\right).$$

Thus $\Pi_Z(z)$ is sub-Gaussian with $\|\Pi_Z(z)\|_{\psi_2} \leq K_Z$, where $K_Z = O(\sigma_Z\sqrt{d})$. By standard results (see (Vershynin, 2018, Thm. 3.1.1)), Euclidean distances between embeddings in $Z$ concentrate sharply around their mean.

**Semantic component.** The label projection $\Pi_Y(z)$ contributes

$$d_Y(\Pi_Y(z_i), \Pi_Y(z_j)) = \mathbf{1}\{y_i \neq y_j\},$$

which is bounded in $\{0, 1\}$ and deterministic once class labels are fixed.

**Product space concentration.** Since $\mathcal{C}$ is equipped with the $\ell_\infty$ product metric,

$$d_\mathcal{C}(z_i, z_j) = \max\{d_Z(\Pi_Z(z_i), \Pi_Z(z_j)),\ d_Y(y_i, y_j)\},$$

the concentration of the spatial component transfers to the product distance (by Proposition 2). Thus deviations of $d_\mathcal{C}(z_i, z_j)$ away from its clean median $\mu_\mathcal{C}$ occur with sub-Gaussian tails: there exist constants $c, C > 0$ such that

$$\mathbb{P}(|d_\mathcal{C}(z_i, z_j) - \mu_\mathcal{C}| \;\geq\; t) \;\leq\; C\exp(-c\,d\,t^2). \tag{13}$$

When controlling the $K$-th neighbor distance, we invoke equation 13, which in turn also allows us control over the neighborhood diameter.

**$K$-NN order statistics to diameter bound**. Fix a clean query $z$ and let $D_i = d_\mathcal{C}(z, z_i)$ denote the distance between $z$ and the i.i.d. clean samples $\{z_i\}_{i=1}^n$. Let $D_{(1)} \leq \cdots \leq D_{(n)}$ denote the *order statistics*. For any threshold $\tau$, the classical characterization of order statistics (David & Nagaraja, 2003, Eq. (2.1.3)) gives

$$\left\{D_{(k)} \geq \tau\right\} \;\Longleftrightarrow\; \left\{\#\{i : D_i \geq \tau\} \;\geq\; n - k + 1\right\}. \tag{14}$$

The event on the right means that there are at least $n - k + 1$ indices for which $D_i \geq \tau$. Equivalently, there exists a subset $S \subseteq \{1, \ldots, n\}$ with $|S| = n - k + 1$ such that

$$D_i \geq \tau \quad \forall i \in S.$$

That is,

$$\{D_{(k)} \geq \tau\} \;\subseteq\; \bigcup_{\substack{S \subseteq \{1,\ldots,n\} \\ |S|=n-k+1}} \bigcap_{i \in S} \{D_i \geq \tau\}. \tag{15}$$

Applying the union bound to equation 15 yields

$$\mathbb{P}\{D_{(k)} \geq \tau\} \;\leq\; \sum_{\substack{S \subseteq \{1,\ldots,n\} \\ |S|=n-k+1}} \mathbb{P}\left(\bigcap_{i \in S} \{D_i \geq \tau\}\right). \tag{16}$$

Since the $D_i$s are i.i.d., the probability for any fixed $S$ factors as

$$\mathbb{P}\left(\bigcap_{i \in S} \{D_i \geq \tau\}\right) = \prod_{i \in S} \mathbb{P}(D_i \geq \tau) = \left(\mathbb{P}(D_i \geq \tau)\right)^{n-k+1}.$$

There are $\binom{n}{n-k+1}$ such subsets $S$. Hence equation 16 simplifies to

$$\mathbb{P}\{D_{(k)} \geq \tau\} \;\leq\; \binom{n}{n-k+1} \left(\mathbb{P}(D_i \geq \tau)\right)^{n-k+1}. \tag{17}$$

We now shift our focus to the next part, where we bound the diameter via pairwise bounds on the $K$-nearest neighbor set. Let $\mathcal{N}_k^\mathcal{C}(z) = \{z_{(1)}, \ldots, z_{(k)}\}$ be the $k$ nearest neighbors of $z$ (ties broken

arbitrarily) and consider their pairwise distances $d_{\mathcal{C}}(z_{(i)}, z_{(j)})$ for $1 \leq i < j \leq k$. By the triangle inequality,

$$d_{\mathcal{C}}(z_{(i)}, z_{(j)}) \leq d_{\mathcal{C}}(z_{(i)}, z) + d_{\mathcal{C}}(z, z_{(j)}) = D_{(i)} + D_{(j)}.$$

Using equation 13 and the fact that sums of independent sub-Gaussian random variables remain sub-Gaussian with the same $d$-scaling up to absolute constants Vershynin (2018), one obtains that there exists $c' > 0$ such that for all $t > 0$,

$$\mathbb{P}\big\{ d_{\mathcal{C}}(z_{(i)}, z_{(j)}) \geq 2\mu_{\mathcal{C}} + t \big\} \leq 2 \exp\big( -c'\, d\, t^2 \big). \tag{18}$$

Note that the inequality equation 18 is an upper bound that does not use any special property of the indices beyond being distinct sample points. Indeed, selecting nearest neighbors to $z$ can only *decrease* the chance that their mutual distance is large.

Applying the union bound over the $\binom{k}{2}$ unordered pairs inside $\mathcal{N}_k^{\mathcal{C}}(z)$ as proposed in Boucheron et al. (2013)), we arrive at

$$\mathbb{P}\big\{ \mathrm{diam}\big(\mathcal{N}_k^{\mathcal{C}}(z)\big) \geq 2\mu_{\mathcal{C}} + t \big\} \leq \binom{k}{2} \cdot 2 \exp\big( -c'\, d\, t^2 \big). \tag{19}$$

Imposing a target failure probability $\delta \in (0, 1)$ on the right-hand side and solving for $t$:

$$\binom{k}{2} \cdot 2\, e^{-c'dt^2} \leq \delta \quad \Longleftrightarrow \quad t^2 \geq \frac{\log\binom{k}{2} + \log(2/\delta)}{c'd} \tag{20}$$

$$\Rightarrow \quad t \geq \frac{1}{\sqrt{c'd}}\Big( \sqrt{2\log k} + \sqrt{2\log(2/\delta)} \Big).$$

where we used $\log\binom{k}{2} \leq 2\log k$ and $\sqrt{a+b} \leq \sqrt{a} + \sqrt{b}$.

Substituting this choice of $t$ into equation 19 yields, with probability at least $1 - \delta$,

$$\mathrm{diam}\big(\mathcal{N}_k^{\mathcal{C}}(z)\big) \leq 2\mu_{\mathcal{C}} + \frac{1}{\sqrt{c'd}}\Big( \sqrt{2\log k} + \sqrt{2\log(2/\delta)} \Big).$$

Equivalently, writing the deviation addend in a multiplicative form and *absorbing absolute constants* into the sub-Gaussian proxy (or normalizing units), one obtains the stated bound:

$$\mathrm{diam}\big(\mathcal{N}_k^{\mathcal{C}}(z)\big) \leq 2\mu_{\mathcal{C}}\Big(1 + \sqrt{\tfrac{2\log k}{d}} + \sqrt{\tfrac{2\log(2/\delta)}{d}}\Big)$$

$\square$

.

**Corollary 4** (95% confidence bound). *For confidence level $\delta = 0.05$, the $K$-nearest neighbor diameter in $\mathcal{C}$ satisfies*

$$\mathrm{diam}\big(\mathcal{N}_K^{\mathcal{C}}(z)\big) \leq 2\mu_{\mathcal{C}}\Big(1 + \sqrt{\tfrac{2\log K}{d}} + \tfrac{2.717}{\sqrt{d}}\Big),$$

*with probability at least* 95%.

**Notation.** We adopt the metric space setup of Definition 8. In particular, $\mathcal{C} = (Z \times Y, d_{\mathcal{C}})$ denotes the CNN product space and $\mathcal{K} = (P \times Y, d_{\mathcal{K}})$ the prime-quantized product space. Let $\mu_{\mathcal{C}}$ and $\mu_{\mathcal{K}}$ denote the clean median pairwise distances in $\mathcal{C}$ and $\mathcal{K}$ respectively. We consider adversarial perturbations $\eta \in \mathbb{R}^d$ with $\|\eta\|_\infty \leq \epsilon$, where $\epsilon > 0$ is the fixed attack budget.

**Definition 9** (Adversarial query). *Let $x \in X$ be a clean input with ground-truth label $y \in Y$. For an $\epsilon$-bounded perturbation $\eta \in \mathbb{R}^d$, the* adversarial query *under representation map $f : X \to \mathbb{R}^m$ is*

$$(f(x + \eta),\, y) \in M^{XY}.$$

*Thus adversarial queries live in the same product space as clean points. Note that a classifier may produce a prediction $\hat{y} \neq y$, but $\hat{y}$ is not part of the definition. Instantiating $f = h_\theta$ yields adversarial queries in $\mathcal{C}$, and instantiating $f = T_k^{(b)}$ yields adversarial queries in $\mathcal{K}$.*

**Assumption 2** (A1: variance-only control via Jacobian proxy). *Let $\Delta(\eta, x) := \|f(x+\eta) - f(x)\|_2$ be the feature displacement under perturbation $\eta$. For small perturbations with $\|\eta\|_\infty \le \epsilon$, we assume there exists a constant $\sigma^2 > 0$ (a* variance *proxy) such that*

$$\mathbb{E}\big[\Delta(\eta, x)^2\big] \le d\,\sigma^2\,\epsilon^2. \tag{21}$$

***Interpretation.*** *By first-order Taylor expansion, $f(x+\eta) - f(x) \approx J(x)\,\eta$, where $J(x) = \nabla f(x) \in \mathbb{R}^{m \times d}$ is the Jacobian of $f$ at $x$ (with $m$ the feature dimension). Thus,*

$$\Delta(\eta, x)^2 \approx \|J(x)\eta\|_2^2 = \sum_{r=1}^{m} \langle J_{r,\cdot}(x),\, \eta \rangle^2.$$

*Assumption equation 21 requires that each row $J_{r,\cdot}(x)$ has second moment bounded by $\sigma^2$, so that the expected squared shift across $m$ features grows at most linearly with $d$ (via $\|\eta\|_\infty \le \epsilon$) and quadratically with $\epsilon$.*

*By Chebyshev's inequality, for any $\delta_{\text{grad}} \in (0, 1)$,*

$$\Delta(\eta, x) \le \frac{\sqrt{d}\,\sigma}{\sqrt{\delta_{\text{grad}}}}\,\epsilon \quad \text{with probability at least } 1 - \delta_{\text{grad}}. \tag{22}$$

*Here $\delta_{\text{grad}}$ acts as a* tolerance parameter*: it specifies the probability mass we are willing to allocate to rare large deviations in feature shifts. Smaller values of $\delta_{\text{grad}}$ yield higher-probability guarantees but make the bound looser. This provides a high-probability control of adversarial feature shifts using only variance information, without assuming Lipschitz continuity or sub-Gaussianity of the Jacobian.*

**Remark 3** (Relating $\ell_\infty$ and $\ell_2$ budgets). *An $\ell_\infty$ budget $\epsilon$ implies an $\ell_2$ budget $\epsilon_2 \le \sqrt{d}\,\epsilon$, and conversely an $\ell_2$ budget $\epsilon_2$ implies $\ell_\infty$ budget $\ge \epsilon_2/\sqrt{d}$. This allows translating $\ell_2$–based results to our $\ell_\infty$ setting and vice versa.*

In the adversarial setting, the $k$-NN neighborhood can enlarge only insofar as the query itself is *displaced* relative to its clean location. Thus, bounding the query's displacement (via Assumption 2) allows us to extend the clean $k$-NN diameter bound to the adversarial case.

**Theorem 7** (Adversarial $K$-NN diameter via Theorem 6 and A1). *Fix $K \ge 2$ and confidence levels $\delta_{\text{clean}}, \delta_{\text{grad}} \in (0, 1)$. Let $x$ be a clean query and $x + \eta$ its adversarially perturbed version with $\varepsilon = \|\eta\|_\infty$. Assume Theorem 6 (clean diameter concentration) and Assumption 2 (variance-only shift). Define $\hat{y}$ as the classifier's predicted label for $x + \eta$. Then, with probability at least $1 - (\delta_{\text{clean}} + \delta_{\text{grad}})$,*

$$\text{diam}\big(\mathcal{N}_K^{\mathcal{C}}(x+\eta)\big) \le 2\,\mu_{\mathcal{C}}\Big(1 + \sqrt{\tfrac{2\log K}{d}} + \sqrt{\tfrac{2\log(2/\delta_{\text{clean}})}{d}}\Big)$$
$$+ \frac{2\,\sqrt{d}\,\sigma}{\sqrt{\delta_{\text{grad}}}}\,\varepsilon + 2\,\mathbf{1}_{\{y \ne \hat{y}\}}. \tag{23}$$

*The indicator term vanishes when the adversarial perturbation does not change the predicted label, and contributes an additional 2 otherwise.*

*Proof.* Reusing our diameter bounds in Theorem 6, for the *clean* query $x$ we have, with probability at least $1 - \delta_{\text{clean}}$,

$$\text{diam}\big(\mathcal{N}_K^{\mathcal{C}}(x)\big) \le 2\,\mu_{\mathcal{C}}\Big(1 + \sqrt{\tfrac{2\log K}{d}} + \sqrt{\tfrac{2\log(2/\delta_{\text{clean}})}{d}}\Big). \tag{24}$$

By Assumption 2 and Chebyshev, with probability at least $1 - \delta_{\text{grad}}$,

$$\|f(x+\eta) - f(x)\|_2 \le \frac{\sqrt{d}\,\sigma}{\sqrt{\delta_{\text{grad}}}}\,\varepsilon. \tag{25}$$

We now analyze how the $K$-NN neighborhood changes when we replace the clean query $z = (f(x), y)$ with its adversarially perturbed versions. Recall our notation:

$$\tilde{z} := (f(x+\eta), y), \quad \text{unsuccessful adversarial query (true label)};$$

$$\hat{z} := (f(x+\eta), \hat{y}), \quad \text{successful adversarial query (misclassified label).}$$

Let $z_{(1)}, \ldots, z_{(K)}$ denote the $K$ nearest neighbors of whichever adversarial query we use (ties arbitrary).

For any two distinct neighbors $z_{(i)}$ and $z_{(j)}$, the triangle inequality with respect to the chosen adversarial query $q \in \{\tilde{z}, \hat{z}\}$ gives

$$d_{\mathcal{C}}\big(z_{(i)}, z_{(j)}\big) \;\leq\; d_{\mathcal{C}}\big(z_{(i)}, q\big) + d_{\mathcal{C}}\big(q, z_{(j)}\big). \tag{26}$$

We next control each of the two addends above by inserting the clean query $z = (f(x), y)$ as a reference point. For the first term we write

$$d_{\mathcal{C}}\big(z_{(i)}, q\big) \;\leq\; d_{\mathcal{C}}\big(z_{(i)}, z\big) + d_{\mathcal{C}}\big(z, q\big). \tag{27}$$

Likewise, for the second term we have

$$d_{\mathcal{C}}\big(q, z_{(j)}\big) \;\leq\; d_{\mathcal{C}}\big(z, q\big) + d_{\mathcal{C}}\big(z, z_{(j)}\big). \tag{28}$$

Thus, each path from a neighbor to the adversarial query is decomposed into a *clean part* (from neighbor to $z$) plus a *shift part* (from $z$ to $q$). The size of the shift depends on which adversarial anchor $q$ is chosen

$$d_{\mathcal{C}}(z, \tilde{z}) = \|f(x+\eta) - f(x)\|_2, \tag{29}$$

because the spatial features move but the label $y$ remains unchanged. On the other hand,

$$d_{\mathcal{C}}(z, \hat{z}) \;\leq\; \|f(x+\eta) - f(x)\|_2 \;+\; \mathbf{1}_{\{y \neq \hat{y}\}}, \tag{30}$$

since the spatial features shift as before, but in addition the semantic label may flip from $y$ to $\hat{y}$, contributing an extra unit in the product metric. Substituting equation 27–equation 30 into equation 26, we obtain for any pair $i \neq j$:

$$d_{\mathcal{C}}\big(z_{(i)}, z_{(j)}\big) \leq d_{\mathcal{C}}\big(z_{(i)}, z\big) + d_{\mathcal{C}}\big(z, z_{(j)}\big) + 2\,\|f(x+\eta) - f(x)\|_2, \tag{31}$$

$$d_{\mathcal{C}}\big(z_{(i)}, z_{(j)}\big) \leq d_{\mathcal{C}}\big(z_{(i)}, z\big) + d_{\mathcal{C}}\big(z, z_{(j)}\big) + 2\,\|f(x+\eta) - f(x)\|_2 + 2\,\mathbf{1}_{\{y \neq \hat{y}\}}, \tag{32}$$

corresponding to the true-label and predicted-label anchors, respectively. Finally, maximizing over all pairs $1 \leq i < j \leq K$ yields

$$\operatorname{diam}\big(\mathcal{N}_K^{\mathcal{C}}(x+\eta)\big) \;\leq\; \operatorname{diam}\big(\mathcal{N}_K^{\mathcal{C}}(x)\big) \;+\; 2\,\|f(x+\eta) - f(x)\|_2 \;+\; 2\,\mathbf{1}_{\{y \neq \hat{y}\}}. \tag{33}$$

This shows that the adversarial $K$-NN diameter can expand relative to the clean case by at most twice the feature shift plus a discrete penalty of 2 if the adversarial perturbation also flips the predicted label.

Finally, intersecting equation 24 and equation 25 and applying the union bound gives probability $\geq 1 - (\delta_{\text{clean}} + \delta_{\text{grad}})$. On this event,

$$\operatorname{diam}\big(\mathcal{N}_K^{\mathcal{C}}(x+\eta)\big) \;\leq\; 2\,\mu_{\mathcal{C}}\Big(1 + \sqrt{\tfrac{2\log K}{d}} + \sqrt{\tfrac{2\log(2/\delta_{\text{clean}})}{d}}\Big) \;+\; \frac{2\sqrt{d}\,\sigma}{\sqrt{\delta_{\text{grad}}}}\,\varepsilon \;+\; 2\,\mathbf{1}_{\{y \neq \hat{y}\}}.$$

$$\square$$

**Assumption 3** (Prime-gap sensitivity under bit flips). *Fix $k \geq 2$ and let $G_k$ be the prime-gap envelope from Assumption 1. Then, for every coordinate $j \in \{1, \ldots, d\}$, every two images $x, x' \in X$, and any two secret keys $b, b' \in \{0, 1\}^d$ that differ only at bit $j$, we have*

$$\underbrace{\big[T_k^{(b)}(x) - T_k^{(b)}(x')\big]_j}_{\substack{\text{difference} \\ \text{before flip}}} - \underbrace{\big[T_k^{(b')}(x) - T_k^{(b')}(x')\big]_j}_{\substack{\text{difference} \\ \text{after flip}}} \;\leq\; 2\,G_k. \tag{34}$$

**Explanation.** *Flipping one bit can move the $j$-th coordinate by at most one local prime gap $\leq G_k$. For two inputs, both coordinates may shift, so by the triangle inequality their pairwise difference changes by at most $2G_k$, regardless of the absolute difference.*

**Definition 10** (Key-annealed (data-quenched) median in $\mathcal{K}$). *Fix a clean query $x \in X$ and a fixed clean image $x' \in X$ (or, more generally, a fixed dataset and query). Let $(b, k)$ denote the environment, where $b \sim \mathrm{Unif}(\{0,1\}^d)$ and $k$ is either fixed or drawn independently from a prescribed distribution $\mathcal{P}_k$. Define*

$$D^{(b,k)}(x, x') := d_{\mathcal{K}}\big(z^{(b,k)}(x),\ z^{(b,k)}(x')\big).$$

*The* key-annealed (data-quenched) median *for the pair $(x, x')$ is*

$$\mu_{\mathcal{K}}^{\mathrm{key}}(x, x') := \inf\Big\{m :\ \mathbb{P}_{(b,k)}\big[D^{(b,k)}(x, x') \le m\big] \ge \tfrac{1}{2}\Big\}.$$

*When a single symbol is used, we write $\mu_{\mathcal{K}}$ for $\mu_{\mathcal{K}}^{\mathrm{key}}$ with the convention that the probability is over $(b, k)$ only (the data are held fixed).*

For brevity, we denote the maximal key-annealed median by

$$\mu_{\mathcal{K}}^{\max}(x) := \max_{1 \le i \le n}\ \mu_{\mathcal{K}}^{\mathrm{key}}(x, x_i).$$

**Theorem 8** (Annealed-over-keys clean $K$-NN diameter in $\mathcal{K}$). *Assume the prime-gap envelope (Assumption 1) and set $C_k := 2 G_k$. Let the key $b \sim \mathrm{Unif}(\{0,1\}^d)$ and let the granularity $k$ be either fixed or drawn independently from a prescribed distribution $\mathcal{P}_k$. Fix a clean query $x$ and a fixed dataset $\{x_i\}_{i=1}^n$ (data quenched). Then for any integer $K \ge 2$ and any $\delta_{\mathrm{env}} \in (0, 1)$, with probability at least $1 - \delta_{\mathrm{env}}$ over the draw of $(b, k)$,*

$$\mathrm{diam}\big(\mathcal{N}_K^{\mathcal{K}}(x; b, k)\big) \ \le\ 2\,\mu_{\mathcal{K}}^{\max}(x)\left(1\ +\ \frac{C_k}{\mu_{\mathcal{K}}^{\max}(x)}\sqrt{\frac{2\log K}{d}}\ +\ \frac{C_k}{\mu_{\mathcal{K}}^{\max}(x)}\sqrt{\frac{2\log(2/\delta_{\mathrm{env}})}{d}}\right),$$

*where $\mu_{\mathcal{K}}^{\max}(x) := \max_{1 \le i \le n}\ \mu_{\mathcal{K}}^{\mathrm{key}}(x, x_i)$ and, if $k$ is random, one may take $C_k := 2 \sup_{k \in \mathrm{supp}(\mathcal{P}_k)} G_k$ to make the bound uniform in $k$.*

*Proof.* Fix the clean dataset $\{x_i\}_{i=1}^n$ and the clean query $x$; these are held deterministic in this theorem. The randomness comes solely from the key $b$ (and $k$ if random). We will (i) establish McDiarmid concentration for pairwise distances under random keys, (ii) apply a union bound over the $\binom{K}{2}$ neighbor pairs, and (iii) control the scale via a two-hop envelope anchored at $2\,\mu_{\mathcal{K}}^{\max}(x)$.

Fix any pair $(i, j)$ of dataset indices. For simplicity of notations, let $Q_i^b := T_k^{(b)}(x_i) \in \mathbb{R}^d$. Define the *spatial* distance under key $b$ as

$$R_{ij}(b) := \big\| Q_i^b - Q_j^b \big\|_2 = \Big( \sum_{m=1}^d \big([Q_i^b]_m - [Q_j^b]_m\big)^2 \Big)^{1/2}. \tag{35}$$

Now flip a single key bit $b_m \mapsto b_m'$ while keeping all other bits fixed. By Assumption 1, the $m$-th coordinate difference can change by at most $2 G_k$ (a single prime-gap shift per image, hence a $2 G_k$ change for a difference), while all other coordinates remain unchanged:

$$\big| [Q_i^{b'}]_m - [Q_j^{b'}]_m - \big([Q_i^b]_m - [Q_j^b]_m\big) \big| \ \le\ 2 G_k, \qquad [Q_i^{b'}]_\ell - [Q_j^{b'}]_\ell = [Q_i^b]_\ell - [Q_j^b]_\ell \quad (\ell \ne m).$$

From a vector viewpoint, let

$$v(b) := Q_i^b - Q_j^b \in \mathbb{R}^d, \qquad v(b') = v(b) + \Delta\, e_m \quad \text{with}\quad |\Delta| \le 2 G_k,$$

where $e_m$ is the $m$-th standard basis vector. Then the Euclidean norm changes by at most

$$\big| \|v(b')\|_2 - \|v(b)\|_2 \big| \ \le\ \| v(b') - v(b) \|_2 = |\Delta| \ \le\ 2 G_k \ =:\ C_k. \tag{36}$$

Therefore, $R_{ij}(b)$ is *coordinate-wise $C_k$-Lipschitz* in each bit $b_m$.

Let $b = (b_1, \ldots, b_d) \in \{0, 1\}^d$ be uniformly random with independent bits. McDiarmid's inequality states: if $F(b_1, \ldots, b_d)$ satisfies $|F(b) - F(b^{(m)})| \le c_m$ whenever $b, b^{(m)}$ differ only at coordinate $m$, then for any $t > 0$,

$$\mathbb{P}\big( |F(b) - \mathbb{E}F(b)| \ \ge\ t \big) \ \le\ 2 \exp\left( -\frac{2 t^2}{\sum_{m=1}^d c_m^2} \right).$$

Applying this to $F = R_{ij}$ with $c_m = C_k$ (by equation 36) yields

$$\mathbb{P}_b\big(\,|R_{ij}(b) - \mathbb{E}_b[R_{ij}(b)]|\, \geq\, t\,\big)\ \leq\ 2\exp\Big(-\frac{2t^2}{d\,C_k^2}\Big). \tag{37}$$

Write the product embedding as $z^{(b,k)}(x)\ :=\ \big(T_k^{(b)}(x),\, y\big) \in \mathcal{K}$, where $y$ is the class label of $x$ (and similarly $y_i$ for $x_i$). Define the *product* distance under key $b$:

$$D_{ij}(b)\ :=\ d_{\mathcal{K}}\big(z^{(b,k)}(x_i),\, z^{(b,k)}(x_j)\big)\ =\ \max\Big\{\,R_{ij}(b),\, \mathbf{1}\big[y_i \neq y_j\big]\,\Big\}. \tag{38}$$

Since the label indicator does not depend on $b$ (clean case), for any $t > 0$,

$$\big|D_{ij}(b) - \mathbb{E}_b[D_{ij}(b)]\big|\ \leq\ \big|R_{ij}(b) - \mathbb{E}_b[R_{ij}(b)]\big|,$$

and therefore equation 37 implies

$$\mathbb{P}_b\big(\,|D_{ij}(b) - \mathbb{E}_b[D_{ij}(b)]|\, \geq\, t\,\big)\ \leq\ 2\exp\Big(-\frac{2t^2}{d\,C_k^2}\Big). \tag{39}$$

Let $q := z^{(b,k)}(x)$ be the (random-key) embedded query and $\mathcal{N}_K^{\mathcal{K}}(x; b, k) = \{z_{(1)}, \ldots, z_{(K)}\}$ its $K$ nearest neighbors in $d_{\mathcal{K}}$ (ties arbitrary). Write $M := \binom{K}{2}$ for the number of unordered pairs among these neighbors. For any fixed pair $(u, v)$, equation 37 gives the one-pair tail bound, where $C_k = 2G_k$ and $D_{uv}(b) := d_{\mathcal{K}}(z_{(u)}, z_{(v)})$.

By the union bound over all $M$ pairs, we have

$$\mathbb{P}_b\Big(\max_{1 \leq u < v \leq K} |D_{uv}(b) - \mathbb{E}_b[D_{uv}(b)]| > t\Big)\ \leq\ M \cdot 2\exp\Big(-\frac{2t^2}{d\,C_k^2}\Big). \tag{40}$$

Given a target failure probability $\delta_{\text{env}} \in (0, 1)$, we choose $t$ such that the RHS of equation 40 equals $\delta_{\text{env}}$:

$$M \cdot 2\exp\Big(-\frac{2t^2}{d\,C_k^2}\Big) = \delta_{\text{env}} \quad\Longleftrightarrow\quad t = C_k\sqrt{\frac{d}{2}}\sqrt{\log M + \log\frac{2}{\delta_{\text{env}}}}.$$

With this choice of $t$, equation 40 is equivalent to the deterministic-looking high-probability bound

$$\mathbb{P}_b\Big(\max_{1 \leq u < v \leq K} |D_{uv}(b) - \mathbb{E}_b[D_{uv}(b)]|\ \leq\ C_k \cdot \sqrt{\frac{d}{2}} \cdot \sqrt{\log\binom{K}{2} + \log(2/\delta_{\text{env}})}\Big)\ \geq\ 1 - \delta_{\text{env}}. \tag{41}$$

Put simply, simultaneously for all $\binom{K}{2}$ neighbor pairs, the deviation $|D_{uv}(b) - \mathbb{E}_b[D_{uv}(b)]|$ is at most the RHS of equation 41 with probability at least $1 - \delta_{\text{env}}$ (over the randomness of the key $b$).

By definition of the $K$-NN diameter, $\text{diam}\big(\mathcal{N}_K^{\mathcal{K}}(x; b, k)\big) = \max_{u < v} D_{uv}(b)$. From equation 39, with probability at least $1 - \delta_{\text{env}}$, every pairwise distance $D_{uv}(b)$ is within a fixed deviation of its expectation $\mathbb{E}_b[D_{uv}(b)]$. Therefore, simultaneously for all pairs $u < v$,

$$D_{uv}(b)\ \leq\ \mathbb{E}_b[D_{uv}(b)]\ +\ C_k\sqrt{\frac{d}{2}}\sqrt{\log\binom{K}{2} + \log(2/\delta_{\text{env}})}.$$

Maximizing over all pairs, we obtain

$$\text{diam}\big(\mathcal{N}_K^{\mathcal{K}}(x; b, k)\big)\ \leq\ \Gamma_K\ +\ C_k\sqrt{\frac{d}{2}}\sqrt{\log\binom{K}{2} + \log(2/\delta_{\text{env}})}, \tag{42}$$

where we have set $\Gamma_K := \max_{u < v} \mathbb{E}_b[D_{uv}(b)]$.

Let $q = z^{(b,k)}(x)$ be the query, and let $z_{(1)}, \ldots, z_{(K)}$ be its $K$ nearest neighbors in $d_{\mathcal{K}}$. By the triangle inequality ("two-hop routing"),

$$D_{uv}(b)\ \leq\ d_{\mathcal{K}}(z_{(u)}, q) + d_{\mathcal{K}}(q, z_{(v)})\ \leq\ 2\,D_{(K)}^{\star}(b), \tag{43}$$

where $D^\star_{(K)}(b)$ is the $K$-th nearest-neighbor radius under key $b$. It is standard in $k$-NN theory that, under mild density bounds on the underlying distribution Devroye et al. (1996), the scale of $D^\star_{(K)}$ is controlled by the same order as the (key-annealed, data-quenched) pairwise median distance. Hence we may conservatively bound

$$\Gamma_K := \max_{u<v} \mathbb{E}_b[D_{uv}(b)] \leq 2\,\mu^{\max}_{\mathcal{K}}(x). \tag{44}$$

Substituting equation 44 into equation 42 and using $\log\binom{K}{2} \leq 2\log K$ together with $\sqrt{a+b} \leq \sqrt{a} + \sqrt{b}$, we conclude that with probability at least $1 - \delta_{\text{env}}$,

$$\text{diam}\left(\mathcal{N}^{\mathcal{K}}_K(x;b,k)\right) \leq 2\,\mu^{\max}_{\mathcal{K}}(x) + C_k\,\sqrt{d}\left(\sqrt{2\log K} + \sqrt{2\log(2/\delta_{\text{env}})}\right).$$

Factoring out $2\,\mu^{\max}_{\mathcal{K}}(x)$ yields the stated bound. $\qquad\square$

**Corollary 5** (Quenched-in-key clean diameter bound). *Under the assumptions of Theorem 8, there exists a set of keys $\mathcal{G} \subseteq \{0,1\}^d$ with $\mathbb{P}_b(\mathcal{G}) \geq 1 - \delta_{\text{env}}$ such that for every $b \in \mathcal{G}$ (and the given $k$), the bound in Theorem 8 holds for the fixed key $b$ and the given clean dataset and query $x$:*

$$\text{diam}\left(\mathcal{N}^{\mathcal{K}}_K(x;b,k)\right) \leq 2\,\mu^{\max}_{\mathcal{K}}(x) + C_k\,\sqrt{d}\left(\sqrt{2\log K} + \sqrt{2\log(2/\delta_{\text{env}})}\right).$$

*Proof sketch. The set $\mathcal{G}$ is the (key, $k$)-event on which the union bound in equation 41 holds; this event has probability $\geq 1 - \delta_{\text{env}}$. On $\mathcal{G}$, the derivation of Theorem 8 is deterministic, hence the bound is valid for* every $b \in \mathcal{G}$ *(quenched).*

**Theorem 9** (Adversarial $K$-NN diameter in $\mathcal{K}$ (concise reuse)). *Fix $K \geq 2$ and $\delta_{\text{env}}, \delta_{\text{grad}} \in (0,1)$. Let $q = (Q^{k,b}(x), y)$ be the clean query and $q_\eta = (Q^{k,b}(x+\eta), \hat{y})$ the adversarial query (predicted label $\hat{y}$ may differ from $y$). Assume the prime-gap sensitivity bound with $G_k$ (Assumption 3) and let $C_k := 2G_k$. Then, with probability at least $1 - (\delta_{\text{env}} + \delta_{\text{grad}})$ over the key $b$ (and $k$, if random),*

$$\text{diam}\left(\mathcal{N}^{\mathcal{K}}_K(q_\eta)\right) \leq 2\,\mu_{\mathcal{K}} + C_k\,\sqrt{d}\left(\sqrt{2\log K} + \sqrt{2\log(2/\delta_{\text{env}})}\right) + \frac{2\sqrt{d}\,\sigma}{\sqrt{\delta_{\text{grad}}}}\,\|\eta\|_\infty + 2\,\mathbf{1}_{\{y\neq\hat{y}\}}.$$

*Proof.* We reuse the clean-case analysis verbatim with one adversarial modification. Define $R_{ij}(b) := \|Q^{k,b}(x_i) - Q^{k,b}(x_j)\|_2$. Flipping one bit $b_m$ changes the $m$-th coordinate difference by at most $2G_k$ (Assumption 3), so the squared norm changes by at most $C^2_k$. Hence $R_{ij}$ is coordinate-wise $C_k$-Lipschitz, and McDiarmid's inequality yields

$$\mathbb{P}_b\left(\left|R_{ij}(b) - \mathbb{E}_b R_{ij}(b)\right| \geq t\right) \leq 2\exp\left(-\frac{2t^2}{d\,C^2_k}\right). \tag{$\star$}$$

For the $K$ neighbors of the adversarial query we union bound equation $\star$ over all pairs. Choosing

$$t = C_k\,\sqrt{\tfrac{d}{2}}\,\sqrt{\log\binom{K}{2} + \log(2/\delta_{\text{env}})},$$

we obtain, with probability $\geq 1 - \delta_{\text{env}}$,

$$\max_{u<v}\left|D_{uv}(b) - \mathbb{E}_b D_{uv}(b)\right| \leq C_k\,\sqrt{\tfrac{d}{2}}\,\sqrt{\log\binom{K}{2} + \log(2/\delta_{\text{env}})}. \tag{$\ddagger$}$$

As in the clean proof, the triangle inequality through the query gives $\max_{u<v} \mathbb{E}_b D_{uv}(b) \leq 2\,\mathbb{E}_b D^{\text{star}}_{(K)}(b) \lesssim 2\,\mu_{\mathcal{K}}$, where $\mu_{\mathcal{K}}$ is the key-annealed median proxy from Definition 10.

Let the $K$ neighbors be taken w.r.t. $q_\eta$. For any pair $z_{(u)}, z_{(v)}$,

$$d_{\mathcal{K}}(z_{(u)}, z_{(v)}) \leq \underbrace{d_{\mathcal{K}}(z_{(u)}, q) + d_{\mathcal{K}}(q, z_{(v)})}_{\text{clean star path}} + 2\underbrace{d_{\mathcal{K}}(q, q_\eta)}_{\leq \frac{\sqrt{d}\,\sigma}{\sqrt{\delta_{\text{grad}}}}\|\eta\|_\infty + \mathbf{1}_{\{y\neq\hat{y}\}}}$$

$$\leq d_{\mathcal{K}}(z_{(u)}, q) + d_{\mathcal{K}}(q, z_{(v)}) + \frac{2\sqrt{d}\,\sigma}{\sqrt{\delta_{\text{grad}}}}\,\|\eta\|_\infty + 2\,\mathbf{1}_{\{y\neq\hat{y}\}}. \tag{45}$$

Maximizing over pairs converts the clean star radius bound into the adversarial one, with an additive $\frac{2\sqrt{d}\,\sigma}{\sqrt{\delta_{\text{grad}}}}\|\eta\|_\infty + 2\mathbf{1}_{\{y \neq \hat{y}\}}$.

Finally, combining the clean star envelope with the uniform deviation equation ‡ and adding the adversarial term from equation 45. Using $\log \binom{K}{2} \leq 2\log K$ yields the stated bound. $\qquad\square$

# F    A BRIEF PRIMER ON METRIC MEASURE SPACES AND GROMOV–WASSERSTEIN DISTANCES

## F.1    METRIC MEASURE SPACES

A *metric measure space* (mm-space) is a triple $(X, d_X, \mu_X)$, where $X$ is a Polish space, $d_X$ is a metric on $X$, and $\mu_X$ is a Borel probability measure on $X$. Intuitively, an mm-space encodes both the *geometry* (via $d_X$) and the *distribution of mass* (via $\mu_X$).

Two mm-spaces $(X, d_X, \mu_X)$ and $(Y, d_Y, \mu_Y)$ are considered *equivalent* if there exists a measure-preserving isometry $\varphi : X \to Y$, i.e. $d_X(x, x') = d_Y(\varphi(x), \varphi(x'))$ and $\mu_Y = \varphi_\# \mu_X$. This quotienting ensures that we compare spaces only up to relabeling of points.

The classical notion of distance between mm-spaces is the *Gromov–Hausdorff* distance, which measures how well two spaces can be embedded into a common metric space with small distortion. However, it is highly combinatorial and not well-suited to data applications.

## F.2    THE GROMOV–WASSERSTEIN DISTANCE

The *Gromov–Wasserstein (GW) distance* relaxes Gromov–Hausdorff by using optimal transport ideas. For two mm-spaces $(X, d_X, \mu_X)$ and $(Y, d_Y, \mu_Y)$, the squared GW distance is defined as

$$\text{GW}^2\big((X, d_X, \mu_X), (Y, d_Y, \mu_Y)\big) := \min_{\pi \in \Pi(\mu_X, \mu_Y)} \iint |d_X(x, x') - d_Y(y, y')|^2 \, d\pi(x, y) \, d\pi(x', y'),$$
(46)

where $\Pi(\mu_X, \mu_Y)$ is the set of couplings with marginals $\mu_X, \mu_Y$. Thus GW finds a soft correspondence $\pi$ between $X$ and $Y$ and penalizes discrepancies between their intra-space distances.

**Properties.**

- GW is a *metric* on the space of mm-spaces up to equivalence.
- If $X = Y$ and $d_X = d_Y$, then $\text{GW} = 0$ regardless of labeling.
- GW generalizes Wasserstein distance: if $X = Y$ as sets with the same underlying metric, then GW reduces to $W_2$.

**Statistical viewpoint.** For empirical datasets $X = \{x_i\}_{i=1}^n$, $Y = \{y_j\}_{j=1}^m$, the metric structure is given by pairwise distance matrices $(d_X(x_i, x_{i'}))$ and $(d_Y(y_j, y_{j'}))$. The GW distance then becomes a quadratic assignment problem over couplings $\pi \in \mathbb{R}^{n \times m}$ with row/column marginals $1/n, 1/m$.

## F.3    ENTROPIC GROMOV–WASSERSTEIN DISTANCE

The GW optimization in equation 46 is computationally hard due to its quadratic objective. To address this, Peyré et al. (2016) introduced the *entropic regularized GW distance*, defined as

$$\text{GW}_\gamma^2(X, Y) := \min_{\pi \in \Pi(\mu_X, \mu_Y)} \left\{ \iint |d_X(x, x') - d_Y(y, y')|^2 \, d\pi(x, y) \, d\pi(x', y') \, - \, \gamma H(\pi) \right\},$$
(47)

where $H(\pi) = -\sum_{i,j} \pi_{ij} \log \pi_{ij}$ is the Shannon entropy and $\gamma > 0$ is the regularization parameter.

**Effects of entropic regularization.**

- **Computational:** The problem becomes smooth and solvable by Sinkhorn-like iterations, scaling to tens of thousands of points.
- **Statistical:** $GW_\gamma$ inherits concentration bounds and enjoys faster empirical convergence (regularization reduces variance).
- **Geometric:** The optimal coupling $\pi$ becomes diffuse, capturing probabilistic alignments between $X$ and $Y$.

**Connections to our work.** In our setting, the product spaces $\mathcal{C}$ (CNN) and $\mathcal{K}$ (Crypto) each define mm-spaces under their product metrics and empirical measures. Our clean vs. adversarial concentration bounds on $k$-NN diameters directly control the *intra-space geometry* terms in equation 46. Thus, these results serve as building blocks for bounding clean/adversarial GW and entropic GW distances, providing rigorous separation guarantees for detection.

# G PROOFS FOR GROMOV–WASSERSTEIN BOUNDS

**Definition 11** (Quadratic Gromov–Wasserstein discrepancy). *Let $(\mathcal{X}, d_\mathcal{X}, \mu)$ and $(\mathcal{Y}, d_\mathcal{Y}, \nu)$ be metric–measure spaces. A coupling $\pi \in \Pi(\mu, \nu)$ is a probability measure on $\mathcal{X} \times \mathcal{Y}$ whose marginals are $\mu$ and $\nu$, i.e.*

$$\pi(A \times \mathcal{Y}) = \mu(A), \qquad \pi(\mathcal{X} \times B) = \nu(B) \quad \text{for all measurable } A \subseteq \mathcal{X}, \ B \subseteq \mathcal{Y}.$$

*The quadratic GW discrepancy is defined as*

$$\mathrm{GW}^2\big((\mathcal{X}, d_\mathcal{X}, \mu), (\mathcal{Y}, d_\mathcal{Y}, \nu)\big) := \inf_{\pi \in \Pi(\mu, \nu)} \mathbb{E}_{\substack{(x,y) \sim \pi \\ (x',y') \sim \pi}} \Big[ \big( d_\mathcal{X}(x, x') - d_\mathcal{Y}(y, y') \big)^2 \Big].$$

**Remark 4** (Upper bound by identity coupling). *For brevity, we write $\mathrm{GW}^2(\mathcal{X}, \mathcal{Y})$ as the quadratic GW discrepancy between two metric–measure spaces $(\mathcal{X}, d_\mathcal{X}, \mu)$ and $(\mathcal{Y}, d_\mathcal{Y}, \nu)$. Let $\mu = \frac{1}{n} \sum_{i=1}^n \delta_{x_i}$ and $\nu = \frac{1}{n} \sum_{i=1}^n \delta_{y_i}$, with distance matrices $D_\mathcal{X}[i,j] = d_\mathcal{X}(x_i, x_j)$ and $D_\mathcal{Y}[i,j] = d_\mathcal{Y}(y_i, y_j)$. Consider the identity coupling $\pi_0 = \frac{1}{n} \sum_{i=1}^n \delta_{(x_i, y_i)}$. Since $\mathrm{GW}^2$ is an infimum over couplings, evaluating at any feasible $\pi$ gives an upper bound:*

$$\mathrm{GW}^2(\mathcal{X}, \mathcal{Y}) \leq \frac{1}{n^2} \big\| D_\mathcal{X} - D_\mathcal{Y} \big\|_F^2.$$

*More generally, for any permutation $\sigma$ (with permutation matrix $P$), the coupling $\pi_\sigma = \frac{1}{n} \sum_i \delta_{(x_i, y_{\sigma(i)})}$ yields*

$$\mathrm{GW}^2(\mathcal{X}, \mathcal{Y}) \leq \frac{1}{n^2} \big\| D_\mathcal{X} - P \, D_\mathcal{Y} \, P^\top \big\|_F^2.$$

*These bounds are typically loose but serve as alignment-dependent certificates.*

**Proposition 3** (Cross-space stability under perturbations). *Let $(\mathcal{X}, d_\mathcal{X}, \mu)$ and $(\mathcal{Y}, d_\mathcal{Y}, \nu)$ be metric–measure spaces with distance matrices $D_\mathcal{X}, D_\mathcal{Y}$. Let $\tilde{D}_\mathcal{X} = D_\mathcal{X} + \Delta_\mathcal{X}$ and $\tilde{D}_\mathcal{Y} = D_\mathcal{Y} + \Delta_\mathcal{Y}$ be perturbed versions. Define the* clean offset *$A := D_\mathcal{X} - D_\mathcal{Y}$ and the* perturbation offset *$E := \Delta_\mathcal{X} - \Delta_\mathcal{Y}$. Then*

$$\Big| \mathrm{GW}^2(\tilde{\mathcal{X}}, \tilde{\mathcal{Y}}) - \mathrm{GW}^2(\mathcal{X}, \mathcal{Y}) \Big| \leq \frac{2}{n^2} \|A\|_F \|E\|_F + \frac{1}{n^2} \|E\|_F^2.$$

*Proof.* By Remark 4, evaluating both GW objectives at the identity coupling gives

$$\mathrm{GW}^2(\mathcal{X}, \mathcal{Y}) \leq \tfrac{1}{n^2} \|A\|_F^2, \qquad \mathrm{GW}^2(\tilde{\mathcal{X}}, \tilde{\mathcal{Y}}) \leq \tfrac{1}{n^2} \|A + E\|_F^2.$$

Hence

$$\Big| \mathrm{GW}^2(\tilde{\mathcal{X}}, \tilde{\mathcal{Y}}) - \mathrm{GW}^2(\mathcal{X}, \mathcal{Y}) \Big| \leq \tfrac{1}{n^2} \big| \|A + E\|_F^2 - \|A\|_F^2 \big|.$$

Expanding and applying Cauchy–Schwarz yields,

$$\big| \|A + E\|_F^2 - \|A\|_F^2 \big| = \big| 2\langle A, E \rangle + \|E\|_F^2 \big| \leq 2\|A\|_F \|E\|_F + \|E\|_F^2.$$

Substituting this bound into the previous inequality gives

$$\Big| \mathrm{GW}^2(\tilde{\mathcal{X}}, \tilde{\mathcal{Y}}) - \mathrm{GW}^2(\mathcal{X}, \mathcal{Y}) \Big| \leq \frac{2}{n^2} \|A\|_F \|E\|_F + \frac{1}{n^2} \|E\|_F^2,$$

which is the desired result. $\square$

**Theorem 1** (Clean cross–space GW upper bound via $K$-NN star radii). *Fix a clean query $x$ and consider its local neighborhoods $\mathcal{N}_K^{\mathcal{C}}(x) \subset \mathcal{C}$ and $\mathcal{N}_K^{\mathcal{K}}(x) \subset \mathcal{K}$, each endowed with the uniform probability measure on $K$ points.*

*Let $R_{\mathcal{C}}$ and $R_{\mathcal{K}}$ denote the corresponding $K$-NN radii (the $K$-th star distances from $x$) in $\mathcal{C}$ and $\mathcal{K}$ respectively. Then, for any confidence levels $\delta_{\mathcal{C}}, \delta_{\mathcal{K}} \in (0,1)$, the following high-probability envelopes hold:*

$$R_{\mathcal{C}} \leq \mu_{\mathcal{C}} \left( 1 + \sqrt{\tfrac{2 \log K}{d}} + \sqrt{\tfrac{2 \log(2/\delta_{\mathcal{C}})}{d}} \right) \quad \text{with probability} \geq 1 - \delta_{\mathcal{C}}, \tag{5}$$

$$R_{\mathcal{K}} \leq \mu_{\mathcal{K}} \left( 1 + \tfrac{C_k}{\mu_{\mathcal{K}}} \sqrt{d} \left( \sqrt{2 \log K} + \sqrt{2 \log(2/\delta_{\mathcal{K}})} \right) \right) \quad \text{with probability} \geq 1 - \delta_{\mathcal{K}}. \tag{6}$$

*Consequently, with probability at least $1 - (\delta_{\mathcal{C}} + \delta_{\mathcal{K}})$,*

$$\mathrm{GW}^2\!\left( \mathcal{N}_K^{\mathcal{C}}(x),\, \mathcal{N}_K^{\mathcal{K}}(x) \right) \leq 4 \left( 1 - \tfrac{1}{K} \right) \left( R_{\mathcal{C}} + R_{\mathcal{K}} \right)^2. \tag{7}$$

*Proof.* Let $X := \mathcal{N}_K^{\mathcal{C}}(x) = \{x_1, \ldots, x_K\}$ and $Y := \mathcal{N}_K^{\mathcal{K}}(x) = \{y_1, \ldots, y_K\}$ be the $K$ neighbors in $\mathcal{C}$ and $\mathcal{K}$, respectively, both with the uniform measure $K^{-1} \sum_{i=1}^{K} \delta_{(\cdot)}$. We denote their distance matrices as $D_{\mathcal{C}}[i,j] := d_{\mathcal{C}}(x_i, x_j)$ and $D_{\mathcal{K}}[i,j] := d_{\mathcal{K}}(y_i, y_j)$, which satisfy $D_{\mathcal{C}}[i,i] = D_{\mathcal{K}}[i,i] = 0$. By Remark 4, evaluating the quadratic GW objective at the identity coupling $\pi_0 = \frac{1}{K} \sum_{i=1}^{K} \delta_{(x_i, y_i)}$ yields

$$\mathrm{GW}^2(X,Y) \leq \frac{1}{K^2} \left\| D_{\mathcal{C}} - D_{\mathcal{K}} \right\|_F^2. \tag{48}$$

Let $q_{\mathcal{C}}$ be the clean query center in $\mathcal{C}$ and $q_{\mathcal{K}}$ the center in $\mathcal{K}$. Define the $K$–NN radii as $R_{\mathcal{C}} := \max_{1 \leq i \leq K} d_{\mathcal{C}}(x_i, q_{\mathcal{C}})$ and $R_{\mathcal{K}} := \max_{1 \leq i \leq K} d_{\mathcal{K}}(y_i, q_{\mathcal{K}})$. By the triangle inequality in each product metric, we have

$$d_{\mathcal{C}}(x_i, x_j) \leq d_{\mathcal{C}}(x_i, q_{\mathcal{C}}) + d_{\mathcal{C}}(q_{\mathcal{C}}, x_j) \leq 2R_{\mathcal{C}}, \qquad d_{\mathcal{K}}(y_i, y_j) \leq 2R_{\mathcal{K}}, \qquad (i \neq j). \tag{49}$$

Hence every *off-diagonal* entry of $D_{\mathcal{C}}$ (resp. $D_{\mathcal{K}}$) is bounded by $2R_{\mathcal{C}}$ (resp. $2R_{\mathcal{K}}$).

There are exactly $K(K-1)$ off-diagonal entries. Using equation 49 we arrive at

$$\|D_{\mathcal{C}}\|_F^2 \leq K(K-1)(2R_{\mathcal{C}})^2, \qquad \|D_{\mathcal{K}}\|_F^2 \leq K(K-1)(2R_{\mathcal{K}})^2.$$

Therefore, by the triangle inequality for $\|\cdot\|_F$,

$$\begin{aligned} \|D_{\mathcal{C}} - D_{\mathcal{K}}\|_F &\leq \|D_{\mathcal{C}}\|_F + \|D_{\mathcal{K}}\|_F \\ &\leq 2\sqrt{K(K-1)}\,(R_{\mathcal{C}} + R_{\mathcal{K}}). \end{aligned} \tag{50}$$

Substituting equation 50 into equation 48:

$$\mathrm{GW}^2(X,Y) \leq \frac{1}{K^2} \left( 2\sqrt{K(K-1)}\,(R_{\mathcal{C}} + R_{\mathcal{K}}) \right)^2 = 4 \left( 1 - \tfrac{1}{K} \right)(R_{\mathcal{C}} + R_{\mathcal{K}})^2,$$

which is exactly equation 7 and completes the proof. $\square$

**Notation (adversarial queries, radii, and separation gap).** Let $x$ be a clean image and $\eta$ a perturbation. For each space $\mathsf{M} \in \{\mathcal{C}, \mathcal{K}\}$, define the *adversarial query* as $\tilde{q}_{\mathsf{M}} := z^{\mathsf{M}}(x + \eta)$, the embedding of the perturbed image $x + \eta$ into $\mathsf{M}$. As the dataset is fixed and only the query moves, so the $K$–NN neighborhood may change membership relative to the clean case. We define the adversarial neighborhood as $\widetilde{\mathcal{N}}_K^{\mathsf{M}}(x + \eta) := \{z_1^{\mathsf{M}}, \ldots, z_K^{\mathsf{M}}\}$, which are the $K$ nearest neighbors to $\tilde{q}_{\mathsf{M}}$ under $d_{\mathsf{M}}$. The associated *adversarial radii* are $r_i^{\mathsf{M}} := d_{\mathsf{M}}(z_i^{\mathsf{M}}, \tilde{q}_{\mathsf{M}})$, and $R_{\mathsf{M}}^{\mathrm{adv}} := \max_{1 \leq i \leq K} r_i^{\mathsf{M}}$.

Adversarial shifts often produce a "cluster split" in these radii, i.e., some neighbors become unusually close to $\tilde{q}_{\mathsf{M}}$, while others remain farther away. To capture this structure, we partition $\widetilde{\mathcal{N}}_K^{\mathsf{M}}(x + \eta)$ into an inner group $L$ and an outer group $H$ of sizes $(1 - \theta)K$ and $\theta K$ by thresholding $\{r_i^{\mathsf{M}}\}$. The *adversarial separation gap* in $\mathsf{M}$ is

$$\gamma_{\mathsf{M}} := \min_{i \in H} r_i^{\mathsf{M}} - \max_{j \in L} r_j^{\mathsf{M}},$$

which is positive when the inner and outer sets are well separated.

**Theorem 2** (Adversarial cross–space GW lower bound). *Fix a query $x$ and perturbation $\eta$, and consider the adversarial neighborhoods $\widetilde{\mathcal{N}}_K^{\mathcal{C}}(x+\eta)$ and $\widetilde{\mathcal{N}}_K^{\mathcal{K}}(x+\eta)$, each with uniform measure on $K$ points. Let $\gamma_{\mathcal{C}}$ be the separation gap and let $R_{\mathcal{K}}^{\mathrm{adv}}$ denote the adversarial $K$–NN radius in $\mathcal{K}$, bounded as in Theorem 8. Then, with probability at least $1-\delta_{\mathcal{K}}^{\mathrm{env}}$, $\mathrm{GW}^2\big(\widetilde{\mathcal{N}}_K^{\mathcal{C}}(x+\eta), \widetilde{\mathcal{N}}_K^{\mathcal{K}}(x+\eta)\big) \geq 2\,\theta^2 \left(\gamma_{\mathcal{C}} - 2R_{\mathcal{K}}^{\mathrm{adv}}\right)_+^2.*

*Proof.* In any metric space $(\mathsf{M}, d)$ with center $q$, one has for all $u, v$,
$$| d(u,q) - d(v,q)| \;\leq\; d(u,v) \;\leq\; d(u,q) + d(v,q).$$

We apply this to the adversarial neighborhoods. For the $\mathcal{C}$-space, each $x_i, x_j \in \widetilde{\mathcal{N}}_K^{\mathcal{C}}(x+\eta)$ has radii $r_i^{\mathcal{C}} = d_{\mathcal{C}}(x_i, \tilde{q}_{\mathcal{C}})$, while for the $\mathcal{K}$-space, each $y_a, y_b \in \widetilde{\mathcal{N}}_K^{\mathcal{K}}(x+\eta)$ has radii $r_a^{\mathcal{K}} = d_{\mathcal{K}}(y_a, \tilde{q}_{\mathcal{K}})$.

Hence,
$$d_{\mathcal{C}}(x_i, x_j) \;\geq\; | r_i^{\mathcal{C}} - r_j^{\mathcal{C}} |, \qquad d_{\mathcal{K}}(y_a, y_b) \;\leq\; r_a^{\mathcal{K}} + r_b^{\mathcal{K}} \;\leq\; 2\,R_{\mathcal{K}}^{\mathrm{adv}}. \tag{51}$$

We now identify which neighbor pairs give us a guaranteed discrepancy. Recall that the adversarial neighborhood in $\mathcal{C}$ is partitioned into an inner set $L$ of size $(1-\theta)K$ and an outer set $H$ of size $\theta K$, with separation gap $\gamma_{\mathcal{C}} := \min_{i \in H} r_i^{\mathcal{C}} - \max_{j \in L} r_j^{\mathcal{C}} > 0$.

A natural question arises, namely, *Why focus on cross pairs?* If both indices come from $H$ (outer–outer) or both from $L$ (inner–inner), the corresponding radii may be very close, and no nontrivial separation is guaranteed. However, whenever one index $i \in H$ and the other $j \in L$, we know $r_i^{\mathcal{C}} - r_j^{\mathcal{C}} \;\geq\; \gamma_{\mathcal{C}}$.

By the triangle inequality bound equation 51,
$$d_{\mathcal{C}}(x_i, x_j) \;\geq\; | r_i^{\mathcal{C}} - r_j^{\mathcal{C}} | \;\geq\; \gamma_{\mathcal{C}}, \qquad d_{\mathcal{K}}(y_a, y_b) \;\leq\; 2\,R_{\mathcal{K}}^{\mathrm{adv}}.$$

Therefore, for any cross pair $(i,j) \in H \times L$ or $(j,i) \in L \times H$, and for *all* choices of $(a,b)$ in $\mathcal{K}$,
$$\big| d_{\mathcal{C}}(x_i, x_j) - d_{\mathcal{K}}(y_a, y_b) \big| \;\geq\; \gamma_{\mathcal{C}} - 2\,R_{\mathcal{K}}^{\mathrm{adv}}. \tag{52}$$

*How many such pairs exist?* Define $S := (H \times L) \cup (L \times H) \subset \{1, \ldots, K\}^2$. Then
$$|S| \;=\; |H||L| + |L||H| \;=\; 2|H||L| \;\geq\; 2\,\theta^2 K^2.$$

Thus, at least $2\theta^2 K^2$ ordered pairs $(i,j)$ enjoy the guaranteed discrepancy equation 52, which will next drive our GW lower bound.

For any coupling $\pi \in \Pi(\tilde{\mu}, \tilde{\nu})$ with uniform marginals, the GW objective can be written as
$$\mathrm{GW}^2(\widetilde{\mathcal{C}}, \widetilde{\mathcal{K}}) = \inf_{\pi} \sum_{i,i'} \sum_{j,j'} \pi[i,j]\pi[i',j']\Big(d_{\mathcal{C}}(x_i, x_{i'}) - d_{\mathcal{K}}(y_j, y_{j'})\Big)^2.$$

From equation 52, we have that each summand in the inner sum (over $j, j'$) is bounded below by $(\gamma_{\mathcal{C}} - 2R_{\mathcal{K}}^{\mathrm{adv}})_+^2$. Since the coupling $\pi$ has uniform marginals, the total weight assigned to the block $\{i\} \times \{i'\}$ after summing over $j, j'$ is fixed:
$$\sum_{j,j'} \pi[i,j]\pi[i',j'] = \Big(\sum_j \pi[i,j]\Big)\Big(\sum_{j'} \pi[i',j']\Big) = \frac{1}{K} \cdot \frac{1}{K} = \frac{1}{K^2}.$$

Therefore for each $(i,i') \in S$, $\sum_{j,j'} \pi[i,j]\pi[i',j'](\cdots) \;\geq\; \frac{1}{K^2}(\gamma_{\mathcal{C}} - 2R_{\mathcal{K}}^{\mathrm{adv}})_+^2$.

Summing over all $(i,i') \in S$ gives
$$\mathbb{E}_{\pi \otimes \pi}\Big[\big(d_{\mathcal{C}}(x,x') - d_{\mathcal{K}}(y,y')\big)^2\Big] \;\geq\; \frac{|S|}{K^2}(\gamma_{\mathcal{C}} - 2R_{\mathcal{K}}^{\mathrm{adv}})_+^2,$$

because contributions from pairs outside $S$ are nonnegative and can be dropped. Finally, recalling $|S| \geq 2\theta^2 K^2$, we arrive at $\mathbb{E}_{\pi \otimes \pi}[\cdots] \;\geq\; 2\theta^2(\gamma_{\mathcal{C}} - 2R_{\mathcal{K}}^{\mathrm{adv}})_+^2$. As this bound is independent of the choice of coupling, it continues to hold after taking the infimum over $\pi$. Finally, substituting the high–probability envelope for $R_{\mathcal{K}}^{\mathrm{adv}}$ from Theorem 8 yields the explicit form of the bound, which completes the proof. $\qquad\square$

**Remark 5** (Instantiating $\gamma_{\mathcal{C}}$ from clean geometry). *The abstract separation parameter $\gamma_{\mathcal{C}}$ in Theorem 2 can be linked to earlier $\mathcal{C}$ results. Let $q$ and $\tilde{q}$ be the clean and adversarial queries, with the clean $K$–NN neighborhood partitioned into inner set $L$ and outer set $H$. The clean separation gap is*

$$\Delta_{\mathrm{clean}} := \min_{i \in H} d_{\mathcal{C}}(x_i, q) \; - \; \max_{j \in L} d_{\mathcal{C}}(x_j, q).$$

*By our earlier adversarial absorption radius analysis in $\mathcal{C}$, adversarial perturbations shift each star distance by at most $\frac{\sqrt{d}\,\sigma}{\sqrt{\delta_{\mathrm{grad}}}}\,\varepsilon + \mathbf{1}_{\{c \neq \hat{c}\}}$. Hence the induced adversarial gap satisfies*

$$\gamma_{\mathcal{C}} \; \geq \; \Big( \Delta_{\mathrm{clean}} - 2 \Big( \tfrac{\sqrt{d}\,\sigma}{\sqrt{\delta_{\mathrm{grad}}}}\,\varepsilon + \mathbf{1}_{\{c \neq \hat{c}\}} \Big) \Big)_{+}.$$

*Thus $\gamma_{\mathcal{C}}$ is not an arbitrary constant: it can be certified from clean geometry plus the perturbation shift and possible label flip.*

**Corollary 6** (Explicit clean–adversarial gap across $\mathcal{C}$ and $\mathcal{K}$). *Combining Theorem 2 with Remark 5, with probability at least $1 - (\delta_{\mathcal{K}}^{\mathrm{env}} + \delta_{\mathrm{grad}})$,*

$$\mathrm{GW}^2\Big( \widetilde{\mathcal{N}}_K^{\mathcal{C}}(x + \eta), \widetilde{\mathcal{N}}_K^{\mathcal{K}}(x + \eta) \Big) \; \geq \; 2\theta^2 \Big( \underbrace{\Delta_{\mathrm{clean}}}_{\text{clean sep.}} - \underbrace{2 \Big( \tfrac{\sqrt{d}\,\sigma}{\sqrt{\delta_{\mathrm{grad}}}}\,\varepsilon + \mathbf{1}_{\{c \neq \hat{c}\}} \Big)}_{\text{adv. shift in } \mathcal{C}} - 2R_{\mathcal{K}}^{\mathrm{adv}} \Big)_{+}^2.$$

*This bound separates the clean structure, the $\mathcal{C}$–side adversarial shift, and the $\mathcal{K}$–side absorption, making the cross–space adversarial gap explicit.*

### G.1 Mirror theorems for upper and lower bounds on GW

**Theorem 10** (Clean cross–space GW lower bound). *Fix a clean query $x$ and its clean neighborhoods $\mathcal{N}_K^{\mathcal{C}}(x)$ and $\mathcal{N}_K^{\mathcal{K}}(x)$, each with uniform measure on $K$ points. Suppose the clean $\mathcal{C}$–radii around the clean center $q_{\mathcal{C}}$ exhibit a separation gap*

$$\gamma_{\mathcal{C}}^{\mathrm{clean}} \; := \; \min_{i \in H} d_{\mathcal{C}}(x_i, q_{\mathcal{C}}) \; - \; \max_{j \in L} d_{\mathcal{C}}(x_j, q_{\mathcal{C}}) \; > 0,$$

*for a partition into inner $L$ and outer $H$ of sizes $(1 - \theta)K$ and $\theta K$. Let $R_{\mathcal{K}}^{\mathrm{clean}}$ denote the clean $K$–NN radius in $\mathcal{K}$, bounded by the clean $\mathcal{K}$ envelope. Then*

$$\mathrm{GW}^2\Big( \mathcal{N}_K^{\mathcal{C}}(x), \mathcal{N}_K^{\mathcal{K}}(x) \Big) \; \geq \; 2\,\theta^2 \Big( \gamma_{\mathcal{C}}^{\mathrm{clean}} - 2R_{\mathcal{K}}^{\mathrm{clean}} \Big)_{+}^2.$$

*Proof sketch.* *Identical to Theorem 2 (Steps 1–4) with "adversarial" replaced by "clean" and $R_{\mathcal{K}}^{\mathrm{adv}}$ replaced by $R_{\mathcal{K}}^{\mathrm{clean}}$. The cross-pairs $(H \times L) \cup (L \times H)$ enforce an entrywise gap of at least $\gamma_{\mathcal{C}}^{\mathrm{clean}} - 2R_{\mathcal{K}}^{\mathrm{clean}}$; uniform marginals then yield the factor $2\theta^2$ after averaging over couplings.*

**Theorem 11** (Adversarial cross–space GW upper bound via $K$-NN star radii). *Fix a query $x$ and perturbation $\eta$, and consider the adversarial neighborhoods*

$$\widetilde{\mathcal{N}}_K^{\mathcal{C}}(x + \eta), \qquad \widetilde{\mathcal{N}}_K^{\mathcal{K}}(x + \eta),$$

*each with uniform measure on $K$ points. Let $R_{\mathcal{C}}^{\mathrm{adv}}$ and $R_{\mathcal{K}}^{\mathrm{adv}}$ denote the adversarial $K$–NN radii in $\mathcal{C}$ and $\mathcal{K}$, respectively (each bounded by the adversarial envelopes in those spaces). Then*

$$\mathrm{GW}^2\Big( \widetilde{\mathcal{N}}_K^{\mathcal{C}}(x + \eta), \widetilde{\mathcal{N}}_K^{\mathcal{K}}(x + \eta) \Big) \; \leq \; 4\Big( 1 - \tfrac{1}{K} \Big) \Big( R_{\mathcal{C}}^{\mathrm{adv}} + R_{\mathcal{K}}^{\mathrm{adv}} \Big)^2.$$

*Proof sketch.* *Copy Theorem 1 verbatim, replacing clean radii by adversarial radii. The identity coupling and Frobenius argument give $\mathrm{GW}^2 \leq K^{-2}\|D_{\mathcal{C}} - D_{\mathcal{K}}\|_F^2$. Each off-diagonal entry is at most $2R^{\mathrm{adv}}$ by triangle inequality via the adversarial centers, so $\|D_{\mathcal{C}} - D_{\mathcal{K}}\|_F \leq 2\sqrt{K(K-1)}(R_{\mathcal{C}}^{\mathrm{adv}} + R_{\mathcal{K}}^{\mathrm{adv}})$, which yields the claim.*

## G.2  GAP THEOREM IN GW

We now combine the clean and adversarial envelopes established in Theorems 1, 2, 10, and 11 to show that their GW discrepancies are separated by a margin that is nonvanishing in high dimension.

**Theorem 3** (Cross–space GW gap). *With probability at least $1 - (\delta_{\mathcal{C}} + \delta_{\mathcal{K}} + \delta_{\mathcal{K}}^{\mathrm{env}} + \delta_{\mathrm{aux}})$, the clean and adversarial GW discrepancies satisfy $|\mathrm{GW}_{\mathrm{adv}}^2 - \mathrm{GW}_{\mathrm{clean}}^2| \geq \tau := \max\{\tau_{\mathrm{adv}}, \tau_{\mathrm{clean}}, 0\}$, where $\tau_{\mathrm{adv}} = L_{\mathrm{adv}} - U_{\mathrm{clean}}$ and $\tau_{\mathrm{clean}} = L_{\mathrm{clean}} - U_{\mathrm{adv}}$. Under Assumption 2, for fixed $K$ and perturbation $\|\eta\|_\infty = \varepsilon$, we obtain $\tau = \Omega(d^2 \sigma^2 \varepsilon^2) - O(\frac{\log K}{d})$.*

*Proof.* On the joint event where all four bounds hold:

$$\mathrm{GW}_{\mathrm{adv}}^2 \ \geq \ L_{\mathrm{adv}}, \qquad \mathrm{GW}_{\mathrm{clean}}^2 \ \leq \ U_{\mathrm{clean}},$$

which implies

$$\mathrm{GW}_{\mathrm{adv}}^2 - \mathrm{GW}_{\mathrm{clean}}^2 \ \geq \ L_{\mathrm{adv}} - U_{\mathrm{clean}}.$$

Similarly,

$$\mathrm{GW}_{\mathrm{clean}}^2 \ \geq \ L_{\mathrm{clean}}, \qquad \mathrm{GW}_{\mathrm{adv}}^2 \ \leq \ U_{\mathrm{adv}},$$

which implies

$$\mathrm{GW}_{\mathrm{clean}}^2 - \mathrm{GW}_{\mathrm{adv}}^2 \ \geq \ L_{\mathrm{clean}} - U_{\mathrm{adv}}.$$

Taking the maximum of these two margins yields the result. $\square$

**Discussion.** *Clean side.* For fixed $K$, as $d \to \infty$, the clean $K$–NN radii in both $\mathcal{C}$ and $\mathcal{K}$ concentrate:

$$R_{\mathcal{C}}, R_{\mathcal{K}} \ = \ O\Big(\mu + \sqrt{\tfrac{\log K}{d}}\Big),$$

so the clean upper envelope $U_{\mathrm{clean}}$ vanishes at rate $O\big(\sqrt{\tfrac{\log K}{d}}\big)$.

*Adversarial side.* A perturbation $\|\eta\|_\infty = \varepsilon$ shifts the $K$–NN star distances in $\mathcal{C}$ by $\delta_{\mathcal{C}}(\varepsilon) = \Theta\Big(\frac{\sqrt{d}}{\sqrt{\delta_{\mathrm{grad}}}} \sigma \varepsilon\Big)$, which induces a separation gap $\gamma_{\mathcal{C}}$. Meanwhile, absorption in $\mathcal{K}$ is controlled by $R_{\mathcal{K}}^{\mathrm{adv}} = O(C_k \sqrt{d})$. Together, this yields $L_{\mathrm{adv}} = \Omega(d \sigma^2 \varepsilon^2)$.

*Gap scaling.* As we have $\tau = \Omega(d \sigma^2 \varepsilon^2) - O\Big(\frac{\log K}{d}\Big)$, so the gap is asymptotically nonvanishing: the clean side contracts while the adversarial side grows linearly in $d$. This proves robustness in high dimension.

*Implication for entropic solvers.* Because the GW gap remains bounded away from zero asymptotically, entropic relaxations that preserve relative ordering inherit the same discriminative power, justifying our detector design.

## G.3  ENTROPIC GW COROLLARIES AND RISK CONTROL

**Corollary 7** (Entropic relaxation preserves lower bounds). *For any metric–measure spaces $(\mathcal{X}, d_{\mathcal{X}}, \mu)$ and $(\mathcal{Y}, d_{\mathcal{Y}}, \nu)$ and any $\lambda > 0$,*

$$\mathrm{GW}_\lambda^2(\mathcal{X}, \mathcal{Y}) \ \geq \ \mathrm{GW}^2(\mathcal{X}, \mathcal{Y}).$$

*Hence the lower bounds of Theorems 2 and 10 remain valid verbatim under entropic GW.*

**Corollary 8** (Entropic slack in upper bounds). *For uniform marginals on $K$ points,*

$$\mathrm{GW}_\lambda^2(\mathcal{X}, \mathcal{Y}) \ \leq \ \mathrm{GW}^2(\mathcal{X}, \mathcal{Y}) \ + \ 2\lambda \log K.$$

*Thus the upper bounds of Theorems 1 and 11 hold with additive slack $2\lambda \log K$.*

**Corollary 9** (Quenched key version). *All high–probability envelopes on the $\mathcal{K}$–side radii (Theorems 8, 2, 11) were stated in the annealed sense, averaging over random keys $(b, k)$. By conditioning, the same inequalities hold for any fixed $(b, k)$ with identical probability bounds over the randomness of image sampling and adversarial perturbations.*

**Risk control**  The GW gap theorem (Theorem 3) ensures that, with high probability, the clean and adversarial discrepancies $\mathrm{GW}^2_{\mathrm{clean}}$ and $\mathrm{GW}^2_{\mathrm{adv}}$ are separated by a margin $\tau > 0$. In practice, however, we only observe the empirical, regularized estimator $\widehat{\mathrm{GW}^2_\lambda}$, which deviates from the truth due to (i) statistical sampling noise and (ii) entropic bias. If these deviations exceed $\tau$, the detector may fail. The next lemma formalizes that controlling the estimation error to $\tau/3$ suffices.

**Lemma 4** (Risk control via GW margin). *If the gap event holds with margin $\tau > 0$ and an estimator $\widehat{\mathrm{GW}^2_\lambda}$ satisfies $\Pr\big(|\widehat{\mathrm{GW}^2_\lambda} - \mathrm{GW}^2| \leq \tau/3\big) \geq 1 - \delta_{\mathrm{est}}$, then thresholding $\widehat{\mathrm{GW}^2_\lambda}$ at the midpoint between clean and adversarial envelopes makes no error on this event. Thus $\Pr(\text{misclassification}) \leq \Pr(E^c_{\mathrm{gap}}) + \delta_{\mathrm{est}}$.*

*Proof.* On the event $E_{\mathrm{gap}}$, the clean and adversarial discrepancies satisfy conditions $\mathrm{GW}^2_{\mathrm{clean}} \leq U_{\mathrm{clean}}$ and $\mathrm{GW}^2_{\mathrm{adv}} \geq L_{\mathrm{adv}}$, with $L_{\mathrm{adv}} - U_{\mathrm{clean}} \geq \tau_{\mathrm{adv}}$, and symmetrically $\mathrm{GW}^2_{\mathrm{clean}} \geq L_{\mathrm{clean}}$ and $\mathrm{GW}^2_{\mathrm{adv}} \leq U_{\mathrm{adv}}$, with $L_{\mathrm{clean}} - U_{\mathrm{adv}} \geq \tau_{\mathrm{clean}}$. By definition, $\tau = \max\{\tau_{\mathrm{adv}}, \tau_{\mathrm{clean}}\} > 0$, so there exists a threshold $t^*$ lying strictly between the clean and adversarial ranges, with a buffer of at least $\frac{\tau}{2}$ to each side.

In the case of a *clean instance*, on $E_{\mathrm{est}}$ we have that $\widehat{\mathrm{GW}^2_\lambda} \leq \mathrm{GW}^2_{\mathrm{clean}} + \frac{\tau}{3} \leq U_{\mathrm{clean}} + \frac{\tau}{3}$. Since $t^* \geq U_{\mathrm{clean}} + \frac{\tau}{2}$, we conclude $\widehat{\mathrm{GW}^2_\lambda} \leq t^* - \frac{\tau}{6} < t^*$, so the classifier correctly outputs "clean."

For an *adversarial instance*, on $E_{\mathrm{est}}$, we have $\widehat{\mathrm{GW}^2_\lambda} \geq \mathrm{GW}^2_{\mathrm{adv}} - \frac{\tau}{3} \geq L_{\mathrm{adv}} - \frac{\tau}{3}$. Since $t^* \leq L_{\mathrm{adv}} - \frac{\tau}{2}$, we similarly conclude $\widehat{\mathrm{GW}^2_\lambda} \geq t^* + \frac{\tau}{6} > t^*$, so the classifier correctly outputs "adversarial."

Thus, on $E_{\mathrm{gap}} \cap E_{\mathrm{est}}$, the plug–in classifier is error–free. Finally, since $E_{\mathrm{gap}}$ holds with probability at least $1 - \delta_{\mathrm{gap}}$ (from Theorem 3) and $E_{\mathrm{est}}$ with probability at least $1 - \delta_{\mathrm{est}}$, a union bound yields $\mathbb{P}(\text{misclassification}) \leq \delta_{\mathrm{gap}} + \delta_{\mathrm{est}}$. $\qquad\square$

**Discussion.**  Lemma 4 formalizes the transition from a *theoretical gap* to a *practical detector*. Theorem 3 ensures a margin $\tau$ exists between clean and adversarial discrepancies. The lemma shows that if the empirical entropic GW estimator concentrates within $\tau/3$ of the truth, then a midpoint threshold $t^*$ separates the two classes with zero error. The factor $1/3$ is convenient: it splits the error budget evenly, allowing statistical variance and entropic bias to each consume at most $\tau/6$. This provides a direct analogue to margin-based classifiers in statistical learning: once the theoretical gap is positive, robust classification depends only on estimator concentration, not on further geometric properties of $\mathcal{C}$ or $\mathcal{K}$. In particular, higher dimension amplifies $\tau$, so the limiting risk is controlled primarily by solver accuracy and sample complexity rather than geometry itself.

# H  DETAILS OF ADVERSARIAL IMAGE GENERATION

We focus on the following white-box and black-box attacks in this work across the supervised and zero-shot settings:

## H.1  WHITE-BOX ATTACKS

White-box attacks assume access to the internal parameters of the target model.

- **Auto Attack**  (Croce & Hein, 2020): A parameter-free ensemble attack combining four complementary attacks: APGD-CE, APGD-DLR, FAB-T, and Square Attack. The ensemble automatically selects optimal hyperparameters and provides reliable robustness evaluation without manual tuning.

- **Carlini & Wagner (C&W) Attack** Carlini & Wagner (2017): An optimization-based attack that formulates adversarial example generation as:

$$\min_\delta \|\delta\|_p + c \cdot f(x + \delta) \tag{53}$$

where $f(x + \delta) = \max(\max\{Z(x + \delta)_i : i \neq t\} - Z(x + \delta)_t, -\kappa)$ with $Z$ representing logits, $t$ the target class, and $\kappa$ the confidence parameter.

- **Projected Gradient Descent (PGD) Attack** Madry et al. (2018): An iterative first-order adversarial attack using projected gradient descent:

$$x_{t+1} = \Pi_S(x_t + \alpha \cdot \text{sign}(\nabla_x \ell(\theta, x_t, y))) \tag{54}$$

where $\Pi_S$ denotes projection onto the constraint set $S = \{x' : \|x' - x\|_\infty \leq \epsilon\}$ and $\ell$ is the loss function.

- **Auto-PGD (APGD) Attack** (Croce & Hein, 2020): An enhanced version of PGD with automatic step size adaptation and momentum. The step size is dynamically adjusted based on the loss trajectory:

$$\alpha_t = \alpha_0 \cdot \rho^{k_t} \tag{55}$$

where $k_t$ counts the number of step size reductions and $\rho = 0.75$.

- **Fast Gradient Sign Method (FGSM)** (Goodfellow et al., 2015): A single-step attack that generates adversarial examples using:

$$x_{adv} = x + \epsilon \cdot \text{sign}(\nabla_x J(\theta, x, y)) \tag{56}$$

where $J$ is the cost function used to train the neural network, $\theta$ are the model parameters, and $\epsilon$ controls the perturbation magnitude.

- **Universal Adversarial Perturbation** (Moosavi-Dezfooli et al., 2017): Generates image-agnostic perturbations that fool classifiers across different inputs:

$$\min_v \|v\|_p \text{ subject to } \mathbb{P}_{x \sim \mu}[\hat{k}(x + v) \neq \hat{k}(x)] \geq 1 - \delta \tag{57}$$

where $v$ is the universal perturbation, $\mu$ is the data distribution, and $\delta$ is the desired fooling rate.

- **Adversarial Patch Attack** Brown et al. (2017): Generates printable adversarial patches that can cause misclassification in the physical world:

$$\hat{p} = \arg\max_p \mathbb{E}_{x,t,l}[\log \Pr(\hat{y}|A(p, x, l, t))] \tag{58}$$

where $A(p, x, l, t)$ applies patch $p$ to image $x$ at location $l$ with transformation $t$, and $\hat{y}$ is the target class.

## H.2 BLACK-BOX ATTACKS

Black-box attacks operate without knowledge of internal model parameters.

- **Frequency Attack** Yin et al. (2019): Exploits the vulnerability of neural networks in the frequency domain by applying perturbations to the Fourier transform:

$$\mathcal{F}(x_{adv}) = \mathcal{F}(x) + \delta_f \tag{59}$$

where $\mathcal{F}$ denotes the Fourier transform and $\delta_f$ represents frequency-domain perturbations.

- **Square Attack** Andriushchenko et al. (2020): A query-efficient score-based black-box attack that uses random search within $\ell_p$ balls:

$$x_{t+1} = x_t + \eta_t \cdot h_t \tag{60}$$

where $h_t$ is a random direction sampled uniformly from $\{-1, +1\}^d$ and $\eta_t$ is the step size adapted based on the attack success.

- **Gaussian Blur Attack** (Zhang et al., 2022): Applies Gaussian blur to exploit the frequency bias of deep neural networks:

$$G_\sigma(x, y) = \frac{1}{2\pi\sigma^2} \exp\left(-\frac{x^2 + y^2}{2\sigma^2}\right) \tag{61}$$

where $\sigma$ controls the blur intensity and the convolution $x_{blur} = x * G_\sigma$ generates the adversarial example.

- **Semantic Rotation Attack** (Hosseini & Poovendran, 2018): Applies geometric transformations including rotations that preserve semantic content while causing misclassification:

$$R_\theta = \begin{bmatrix} \cos\theta & -\sin\theta \\ \sin\theta & \cos\theta \end{bmatrix} \tag{62}$$

  where $\theta$ represents the rotation angle applied to the input image coordinates.

- **Pixel Flip Attack** (Su et al., 2019): A sparse attack that modifies only a few pixels to cause misclassification:

$$\min |S| \text{ subject to } f(x \oplus \delta_S) \neq f(x) \tag{63}$$

  where $S$ is the set of modified pixel locations, $\delta_S$ represents the pixel modifications, and $\oplus$ denotes the modification operation.

### H.3 Adversarial Image Generation in the Supervised Setting

In the supervised setting, adversarial examples are generated against traditional classification models trained on labeled datasets. The model produces logits through a standard forward pass:

$$z = f_\theta(x), \tag{64}$$

where $f_\theta(x)$ denotes the neural network with parameters $\theta$, $x \in \mathbb{R}^{H \times W \times C}$ is the input image, and $z \in \mathbb{R}^K$ are the raw logits for $K$ classes.

The final classification layer is typically a linear transformation:

$$z = W^T h + b, \tag{65}$$

where $h$ is the penultimate layer representation, $W \in \mathbb{R}^{d \times K}$ is the weight matrix, and $b \in \mathbb{R}^K$ is the bias vector.

The predicted class probabilities are obtained via the softmax function:

$$P(y = c \mid x) = \frac{\exp(z_c)}{\sum_{j=1}^K \exp(z_j)}, \tag{66}$$

where $z_c$ is the logit for class $c$.

During adversarial attack generation, the commonly used loss function is the cross-entropy loss:

$$\mathcal{L}(x, y) = -\log P(y \mid x) = -z_y + \log\left(\sum_{j=1}^K \exp(z_j)\right). \tag{67}$$

### H.4 Adversarial Image Generation in the Zero-Shot Setting

In the zero-shot setting, adversarial examples are generated against Vision-Language Models (VLMs) using its image encoder model (in our case CLIP Radford et al. (2021)), which do not require training on the target classes. The model consists of separate image and text encoders that project inputs into a shared embedding space.

Given an input image $x \in \mathbb{R}^{H \times W \times C}$ and a set of $K$ class names $\{c_1, c_2, \ldots, c_K\}$, the zero-shot classification process proceeds as follows:

**Image Encoding:** The image encoder $E_I : \mathbb{R}^{H \times W \times C} \to \mathbb{R}^d$ maps the input image to an $\ell_2$-normalized embedding in the shared representation space:

$$v_I = \frac{E_I(x)}{\|E_I(x)\|_2}, \tag{68}$$

where $v_I \in \mathbb{R}^d$ represents the normalized image embedding with unit norm.

**Text Encoding:** For each class $c_i$, a text prompt is constructed using the template "A photo of $c_i$". The text encoder $E_T : \mathcal{V}^* \to \mathbb{R}^d$ maps each prompt to a normalized embedding in the same shared space:

$$v_{T,i} = \frac{E_T(\text{"A photo of } c_i\text{"})}{\|E_T(\text{"A photo of } c_i\text{"})\|_2}, \tag{69}$$

where $\mathcal{V}^*$ denotes the vocabulary space and $v_{T,i} \in \mathbb{R}^d$ is the normalized text embedding for class $c_i$.

**Logit Computation:** The logits are computed as the temperature-scaled cosine similarities between the image embedding and each text embedding:

$$z_i = \tau \cdot v_I^T v_{T,i} = \tau \cdot \cos(v_I, v_{T,i}), \tag{70}$$

where $\tau > 0$ is a temperature parameter that controls the sharpness of the similarity distribution. The complete logit vector is:

$$z = \tau \cdot \begin{bmatrix} v_I^T v_{T,1}, & v_I^T v_{T,2}, & \ldots, & v_I^T v_{T,K} \end{bmatrix}^T \in \mathbb{R}^K. \tag{71}$$

**Classification Decision:** The predicted class is obtained by selecting the class with the maximum logit value:

$$\hat{y} = \arg \max_{i \in \{1,\ldots,K\}} z_i = \arg \max_{i \in \{1,\ldots,K\}} v_I^T v_{T,i}. \tag{72}$$

The class posterior probabilities are obtained through softmax normalization:

$$P(y = c_i \mid x) = \frac{\exp\left(\tau \cdot v_I^T v_{T,i}\right)}{\sum_{j=1}^K \exp\left(\tau \cdot v_I^T v_{T,j}\right)}. \tag{73}$$

### H.5 Adversarial Attack Hyperparameter Selection

In this section, we provide the detailed configuration of adversarial attack methods used in our experiments (Section 6). Table 6 summarizes the set of hyperparameters chosen for each attack. These values are selected following common practice in the adversarial robustness literature to ensure a fair comparison across methods.

We consider a diverse set of attack strategies, including optimization-based, gradient-based, score-based, and patch-based approaches. For gradient-based methods such as PGD, Auto-PGD, FGSM, and Square Attack, we evaluate under multiple perturbation budgets with $\epsilon \in \{4/255, 8/255\}$. Universal Perturbation is evaluated with a wider range of perturbation strengths, namely $\epsilon \in \{4/255, 8/255, 12/255\}$.

For optimization-based attacks, the Carlini & Wagner (CW) attack is configured with confidence parameter $\kappa = 0.0$, following the default setting to generate minimally perturbed adversarial examples. AutoAttack is evaluated under $\epsilon \in \{4/255, 8/255\}$, consistent with its standardized benchmark protocol.

Patch-based and spatial transformations are included to account for more physically realizable adversarial scenarios. The Patch Attack is tested with square patches of shape $(3, 8, 8)$ and $(3, 16, 16)$, while the Spatial Attack allows for up to $30°$ rotation and translations of up to $10\%$ of the image dimensions.

In addition to the attacks described above, we also include several specialized perturbations and image corruptions to probe robustness across different perturbation modalities: a Frequency Attack with noise strength noise_strength $= 0.05$, Gaussian Blur with $\sigma = 1.0$ (blur_type set to "uniform"), Pixel Flip with num_pixel $= 5$ and attack mode set to "random", and a Semantic Rotation with angle $= 8°$. These parameter choices are summarized in Table 6 and were chosen to reflect common settings used in prior work while providing a broad coverage of perturbation types.

These configurations ensure that the evaluation captures a broad spectrum of attack types, ranging from small-norm pixel perturbations to structured and geometric transformations. See Table 6 for an overview of the attack methods and the full set of parameter values used in our experiments.

| Attack Method | Parameter Values |
|---|---|
| **AutoAttack (AA)** | $\epsilon = \mathbf{4/225}, 8/225$ |
| **C&W (CW)** | $\kappa = \mathbf{0.0}$ |
| **Patch Attack (PT)** | $\text{patch\_shape} = (\mathbf{3, 8, 8}), (3, 16, 16)$ |
| **PGD** | $\epsilon = \mathbf{4/255}, 8/255$ |
| **Spatial Attack (SA)** | max rotation: $\mathbf{30°}$, max translation: $\mathbf{10\%}$ of the image size |
| **Square Attack (SQ)** | $\epsilon = \mathbf{4/255}, 8/255$ |
| **Universal Perturbation (UP)** | $\epsilon = \mathbf{4/255}, 8/255, 12/255$ |
| **Auto-PGD (AP)** | $\epsilon = \mathbf{4/255}, 8/255$ |
| **FGSM (FG)** | $\epsilon = \mathbf{4/255}, 8/255$ |
| **Frequency Attack (FA)** | $\text{noise\_strength} = \mathbf{0.05}$ |
| **Gaussian Blur (GB)** | $\sigma = \mathbf{1.0}, \text{blur\_type} = \text{uniform}$ |
| **Pixel Flip (PF)** | $\text{num\_pixel} = \mathbf{5}, \text{attack\_mode} = \text{random}$ |
| **Semantic Rotation (SR)** | $\text{angle} = \mathbf{8}$ |

Table 6: Overview of adversarial attack methods and their parameter settings. Parameter value written with **bold** represents the default value of the corresponding attack among its parameter configurations.

## H.6 EXAMPLES OF ADVERSARIAL IMAGES SUPERVISED SETTING

### H.6.1 GAUSSIAN BLUR ATTACK

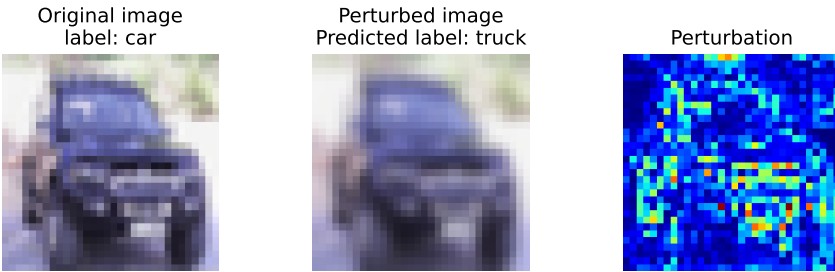

Figure 3: Comparison between Original and Perturbed Images using Gaussian blur attack. Left: Original Image with True Label *car*, Center: Adversarial Image with Predicted Label *truck* and Right: Perturbation

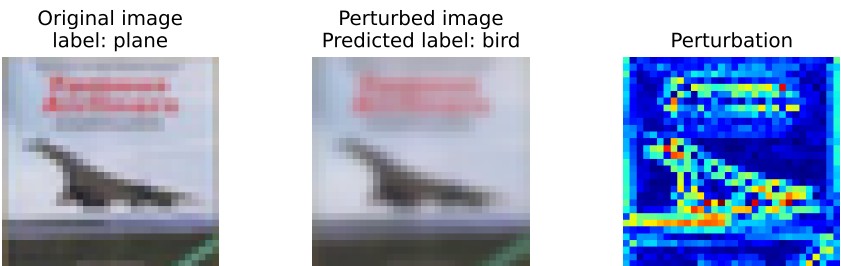

Figure 4: Comparison between Original and Perturbed Images using Gaussian blur attack. Left: Original Image with True Label *plane*, Center: Adversarial Image with Predicted Label *bird* and Right: Perturbation

## H.6.2 PATCH ATTACK

Original image
label: car

Perturbed image
Predicted label: truck

Perturbation

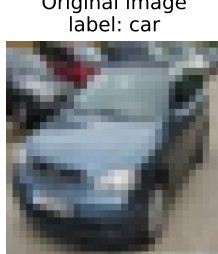 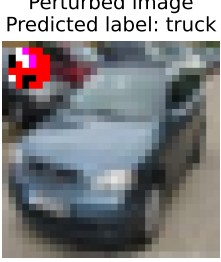 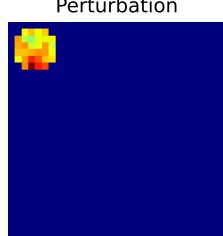

Figure 5: Comparison between Original and Perturbed Images using Patch attack. Left: Original Image with True Label *car*, Center: Adversarial Image with Predicted Label *truck* and Right: Perturbation

Original image
label: plane

Perturbed image
Predicted label: bird

Perturbation

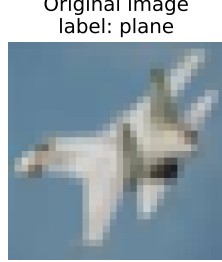 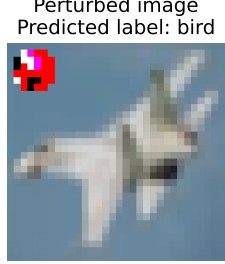 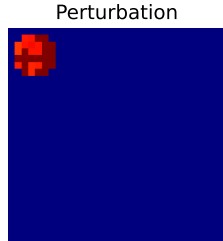

Figure 6: Comparison between Original and Perturbed Images using Patch attack. Left: Original Image with True Label *plane*, Center: Adversarial Image with Predicted Label *bird* and Right: Perturbation

## H.7 EXAMPLES OF ADVERSARIAL IMAGES ZERO SHOT SETTING

### H.7.1 APGD ATTACK

True
Dolphin

Predicted
Camera

Perturbation

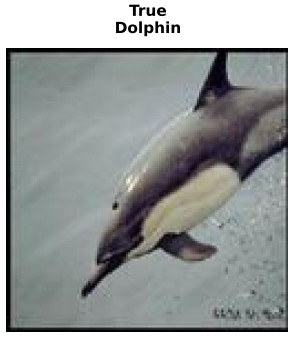 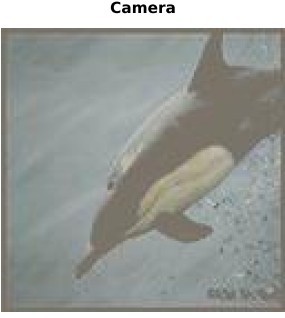 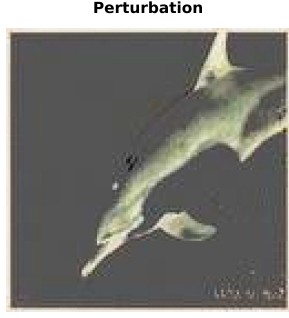

Figure 7: Comparison between Original and Perturbed Images using APGD attack. Left: Original Image with True Label *Dolphin*, Center: Adversarial Image with Predicted Label *Camera* and Right: Perturbation

### H.7.2 PGD Attack

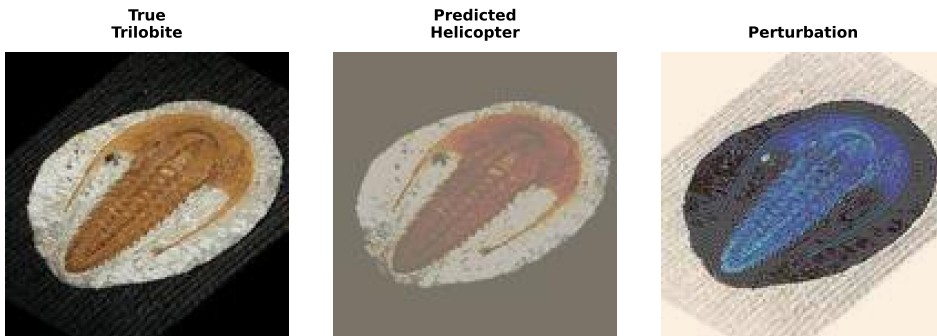

Figure 8: Comparison between Original and Perturbed Images using APGD attack. Left: Original Image with True Label *Trilobite*, Center: Adversarial Image with Predicted Label *Helicopter* and Right: Perturbation

## I  Detection Methods Configuration

This section provides detailed descriptions of the adversarial detection methods evaluated in our experiments, with configurations specified in Table 7. Further, we also mention the hyperparameter settings used in our defense approach.

### I.1  Mahalanobis Detector

The Mahalanobis detector (Lee et al., 2018) leverages the Mahalanobis distance to measure distributional deviations of test samples from training data in the neural network's feature space. For a given sample $\mathbf{x}$ and its feature representation $\mathbf{f}(\mathbf{x})$ at layer $l$, the method computes class-conditional Gaussian distributions $\mathcal{N}(\boldsymbol{\mu}_c^{(l)}, \boldsymbol{\Sigma}^{(l)})$ from clean training data. The Mahalanobis distance is defined as:

$$M_c^{(l)}(\mathbf{x}) = (\mathbf{f}^{(l)}(\mathbf{x}) - \boldsymbol{\mu}_c^{(l)})^T (\boldsymbol{\Sigma}^{(l)})^{-1} (\mathbf{f}^{(l)}(\mathbf{x}) - \boldsymbol{\mu}_c^{(l)})$$

The minimum distance across all classes serves as the confidence score for adversarial detection, exploiting the property that adversarial perturbations typically push samples away from the natural data manifold.

### I.2  Feature Squeezing

Feature Squeezing (Xu et al., 2018) reduces the degrees of freedom available to adversarial perturbations by applying input transformations that compress the feature space. Our implementation uses median smoothing with a $2 \times 2$ kernel and $L_1$ distance metric for comparing predictions. For an input $\mathbf{x}$ and its squeezed version $\mathbf{x}'$, the detection score is computed as:

$$\text{score}(\mathbf{x}) = \|\mathbf{p}(\mathbf{x}) - \mathbf{p}(\mathbf{x}')\|_1$$

where $\mathbf{p}(\cdot)$ represents the model's prediction probabilities. A threshold is determined using the training false positive rate (FPR) of 0.2, assuming legitimate inputs remain robust to minor spatial transformations while adversarial examples exhibit significant prediction changes.

### I.3  MetaDetect

MetaDetect (Ma et al., 2019) formulates adversarial detection as a few-shot learning problem using meta-learning principles. The method employs episodic training with support sets $\mathcal{S} = \{(\mathbf{x}_i, y_i)\}_{i=1}^{N_s}$ and query sets $\mathcal{Q} = \{(\mathbf{x}_j, y_j)\}_{j=1}^{N_q}$, where $y_i \in \{0, 1\}$ indicates clean (0) or adversarial (1) samples. Our configuration uses $N_s = 1$ support example and $N_q = 15$ query examples with a conv3 architecture. The meta-detector learns a function $f_\theta$ that maps from support-query episode pairs to detection decisions, optimizing over episode distributions to generalize across different attack types.

### I.4 MagNet

MagNet (Meng & Chen, 2017) combines detection and defense mechanisms using autoencoder-based reconstruction and probability estimation. The method trains an autoencoder $\mathcal{E}_\phi : \mathbb{R}^d \to \mathbb{R}^d$ on clean data to approximate the natural data manifold. For detection, it computes the reconstruction error:

$$\mathcal{L}_{\text{rec}}(\mathbf{x}) = \|\mathbf{x} - \mathcal{E}_\phi(\mathbf{x})\|_1^2$$

Additionally, MagNet estimates the probability density using the Jensen-Shannon divergence between the original and reconstructed inputs' predicted distributions. The underlying assumption is that adversarial examples, lying off the natural manifold, will exhibit higher reconstruction errors and lower probability estimates compared to legitimate inputs.

| Detection Method | Parameters |
|---|---|
| Mahalanobis Detector | train_fpr = 0.15 |
| Feature Squeezing | distance metric: $L_1$, squeezer: median smoothing ($2\times2$), train_fpr=0.2 |
| MetaDetect | num_support = 1, num_query = 15, arch = conv3 |
| MagNet | $l_1$ norm reconstruction error, train_fpr = 0.15 |

Table 7: Parameters and configurations for different adversarial detection methods. Methods referenced: Mahalanobis (Lee et al., 2018), Feature Squeezing (Xu et al., 2018), MetaDetect (Ma et al., 2019), and MagNet (Meng & Chen, 2017).

### I.5 Hyper Parameter Configuration for the Proposed Method

| Hyperparameter | Candidate Values | Optimal Value | Description |
|---|---|---|---|
| $k_{\text{local}}$ | $\{8, 10, 12, 15\}$ | 8 | Local Gromov-Wasserstein features |
| $k_{\text{global}}$ | $\{3, 5, 7\}$ | 3 | Global Gromov-Wasserstein features |
| $\epsilon_{\text{gw}}$ | $\{0.2, 0.5, 0.8\}$ | 0.5 | Entropic regularization strength |

Table 8: **Hyperparameter search for GW features:** We performed a grid search over the candidate values of each hyperparameter and chose the values that achieved the best trade-off between robustness and model usability. Based on this search, we selected the optimal parameters as $k_{\text{local}} = 8$, $k_{\text{global}} = 3$, and $\epsilon_{\text{gw}} = 0.5$.

| Hyperparameter | Candidate Values | Optimal Value | Description |
|---|---|---|---|
| Kernel | $\{$linear, rbf, poly$\}$ | rbf | Choice of kernel function |
| $C$ | $\{0.1, 1, 10, 100\}$ | 1 | Regularization parameter |
| $\gamma$ | $\{$scale, auto, 0.01, 0.001$\}$ | scale | Kernel Coeff. for RBF |

Table 9: **Hyperparameter search for SVM:** We performed a grid search over the candidate values of each hyperparameter and selected the optimal configuration based on validation accuracy. The chosen parameters are Kernel = rbf, $C = 1$, $\gamma = $ scale.

## J Adaptive Attack Formulation

Evaluating the robustness of a defense mechanism against an adaptive adversary is crucial. We consider an adversary who possesses complete knowledge of the defense's architecture, including the classifier $f_\theta(x)$, the CNN feature extractor $\phi_{\text{cnn}}(x)$, and the crypto feature extractor $\phi_{\text{cr}}^{(b)}(x)$ with its associated transform $T_b$. However, the adversary is unaware of the defender's specific, fixed secret bit vector $b^\star \in \{0, 1\}^D$ used in deployment. This section formalizes the adversary's objective and optimization strategy to generate adversarial examples under this realistic uncertainty, focusing on two distinct consistency-based attacks. This approach is typical for evaluating defenses against strong, adaptive attackers Athalye et al. (2018).

### J.1 ATTACKER'S PRIOR OVER SECRET BITS

To account for the unknown secret $b^\star$, the adversary models it as a random variable $b$ drawn from a prior distribution $p(b)$. This prior is constructed as a mixture model over a set of plausible Bernoulli distributions, $\mathcal{M}$, reflecting the adversary's uncertainty about the specific statistical properties of $b^\star$:

$$p(b) = \frac{1}{|\mathcal{M}|} \sum_{m \in \mathcal{M}} \prod_{j=1}^{D} \text{Bernoulli}(b_j; p_m),$$

where $p_m$ corresponds to the individual Bernoulli success probability for each distribution type in $\mathcal{M}$ (e.g., $p_m = 0.5$ for uniform or Gaussian-threshold components, and specific probabilities like $0.3, 0.7$ for biased Bernoulli components). This mixture prior allows the adversary to account for various possibilities of how the defender might have generated $b^\star$. Modeling unknown parameters in this manner is a standard robust optimization technique Ben-Tal et al. (2009).

### J.2 ATTACKER'S OBJECTIVE FUNCTION

The adversary's goal is to craft an adversarial example $x$ from a benign input $x_0$ that achieves misclassification by $f_\theta(x)$ while simultaneously maintaining a high degree of feature consistency with $x_0$. The latter ensures the adversarial example does not trip the defense's detection mechanisms, particularly those relying on the crypto features. Since the specific $b^\star$ is unknown, the adversary targets an *average* consistency, minimizing the expected penalty under their prior $p(b)$.

The general adversarial objective is:

$$\max_{x \in \mathcal{X}} \mathcal{L}(x; x_0, y) = \ell(f_\theta(x), y) - \lambda \, \mathbb{E}_{b \sim p(b)}[C(x, x_0; b)],$$

where $\ell(f_\theta(x), y)$ is the cross-entropy loss for the true label $y$, which the adversary seeks to maximize; $\lambda > 0$ is a weighting factor that balances the misclassification objective against the consistency penalty — a formulation commonly used in adversarial attacks to trade off attack success and imperceptibility or stealth Carlini & Wagner (2017); $C(x, x_0; b)$ quantifies the discrepancy between features of $x$ and $x_0$ for a given $b$, with lower values of $C$ implying better stealth against consistency checks; and $\mathcal{X}$ defines the allowed perturbation space, typically restricted to a range $[a, b]^{H \times W \times C}$ for pixel values.

### J.3 CONSISTENCY PENALTIES ($C(x, x_0; b)$)

We define the consistency penalty $C(x, x_0; b)$ using an OT-like discrepancy metric, $D_{\text{OT}}(\cdot, \cdot)$. This metric compares feature vectors ($L_2$ distance if dimensions match) or their statistical summaries ($L_2$ distance between mean, std, min, max, skewness, kurtosis if dimensions differ or comparison of statistics is explicitly requested).

For this study, we consider two specific attack formulations based on distinct consistency penalties:

#### J.3.1 CROSS-SPACE CONSISTENCY ATTACK ($C_{\text{cross}}$)

This attack targets the defense by imposing consistency across both the standard CNN feature space and the specialized crypto feature space. The adversary aims to ensure that the features extracted from the adversarial example $x$ remain similar to those from the clean input $x_0$ in both domains. The penalty term is defined as the sum of discrepancies in each feature space:

$$C_{\text{cross}}(x, x_0; b) = D_{\text{OT}}\big(\phi_{\text{cnn}}(x), \phi_{\text{cnn}}(x_0)\big) + D_{\text{OT}}\big(\phi_{\text{cr}}^{(b)}(x), \phi_{\text{cr}}^{(b)}(x_0)\big).$$

By minimizing this penalty, the adversarial example is constrained to modify the input in a way that, on average over $b$, preserves the inherent characteristics captured by both $\phi_{\text{cnn}}$ and $\phi_{\text{cr}}^{(b)}$.

#### J.3.2 MULTI-SCALE CONSISTENCY ATTACK ($C_{\text{ms}}$)

This attack extends the cross-space consistency by introducing an additional constraint on global CNN feature similarity. This reflects a defense that might perform multi-scale or global consistency checks specifically on CNN features. The $C_{\text{ms}}$ penalty is structured as:

$$C_{\text{ms}}(x, x_0; b) = C_{\text{local}}(x, x_0; b) + C_{\text{global}}(x, x_0),$$

where:

- $C_{\text{local}}(x, x_0; b) = C_{\text{cross}}(x, x_0; b)$ represents the local, cross-space consistency across CNN and crypto features.

- $C_{\text{global}}(x, x_0) = D_{\text{OT}}\big(\phi_{\text{cnn}}(x), \phi_{\text{cnn}}(x_0)\big)$ ensures global consistency focusing solely on CNN features. Note that $C_{\text{global}}$ does not depend on $b$, as the global consistency check is assumed to be deterministic based on CNN features, which are not secrets-dependent.

The combined penalty

$$C_{\text{ms}}(x, x_0; b) = 2 \cdot D_{\text{OT}}\big(\phi_{\text{cnn}}(x), \phi_{\text{cnn}}(x_0)\big) + D_{\text{OT}}\big(\phi_{\text{cr}}^{(b)}(x), \phi_{\text{cr}}^{(b)}(x_0)\big)$$

effectively doubles the weight on CNN feature consistency, making the adversarial example potentially harder to detect by defenses performing aggregated checks on CNN features.

To optimize the objective function, the adversary utilizes an iterative PGD Madry et al. (2018) is used. Since the objective involves an expectation over the unknown $b$, a Monte Carlo (MC) approximation is employed Rubinstein & Kroese (2016).

## K  IMPLEMENTATION DETAILS

### K.1  PSEUDOCODE FOR CROSS-SPACE DETECTOR

---
**Algorithm 1** Multi-Scale Cross-Space GW Detector

---
**Require:** Image $x$; $z \leftarrow h_\theta(x)$; $p \leftarrow T_k^{(b)}(x)$
1: **for all** $s \in \{\text{lo, gl}\}$ **do**
2:     Build $\mathcal{N}_s^Z(z), \mathcal{N}_s^P(p)$; compute $\mu_s^Z, \mu_s^P, \psi_s^Z, \psi_s^P$
3:     $g_1 \leftarrow \text{GW}_\lambda^2(\mu_s^Z, \mu_s^P)$; $g_2 \leftarrow \text{GW}_\lambda^2(\psi_s^Z, \psi_s^P)$
4:     $h \leftarrow \text{ENTROPY}(\psi_s^Z, \psi_s^P)$; $\mathbf{f} \leftarrow \mathbf{f} \| [g_1, g_2, h]$
5: **end for**
6: **return** $\text{SVM}(\mathbf{f}) \in \{\text{clean, adv}\}$ =0

---

We provide the hyperparmeter selection details for GW features in Table 8 and for SVM classifier in Table 9.

## L  ADDITIONAL EXPERIMENTS

### L.1  ADDITIONAL BASELINE DEFENCES AND ATTACKS

To evaluate performance against imperceptible and optimization-free attacks, we extend our experiments to three recent low-magnitude adversarial methods, i.e., AdvAD, PGN, and BSR, which explicitly target the small-perturbation regime. These attacks probe the limits of pixel-level stability and serve as stringent tests for detectors relying on fine-grained geometric discrepancies.

In addition to our default ResNet-18 backbone, we generate all attacks using a Vision Transformer (ViT) to assess robustness across fundamentally different architectural families. Table 10 reports binary detection accuracy for all attacks and defenses. Table 11 reports corresponding AUROC values.

Across all attack types—including classical gradient-based attacks (PGD, SQ, PT) and modern diffusion-based or non-parametric attacks (AdvAD, PGN, BSR)—our detector achieves the strongest performance on both ResNet-18 and ViT. Competing baselines degrade substantially under stronger or low-magnitude attacks, whereas our Z–P discrepancy remains highly separable across architectures, perturbation magnitudes, and attack mechanisms.

| Attack | Model | Ours | MD | FS | MAD | MN | EA | BY |
|---|---|---|---|---|---|---|---|---|
| PGD | ResNet-18 | **97.8** | 91.7 | 74.4 | 46.3 | 81.9 | 97.5 | 69.5 |
| | ViT | **95.7** | 49.8 | 70.3 | 57.5 | 49.4 | 93.2 | 77.5 |
| SQ | ResNet-18 | **97.6** | 89.1 | 88.5 | 45.8 | 91.9 | 91.9 | 59.5 |
| | ViT | **96.2** | 51.0 | 87.3 | 56.3 | 49.7 | 90.5 | 62.5 |
| PT | ResNet-18 | **98.0** | 86.4 | 67.3 | 46.5 | 50.1 | 90.1 | 78.0 |
| | ViT | **95.4** | 51.3 | 71.2 | 54.9 | 49.3 | 89.5 | 78.5 |
| AAD | ResNet-18 | **96.4** | 54.4 | 41.3 | 49.6 | 53.2 | 94.6 | 61.0 |
| | ViT | **93.7** | 52.6 | 52.7 | 53.6 | 50.1 | 92.7 | 80.0 |
| PGN | ResNet-18 | **96.5** | 70.8 | 61.0 | 45.6 | 49.6 | 94.9 | 75.0 |
| | ViT | **96.9** | 52.2 | 65.8 | 53.7 | 49.9 | 94.9 | 75.5 |
| BSR | ResNet-18 | **95.0** | 73.0 | 42.4 | 45.5 | 62.1 | 92.9 | 78.5 |
| | ViT | **98.7** | 51.9 | 53.9 | 53.7 | 50.0 | 93.2 | 76.0 |

Table 10: Detection accuracy (%) on CIFAR-10 for a range of classical and low-magnitude attacks, evaluated using ResNet-18 and ViT. Best results are in bold; second best are underlined.

| Attack | Model | Ours | MD | FS | MAD | MN | EA | BY |
|---|---|---|---|---|---|---|---|---|
| PGD | ResNet-18 | **0.99** | 0.81 | 0.69 | 0.49 | 0.51 | 0.99 | 0.72 |
| | ViT | **0.99** | 0.72 | 0.72 | 0.57 | 0.49 | 0.96 | 0.89 |
| SQ | ResNet-18 | **0.99** | 0.55 | 0.87 | 0.48 | 0.49 | 0.95 | 0.54 |
| | ViT | **0.99** | 0.40 | 0.88 | 0.56 | 0.50 | 0.95 | 0.60 |
| PT | ResNet-18 | **0.99** | 0.93 | 0.52 | 0.50 | 0.54 | 0.95 | 0.89 |
| | ViT | **0.99** | 0.47 | 0.63 | 0.54 | 0.52 | 0.94 | 0.90 |
| AAD | ResNet-18 | **0.99** | 0.61 | 0.56 | 0.49 | 0.51 | 0.97 | 0.55 |
| | ViT | 0.98 | 0.88 | 0.55 | 0.53 | 0.53 | **0.97** | 0.94 |
| PGN | ResNet-18 | **0.99** | 0.79 | 0.59 | 0.48 | 0.50 | 0.98 | 0.80 |
| | ViT | **0.97** | 0.54 | 0.61 | 0.55 | 0.51 | 0.98 | 0.81 |
| BSR | ResNet-18 | **0.99** | 0.64 | 0.53 | 0.48 | 0.51 | 0.95 | 0.87 |
| | ViT | **0.99** | 0.59 | 0.58 | 0.55 | 0.54 | 0.96 | 0.84 |

Table 11: AUROC comparison on CIFAR-10 for various attacks generated using ResNet-18 and ViT. Best results are in bold; second best are underlined.

## L.2 EXPERIMENTAL EVALUATIONS WITH ADDITIONAL METRICS

| Attacks | Model | Ours | MD | FS | MAD | MN | EA | BY |
|---|---|---|---|---|---|---|---|---|
| PGD | ResNet-18 | 0.97/0.98 | 0.89/0.95 | 0.71/0.72 | 0.17/0.54 | 0.97/0.65 | **0.98/0.95** | 0.67/0.74 |
| | ViT | **1.00/0.90** | 0.49/0.95 | 0.73/0.78 | 0.57/0.57 | 0.25/0.01 | 0.98/0.93 | 0.72/0.90 |
| SQ | ResNet-18 | **0.96/0.97** | 0.88/0.89 | 0.83/0.88 | 0.16/0.52 | 0.22/0.01 | 0.89/0.92 | 0.60/0.54 |
| | ViT | **0.95/0.94** | 0.50/0.98 | 0.93/0.80 | 0.56/0.57 | 0.40/0.01 | 0.88/0.90 | 0.63/0.60 |
| PT | ResNet-18 | **0.97/0.98** | 0.87/0.84 | 0.62/0.49 | 0.17/0.57 | 0.55/0.01 | 0.91/0.88 | 0.72/0.91 |
| | ViT | **0.99/0.93** | 0.50/0.98 | 0.75/0.61 | 0.53/0.56 | 0.25/0.01 | 0.89/0.87 | 0.72/0.92 |
| AAD | ResNet-18 | **0.98/0.93** | 0.63/0.21 | 0.55/0.41 | 0.49/0.54 | 0.87/0.07 | 0.89/0.93 | 0.61/0.57 |
| | ViT | **0.93/0.92** | 0.51/1.00 | 0.55/0.61 | 0.82/0.53 | 0.41/0.01 | 0.89/0.95 | 0.73/0.95 |
| PGN | ResNet-18 | **0.97/0.93** | 0.83/0.51 | 0.67/0.59 | 0.15/0.51 | 0.34/0.01 | 0.90/0.91 | 0.70/0.85 |
| | ViT | **0.98/0.95** | 0.51/0.99 | 0.67/0.58 | 0.82/0.53 | 0.44/0.01 | 0.92/0.94 | 0.71/0.86 |
| BSR | ResNet-18 | **0.99/0.91** | 0.84/0.55 | 0.45/0.62 | 0.15/0.51 | 0.94/0.25 | 0.96/0.88 | 0.72/0.92 |
| | ViT | **0.99/0.98** | 0.50/0.98 | 0.51/0.64 | 0.82/0.53 | 0.42/0.01 | 0.93/0.91 | 0.71/0.87 |

Table 12: Precision / Recall comparison of adversarial detection on CIFAR-10 under various attacks generated using ResNet-18 and ViT, with best results in bold and second best underlined.

Table 12 reports the *precision* and *recall* of different adversarial detection methods across six attack types using both ResNet-18 and ViT. Across nearly all attacks and architectures, **our method achieves the highest precision and recall values** (highlighted in bold), often reaching **near-perfect scores close to** 1.00, demonstrating its strong ability to accurately detect adversarial inputs. In contrast, EA generally appears as the second-best performer, but still lags behind our approach, particularly on challenging attacks such as AAD, PGN, and BSR. Overall, these results show that our

detector is **highly robust and consistent**, maintaining **superior numerical performance across all attack families and both model architectures**.

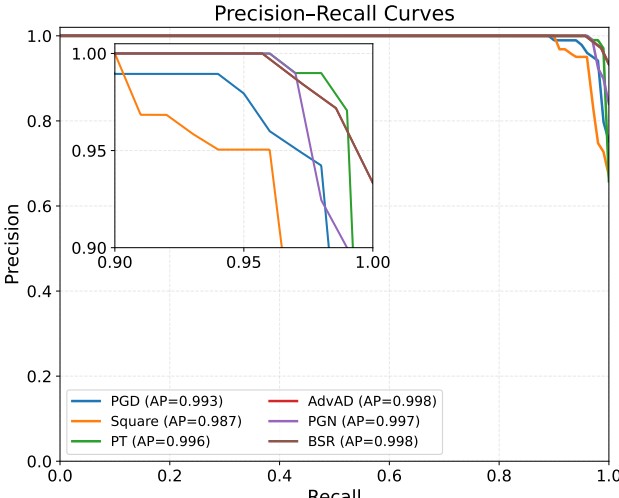

Figure 9: PR curves for our method on attacks generated via ResNet-18 model on CIFAR10 dataset. The small inset box displays a zoomed-in view of the upper-right corner of the PR curves.

In Figure 9, the Precision–Recall (PR) curves show that our detector maintains consistently high precision and recall across all six attack types. All curves remain close to the top-right region, indicating strong overall detection performance. The zoomed inset further illustrates that even at high recall values (close to 1.0), the precision for each attack remains above 0.95 with only minimal degradation. The high Average Precision (AP) scores (0.987–0.998) confirm the robustness of our method, demonstrating reliable performance even against stronger modern attacks such as AdvAD, PGN, and BSR. Overall, the PR curves highlight that our approach is highly accurate, stable, and generalizes well across diverse adversarial attacks.

| | **Ours** | MD | FS | MAD | MN |
|---|---|---|---|---|---|
| AA | **94.9** | 67.2 | 82.0 | 48.3 | 71.6 |
| CW | **93.8** | 72.1 | 85.3 | 48.9 | 54.2 |
| PT | **94.4** | 84.9 | 67.1 | 49.5 | 54.8 |
| PGD | **94.8** | 90.0 | 73.7 | 49.0 | 78.5 |
| SA | **93.4** | 76.7 | 73.7 | 39.2 | 52.5 |
| SQ | **93.4** | 87.7 | 87.8 | 48.5 | 41.9 |
| UP | **94.4** | 65.0 | 52.9 | 49.0 | 45.4 |
| AP | **93.8** | 66.9 | 80.7 | 48.6 | 71.1 |
| FG | **94.3** | 72.3 | 60.2 | 49.1 | 42.2 |
| FA | **94.6** | 48.5 | 49.3 | 48.7 | 47.3 |
| GB | **73.2** | 48.5 | 50.9 | 48.1 | 46.1 |
| PF | **93.3** | 50.3 | 50.7 | 48.5 | 46.9 |
| SR | **93.1** | 48.9 | 52.0 | 48.9 | 46.8 |

Table 13: End-to-end accuracy (%). Best results are in bold and second best are underlined.

Table 13 shows that our method consistently achieves the highest end-to-end detection accuracy across all evaluated attacks. For every attack type, our detector outperforms all competing defenses, often by margins of 10–40%. FS and MD occasionally achieve the second-best performance, but they remain substantially weaker overall, while MAD and MN lag far behind on most attacks. These results demonstrate that our approach generalizes robustly across a broad spectrum of adversarial perturbations and maintains reliable detection performance even against diverse and challenging attack strategies.

| Attacks | Model | Ours | MD | FS | MAD | MN | EA | BY |
|---------|-------|------|------|------|-------|-------|------|----|
| PGD | ResNet-18 | **95.6** | 50.00 | 68.9 | 45.8 | 91.00 | 94.9 | – |
| SQ | ResNet-18 | **96.7** | 50.00 | 84.2 | 45.5 | 49.00 | 89.6 | – |
| PT | ResNet-18 | **96.2** | 50.00 | 73.7 | 45.0 | 50.50 | 88.2 | – |
| AAD | ResNet-18 | **97.6** | 50.00 | 55.9 | 49.6 | 59.38 | 95.4 | – |
| PGN | ResNet-18 | **96.3** | 50.00 | 63.2 | 46.3 | 49.06 | 93.2 | – |
| BSR | ResNet-18 | **97.8** | 50.00 | 59.4 | 46.35 | 50.31 | 93.6 | – |

Table 14: Detection accuracy (%) comparison on ImageNet + ResNet-18 with best (bold) and second best (underlined).

| Attacks | Model | Ours | MD | FS | MAD | MN | EA | BY |
|---------|-------|------|------|------|-------|------|------|----|
| PGD | ResNet-18 | **0.98** | 0.53 | 0.73 | 0.503 | 0.68 | **0.98** | – |
| SQ | ResNet-18 | **0.97** | 0.55 | 0.79 | 0.489 | 0.49 | 0.94 | – |
| PT | ResNet-18 | **0.99** | 0.52 | 0.71 | 0.45 | 0.50 | 0.95 | – |
| AAD | ResNet-18 | **0.99** | 0.49 | 0.55 | 0.495 | 0.45 | 0.96 | – |
| PGN | ResNet-18 | **0.98** | 0.50 | 0.66 | 0.488 | 0.44 | 0.95 | – |
| BSR | ResNet-18 | **0.99** | 0.50 | 0.62 | 0.488 | 0.43 | 0.95 | – |

Table 15: AUC comparison on ImageNet with best (bold) and second best (underlined).

### L.3 EVALUATION ON THE IMAGENET DATASET ON DEFAULT RESNET-18 MODEL

In this section, we additionally evaluate our method on the large-scale ImageNet Deng et al. (2009) dataset using the default ResNet-18 backbone. This allows us to verify that the detector remains effective when applied to high-resolution images and a significantly more challenging data distribution.

*Note:* The BY He et al. (2022) baseline does not provide a publicly available codebase, so we reproduce the method following the details reported in the original paper. Running BY requires a self-supervised model trained on the target dataset; while we trained such a model for CIFAR-10, training an equivalent backbone for ImageNet is computationally prohibitive. Consequently, we omit BY from the ImageNet experiments due to the intractable training cost and scale of the dataset.

Tables 14, 15, and 16 jointly show that our detector achieves the strongest overall performance on ImageNet across all six adversarial attacks. In terms of detection accuracy and AUC, our method consistently reaches top performance—typically in the 0.98–0.99 range—while the next-best baseline, EA, trails by $1-5\%$ depending on the attack. The precision and recall results further reinforce this trend: our detector attains the best or second-best PR scores in nearly all settings, maintaining high recall even for challenging attacks such as AdvAD, PGN, and BSR. Competing defenses such as MD, FS, MAD, and MN perform substantially worse across all metrics. Together, these results confirm that our approach scales robustly to high-resolution ImageNet data and preserves strong discriminative ability under a wide range of adversarial perturbations.

### L.4 SENSITIVITY TO THE PRIME-RESOLUTION PARAMETER $k$

Figure 10 shows how detection accuracy varies as the prime-resolution parameter $k$ is changed. Across the entire tested range ($k = 3$ to $k = 8$), the mean detection accuracy remains extremely stable, fluctuating only within a narrow interval of approximately 0.954–0.957. The shaded region captures variability across repeated runs and likewise remains tightly concentrated.

These results indicate that the detector is not sensitive to the specific choice of $k$. Even when $k$ is varied over a relatively broad range, the performance remains effectively unchanged. This

| Attacks | Model | Ours | MD | FS | MAD | MN | EA | BY |
|---------|-------|------|------|------|------|------|------|----|
| PGD | ResNet-18 | **1/0.93** | 0.50/1.00 | 0.65/0.82 | 0.05/0.55 | 0.92/1.00 | 0.98/0.97 | – |
| SQ | ResNet-18 | **0.98/0.96** | 0.50/1.00 | 0.84/0.76 | 0.05/0.52 | 0.42/0.06 | 0.89/0.91 | – |
| PT | ResNet-18 | **0.96/0.95** | 0.50/1.00 | 0.82/0.65 | 0.04/0.45 | 0.52/0.09 | 0.87/0.89 | – |
| AAD | ResNet-18 | **0.99/0.96** | 0.50/1.00 | 0.51/0.55 | 0.49/0.54 | 0.87/0.21 | 0.93/0.96 | – |
| PGN | ResNet-18 | 0.94/0.97 | 0.50/1.00 | 0.61/0.68 | 0.16/0.52 | 0.28/0.01 | **0.95/0.92** | – |
| BSR | ResNet-18 | **0.97/0.96** | 0.50/1.00 | 0.50/0.73 | 0.16/0.52 | 0.54/0.03 | 0.96/0.91 | – |

Table 16: Precision / Recall comparison on ImageNet under various attacks, with best values in bold and second best underlined.

| Attacks | Model | Ours | MD | FS | MAD | MN | EA | BY |
|---------|-------|------|------|------|------|------|------|------|
| **PGD** | ResNet-18 | 0.96 | 0.66 | 0.72 | 0.10 | 0.96 | **0.97** | - |
| **SQ** | ResNet-18 | **0.96** | 0.66 | 0.79 | 0.09 | 0.10 | 0.90 | - |
| **PT** | ResNet-18 | **0.95** | 0.66 | 0.72 | 0.08 | 0.15 | 0.88 | - |
| **AAD** | ResNet-18 | **0.97** | 0.66 | 0.52 | 0.51 | 0.35 | 0.95 | - |
| **PGN** | ResNet-18 | **0.95** | 0.66 | 0.64 | 0.24 | 0.02 | 0.93 | - |
| **BSR** | ResNet-18 | **0.96** | 0.66 | 0.59 | 0.24 | 0.07 | 0.93 | - |

Table 17: F1 score comparison on ImageNet under various adversarial attacks. Best results are shown in bold, second best underlined.

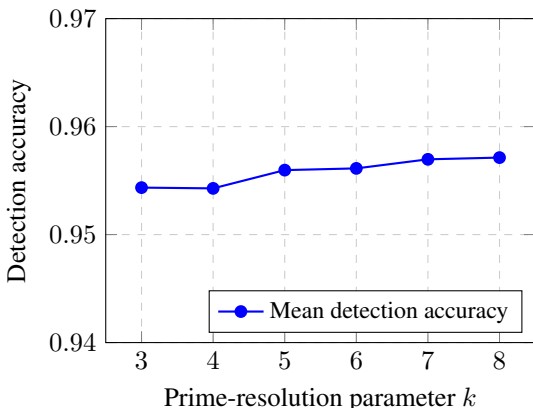

Figure 10: Sensitivity of the detector to the prime-resolution parameter $k$.

robustness reflects an intrinsic property of our method, i.e., once the prime gaps are sufficiently fine to induce stable absorption and injectivity behavior, further refinement provides little additional benefit. In practice, this means that $k$ does not require fine-tuning to achieve strong performance, simplifying deployment across different datasets and architectures.

### L.5 COMPARISON WITH ALTERNATIVE QUANTIZATION SCHEMES

To assess whether the advantages of our quantization scheme can be replicated by other non-uniform discretization strategies, we compare it against three alternatives: *Fibonacci quantization*, *logarithmic quantization*, and standard *uniform quantization*. All approaches are evaluated under the same experimental setup and across five random seeds $(1, 3, 5, 21, 42)$. Table 18 reports detection accuracy for each seed, together with the mean and standard deviation.

| Quantization | Seed | | | | | Mean $\pm$ Std |
|--------------|------|------|------|------|------|----------------|
| | 1 | 3 | 5 | 21 | 42 | |
| **Ours** | **0.94** | **0.92** | **0.95** | **0.94** | **0.96** | **0.94 $\pm$ 0.01** |
| Fibonacci | 0.62 | 0.61 | 0.55 | 0.60 | 0.68 | 0.61 $\pm$ 0.04 |
| Logarithmic | 0.70 | 0.61 | 0.61 | 0.56 | 0.58 | 0.62 $\pm$ 0.05 |
| Uniform | 0.65 | 0.65 | 0.60 | 0.56 | 0.65 | 0.62 $\pm$ 0.03 |

Table 18: Detection accuracy of different quantization schemes across five random seeds. Values reported using a ResNet backbone and CIFAR-10 under mixed attacks (PGD, FGSM, APGD).

Across all seeds, our method achieves the highest mean accuracy and the lowest variance, demonstrating both improved performance and greater stability compared to all alternative discretization schemes. Notably, replacing our scheme with Fibonacci or logarithmic quantization does *not* yield comparable results, despite also introducing non-uniform discretization. Uniform quantization performs similarly poorly. These results suggest that the structural properties of our secret prime quantization strategy are essential for producing a reliable Z–P discrepancy signal, and that generic discretization methods do not replicate this behavior in practice.

## L.6 SENSITIVITY ANALYSIS OF $k_{\text{LOCAL}}$ AND $k_{\text{GLOBAL}}$

| $k_{\text{local}}$ | Accuracy | $k_{\text{global}}$ | Accuracy |
|---|---|---|---|
| 8 | **97.90** | 3 | **97.90** |
| 10 | 86.15 | 5 | 95.60 |
| 12 | 92.57 | 7 | 83.45 |
| 15 | 93.24 | 10 | 85.35 |

Table 19: Detection accuracy as a function of the local neighborhood size $k_{\text{local}}$ (left) and the global neighborhood size $k_{\text{global}}$ (right).

We evaluate the sensitivity of our detector to the neighborhood parameters $k_{\text{local}}$ and $k_{\text{global}}$, which govern the local consistency scale and the global support for GW coupling, respectively.

For the local neighborhood size, we test $k_{\text{local}} \in \{8, 10, 12, 15\}$. Accuracy remains consistently high across all settings, with the best performance (**97.9%**) obtained at $k_{\text{local}} = 8$. While performance dips for $k_{\text{local}} = 10$, it recovers for larger values (12 and 15), indicating that the method is broadly robust to the choice of local scale.

For the global neighborhood size, we test $k_{\text{global}} \in \{3, 5, 7, 10\}$. Smaller global neighborhoods yield the best performance, with $k_{\text{global}} = 3$ achieving **97.9%**. Larger values gradually degrade performance, suggesting that excessively large global supports may introduce noise or dilute the structural alignment captured during cross-space GW coupling.

Overall, the detector displays stable performance across a wide range of neighborhood sizes, with optimal performance achieved at smaller values. Based on this analysis, we adopt $k_{\text{local}} = 8$ and $k_{\text{global}} = 3$ for all main experiments.

## L.7 ROBUSTNESS AGAINST COMMON CORRUPTIONS

To assess whether the detector responds specifically to adversarial perturbations, rather than generic input noise, we evaluate its behaviour under benign corruptions using the CIFAR-C benchmark Hendrycks & Dietterich (2019a). CIFAR-C includes a diverse set of naturally occurring degradation types, such as Gaussian, shot, and impulse noise; blur corruptions (defocus, frosted glass, motion, zoom); weather effects (snow, frost, fog); brightness and contrast shifts; elastic distortions; pixelation; and JPEG compression.

Table 20 reports results for the *Gaussian noise* corruption. The detector maintains high accuracy and AUROC, indicating that benign perturbations do not trigger the characteristic Z–P discrepancies associated with adversarial attacks. This supports our claim that the method does not misclassify natural corruptions as adversarial.

| Corruption Type | Detection Accuracy | AUROC | Precision / Recall / F1 |
|---|---|---|---|
| Gaussian Noise | 96.43% | 0.99 | 0.97 / 0.96 / 0.96 |

Table 20: Performance of our detector under Gaussian noise (CIFAR-C).

## L.8 RUNTIME AND MEMORY EFFICIENCY

Table 21 compares inference-time and memory footprint across several recent adversarial detection baselines. Our detector achieves a competitive runtime of 0.12 seconds per sample while maintaining a moderate CPU and GPU memory footprint. Methods such as BY He et al. (2022) require substantially larger memory because they process multiple transformed copies of each input, resulting in expanded intermediate activations. In contrast, our detector relies on a single forward pass through a ResNet backbone, leading to a modest GPU footprint.

Lightweight approaches such as FS Xu et al. (2018) and MN Meng & Chen (2017) offer faster runtimes but exhibit either higher memory usage or weaker robustness. Overall, our method strikes an effective balance among runtime efficiency, memory consumption, and detection performance.

| Method | Runtime (s) | CPU (GB) | GPU (GB) |
|--------|-------------|----------|----------|
| BY | 0.152 | 2.00 | 0.60 |
| EA | 0.115 | **0.20** | 11.80 |
| MD | 0.141 | 0.68 | 0.10 |
| FS | **0.005** | 0.25 | 0.17 |
| MAD | 0.105 | 3.30 | 0.00 |
| MN | 0.012 | 0.78 | **0.10** |
| **Ours** | 0.120 | 0.62 | 2.04 |

Table 21: Runtime and memory usage of different adversarial detection methods on CIFAR-10. Best values in bold, second-best underlined.

## L.9 ROBUSTNESS AND GENERALIZATION CAPABILITIES

### L.9.1 RELIABILITY ANALYSIS

We study detector reliability across a range of perturbation levels using TPR/FPR heatmaps (Fig. 11). With dataset and backbone fixed, only adversarial conditions vary, revealing how consistently detectors identify adversarial inputs while avoiding false positives on clean samples.

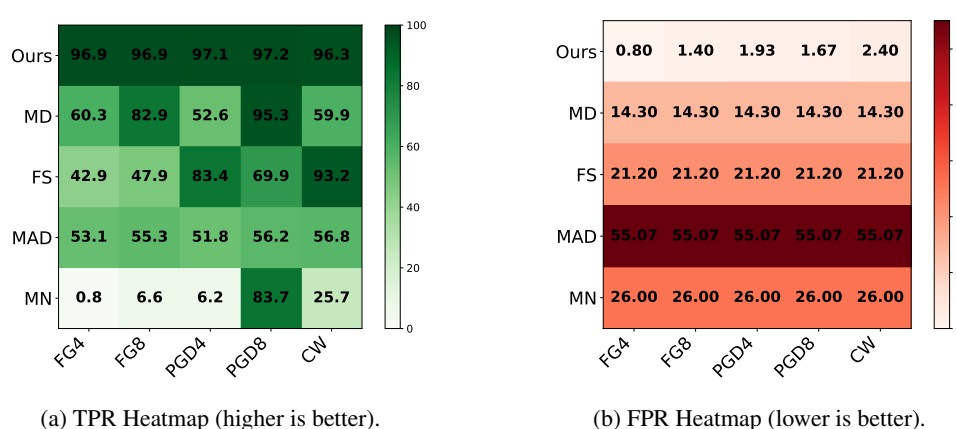

(a) TPR Heatmap (higher is better).          (b) FPR Heatmap (lower is better).

Figure 11: **Reliability via TPR/FPR heatmaps.** X-axis: attack types with perturbation levels, Y-axis: defenses. Thresholds are calibrated only on clean samples, so baselines yield constant FPR across attacks.

### L.9.2 CROSS-ATTACK GENERALIZATION

To evaluate generalization to unseen attacks, we train on ResNet18 adversarial samples from FG and PGD, then test on CW, SQ, SA, and PT. Results (Table 22) report accuracy, adversarial recall, and adversarial precision. Our method consistently outperforms baselines, demonstrating robustness to unseen attack families.

| Attack | Ours | MD | FS | MAD | MN |
|--------|------|-----|-----|------|-----|
| CW | **97.5/100/95** | 71/51/86 | 90/90/90 | 49.5/69/46.6 | 60/45/63 |
| SQ | 93.5/98/89 | 88/85/91 | **94.5/99/90.8** | 48.5/63/47 | 45/15/46 |
| SP | **95.5/99/92** | 86/80/90 | 92.5/95/90.5 | 49/67/49 | 48/21/46 |
| PT | **95/100/90** | 91/90/91 | 89.5/89/89.9 | 49/67/49 | 45/15/37 |

Table 22: **Cross-Attack Generalization.** Accuracy / Recall$_{adv}$ / Precision$_{adv}$ (%).

### L.9.3 Cross-Model Generalization

We also test transfer robustness by training on ResNet18 attacks and evaluating on FourierNet (FNet) adversarial samples. Table 23 reports detection accuracy, recall, and precision. Our method again outperforms baselines, showing resilience to model transfer attacks.

| Attack | Ours | MD | FS | MAD | MN |
|---|---|---|---|---|---|
| CW | **95/97/94** | 54/17/68 | 73.5/91/67.4 | 50/69/50 | 50/25/52 |
| SQ | **87/93/83** | 51/10/55 | 77.0/98/69.0 | 49/67/49 | 52/29/54 |
| SP | **94/93/95** | 64/37/82 | 74/92/67.6 | 49/67/49 | 53/31/56 |
| PT | **96/95/97** | 57/22/73 | 72.5/89/66.9 | 47/64/48 | 49/22/47 |
| FG | **93/99/89** | 55/18/69 | 70.0/84/65.6 | 46/61/47 | 50/25/50 |
| PGD | **98/99/99** | 56/21/72 | 74.5/93/67.9 | 50/69/50 | 50/26/52 |

Table 23: **Cross-Model Generalization.** Accuracy / Recall$_{adv}$ / Precision$_{adv}$ (%).

### L.10 True Positive Rate (TPR) Analysis

Table 24 reports the *true positive rate* (TPR) across adversarial attack types, i.e., the fraction of adversarial inputs correctly detected as adversarial. The formula is, TPR $= \frac{\text{detected adversarial}}{\text{all adversarial}}$. Bold entries denote the best-performing method, and underlined entries denote the second best. Our method consistently achieves the highest TPR in 12 out of 13 attacks, showing large margins especially for transfer-based (UP) and perceptual/frequency attacks (FA, PF, SR). The only exception is Gaussian blur (GB), where all detectors struggle, but our method still provides a clear advantage over baselines. These results highlight that our cross-space framework is particularly effective in reliably flagging adversarial samples, even under challenging attack families.

| Attack / Method | Ours | MD | FS | MAD | MN |
|---|---|---|---|---|---|
| AA | **97.40 ± 1.07** | 50.60 ± 1.99 | 86.50 ± 1.24 | 57.96 ± 3.10 | 74.09 ± 1.26 |
| CW | **96.33 ± 0.57** | 59.90 ± 0.65 | 93.20 ± 0.14 | 56.82 ± 1.52 | 39.30 ± 1.60 |
| PT | **97.73 ± 0.34** | 85.50 ± 0.85 | 56.70 ± 2.49 | 55.33 ± 3.92 | 40.50 ± 1.66 |
| PGD | **97.20 ± 1.07** | 95.30 ± 0.62 | 69.90 ± 2.68 | 56.22 ± 1.18 | 87.90 ± 0.29 |
| SA | **96.67 ± 0.23** | 69.00 ± 1.91 | 69.90 ± 0.17 | 36.20 ± 0.62 | 35.80 ± 2.86 |
| SQ | 96.67 ± 0.90 | 91.20 ± 0.77 | **98.20 ± 1.12** | 55.90 ± 3.34 | 14.80 ± 1.12 |
| UP | **97.00 ± 1.14** | 45.60 ± 3.06 | 28.50 ± 2.52 | 55.80 ± 2.12 | 21.70 ± 1.17 |
| AP | **96.22 ± 1.11** | 49.30 ± 1.65 | 83.90 ± 1.44 | 54.28 ± 1.59 | 73.10 ± 1.51 |
| FG | **96.87 ± 1.57** | 60.30 ± 2.64 | 42.90 ± 3.26 | 53.09 ± 0.72 | 15.30 ± 1.57 |
| FA | **94.33 ± 1.23** | 12.50 ± 0.88 | 21.10 ± 1.23 | 53.34 ± 3.08 | 25.50 ± 1.78 |
| GB | **73.00 ± 10.27** | 12.40 ± 1.92 | 24.50 ± 0.53 | 50.42 ± 2.38 | 23.10 ± 2.06 |
| PF | **96.33 ± 0.57** | 16.10 ± 1.20 | 24.00 ± 0.46 | 53.01 ± 3.65 | 24.70 ± 1.21 |
| SR | **95.33 ± 1.07** | 13.30 ± 0.19 | 26.60 ± 1.24 | 54.20 ± 2.37 | 24.40 ± 2.29 |

Table 24: **True positive rate (%) on adversarial samples:** This table shows the results of TPR measured on adversarial samples.

### L.11 Experiments on FMNIST and KMNIST

Table 25 and Table 26 show the result of the detection accuracy on FMNIST and KMNIST respectively. Regarding the performance of MD on FMNIST and KMNIST, we observed that the distributions of Mahalanobis scores for clean and adversarial samples did not show significant difference. This is because clean and adversarial features for FMNIST and KMNIST are fairly similar in the feature space of ResNet18, making it difficult for MD to distinguish between clean and adversarial samples. The similar feature representations in these datasets limit the separability of the two distributions of Mahalanobis scores. A similar phenomenon regarding the distributions of clean and adversarial features used for detection was observed in MN.

| Attack | Ours | MD | MN |
|--------|------|-----|-----|
| AA4 | $\mathbf{97.60 \pm 0.22}$ | $50.00 \pm 0.56$ | $49.90 \pm 1.59$ |
| AA8 | $\mathbf{96.43 \pm 0.59}$ | $\underline{50.40 \pm 0.86}$ | $50.10 \pm 1.75$ |
| CW | $\mathbf{96.10 \pm 1.63}$ | $\underline{51.60 \pm 0.67}$ | $54.05 \pm 1.43$ |
| PT7 | $\mathbf{98.63 \pm 0.17}$ | $89.45 \pm 0.45$ | $\underline{90.20 \pm 0.32}$ |
| PT14 | $\mathbf{94.10 \pm 0.49}$ | $\underline{90.20 \pm 0.43}$ | $90.09 \pm 0.32$ |
| PGD | $\mathbf{97.63 \pm 0.53}$ | $\underline{49.80 \pm 0.67}$ | $49.90 \pm 1.80$ |
| PGD8 | $\mathbf{95.33 \pm 0.97}$ | $\underline{51.45 \pm 0.69}$ | $50.30 \pm 1.82$ |
| SA | $\mathbf{97.21 \pm 1.36}$ | $\underline{80.80 \pm 2.01}$ | $53.10 \pm 0.15$ |
| SQ | $\mathbf{97.20 \pm 0.57}$ | $\underline{50.05 \pm 0.55}$ | $50.00 \pm 0.18$ |
| SQ8 | $\mathbf{96.47 \pm 4.01}$ | $\underline{52.20 \pm 0.67}$ | $50.73 \pm 0.52$ |
| UP | $\mathbf{99.87 \pm 0.05}$ | $50.05 \pm 0.71$ | $\underline{50.23 \pm 1.85}$ |
| UP8 | $\mathbf{98.30 \pm 1.27}$ | $51.15 \pm 0.55$ | $\underline{50.65 \pm 1.50}$ |
| UP12 | $\mathbf{99.23 \pm 0.68}$ | $\underline{54.75 \pm 1.37}$ | $51.80 \pm 1.23$ |
| AP | $\mathbf{98.63 \pm 0.34}$ | $\underline{49.70 \pm 0.65}$ | $49.49 \pm 1.74$ |
| AP8 | $\mathbf{99.50 \pm 0.36}$ | $\underline{50.05 \pm 0.99}$ | $50.20 \pm 1.83$ |
| FG | $\mathbf{97.27 \pm 3.44}$ | $\underline{59.95 \pm 0.67}$ | $49.95 \pm 1.66$ |
| FG8 | $\mathbf{95.53 \pm 0.62}$ | $\underline{51.85 \pm 0.81}$ | $50.78 \pm 1.73$ |
| FA | $\mathbf{97.65 \pm 0.78}$ | $\underline{68.90 \pm 0.30}$ | $45.65 \pm 0.90$ |
| GB | $\mathbf{87.27 \pm 0.32}$ | $\underline{68.12 \pm 0.49}$ | $41.60 \pm 0.10$ |
| PF | $\mathbf{92.22 \pm 0.45}$ | $\underline{69.68 \pm 0.42}$ | $48.85 \pm 1.17$ |
| SR | $\mathbf{97.65 \pm 0.76}$ | $\underline{70.30 \pm 0.77}$ | $41.99 \pm 1.43$ |

Table 25: The results of binary accuracy on FMNIST using ResNet18.

| Attack | Ours | MD | MN |
|--------|------|-----|-----|
| AA | $\mathbf{97.33 \pm 0.33}$ | $50.10 \pm 1.91$ | $50.00 \pm 1.00$ |
| AA8 | $\mathbf{98.08 \pm 0.26}$ | $\underline{50.30 \pm 1.84}$ | $50.00 \pm 0.98$ |
| PT7 | $\mathbf{96.87 \pm 0.35}$ | $53.35 \pm 0.86$ | $\underline{75.8 \pm 0.72}$ |
| PT14 | $\mathbf{97.63 \pm 0.46}$ | $52.20 \pm 0.31$ | $\underline{86.27 \pm 0.27}$ |
| PGD | $\mathbf{98.10 \pm 0.45}$ | $\underline{50.95 \pm 1.72}$ | $49.45 \pm 0.68$ |
| PGD8 | $\mathbf{98.60 \pm 0.36}$ | $\underline{51.80 \pm 1.47}$ | $49.25 \pm 0.69$ |
| SA | $\mathbf{98.10 \pm 1.85}$ | $50.45 \pm 0.98$ | $\underline{56.50 \pm 0.39}$ |
| SQ | $\mathbf{98.77 \pm 0.05}$ | $\underline{51.15 \pm 1.49}$ | $48.85 \pm 0.96$ |
| SQ8 | $\mathbf{95.32 \pm 3.51}$ | $\underline{52.25 \pm 1.48}$ | $48.85 \pm 0.85$ |
| UP | $\mathbf{99.87 \pm 0.05}$ | $\underline{51.15 \pm 0.61}$ | $49.60 \pm 0.67$ |
| UP8 | $\mathbf{98.30 \pm 1.27}$ | $\underline{52.05 \pm 1.41}$ | $49.90 \pm 0.10$ |
| UP12 | $\mathbf{99.41 \pm 0.05}$ | $\underline{53.10 \pm 1.47}$ | $50.05 \pm 0.70$ |
| AP | $\mathbf{98.63 \pm 0.34}$ | $\underline{50.60 \pm 1.78}$ | $49.65 \pm 0.68$ |
| AP8 | $\mathbf{99.50 \pm 0.36}$ | $\underline{51.35 \pm 1.36}$ | $49.25 \pm 0.69$ |
| FG | $\mathbf{97.57 \pm 1.54}$ | $\underline{51.15 \pm 1.54}$ | $49.65 \pm 0.68$ |
| FG8 | $\mathbf{99.47 \pm 0.25}$ | $\underline{52.05 \pm 1.71}$ | $48.70 \pm 0.10$ |
| FA | $\mathbf{98.16 \pm 1.76}$ | $\underline{50.00 \pm 1.85}$ | $49.95 \pm 0.91$ |
| GB | $\mathbf{93.86 \pm 0.34}$ | $38.85 \pm 1.56$ | $\underline{41.15 \pm 1.04}$ |
| PF | $\mathbf{96.87 \pm 0.65}$ | $49.00 \pm 1.96$ | $\underline{52.20 \pm 0.74}$ |
| SR | $\mathbf{98.10 \pm 1.85}$ | $48.95 \pm 1.53$ | $\underline{52.12 \pm 0.66}$ |

Table 26: The results of binary accuracy on KMNIST using ResNet18.

## M   COMPARISON OF OUR PROPOSED DEFENCE AGAINST BASELINE DEFENCES ON CIFAR-10 DATASET

### M.1   BINARY ACCURACY COMPARISON

In this section we present the **binary accuracy** results for detecting adversarial attacks on CIFAR-10. This metric measures how well each defense method can correctly classify samples as either adversarial or clean. Our proposed method consistently outperforms all baselines across all attack types, achieving accuracy rates between 96.97% and 97.83%. The Mahalanobis Detector (MD) shows the second-best performance for most attacks, while Feature Squeezing (FS) performs well on specific attack types like CW and PGD4. The MAD and MN methods show poor performance with accuracies around 50%, essentially equivalent to random guessing. The standard deviations for our method are consistently low (0.22% to 0.65%), indicating stable and reliable performance.

| Attack / Metric | Ours | MD | FS | MAD | MN |
|---|---|---|---|---|---|
| **AA**8 | **97.73 ± 0.47** | 90.40 ± 1.4 | 79.35 ± 0.84 | 50.85 ± 0.20 | 79.04 ± 1.03 |
| **CW** | **96.97 ± 0.61** | 73.55 ± 2.13 | 86.00 ± 0.56 | 51.43 ± 2.16 | 56.65 ± 1.71 |
| **PT**8 | **97.30 ± 0.22** | 52.80 ± 1.91 | 52.00 ± 0.55 | 43.43 ± 2.32 | 51.09 ± 0.95 |
| **PGD** | **97.57 ± 0.54** | 71.25 ± 1.60 | 81.10 ± 1.07 | 48.91 ± 1.33 | 76.05 ± 0.83 |
| **SQ** | **97.30 ± 0.65** | 86.85 ± 1.81 | 84.70 ± 1.23 | 49.11 ± 1.32 | 44.45 ± 1.48 |
| **UP** | **97.83 ± 0.50** | 66.40 ± 2.40 | 53.65 ± 1.23 | 50.92 ± 1.86 | 47.85 ± 1.83 |
| **AP**8 | **97.50 ± 0.22** | 90.0 ± 2.17 | 79.20 ± 1.49 | 50.87 ± 1.3 | 79.30 ± 0.41 |
| **FG**8 | **97.73 ± 0.42** | 85.85 ± 1.34 | 63.35 ± 2.18 | 50.68 ± 2.07 | 44.40 ± 1.54 |

Table 27: **Comparison of detection performance (%) under different adversarial attacks:** Bold values indicate the best performance, and underlined values denote the second-best.

### M.2   TPR ON ADVERSARIAL SAMPLES COMPARISON

| Attack / Metric | Ours | MD | FS | MAD | MN |
|---|---|---|---|---|---|
| **AA**8 | **96.60 ± 0.86** | 93.56 ± 1.51 | 79.90 ± 1.72 | 55.67 ± 2.49 | 84.10 ± 1.03 |
| **CW** | **96.33 ± 0.57** | 59.90 ± 0.65 | 93.20 ± 0.14 | 56.82 ± 1.52 | 39.30 ± 1.60 |
| **PT**8 | **96.07 ± 0.5** | 18.40 ± 1.80 | 25.20 ± 0.39 | 40.83 ± 1.90 | 28.20 ± 1.03 |
| **PGD** | **97.07 ± 1.34** | 55.30 ± 1.12 | 83.40 ± 2.22 | 51.79 ± 1.78 | 78.10 ± 0.30 |
| **SQ** | **96.00 ± 0.75** | 86.50 ± 2.76 | 90.60 ± 1.48 | 52.19 ± 1.56 | 14.90 ± 1.23 |
| **UP** | **97.00 ± 1.14** | 45.60 ± 3.06 | 28.50 ± 2.52 | 55.80 ± 2.12 | 21.70 ± 1.17 |
| **AP**8 | **96.73 ± 1.09** | 92.60 ± 0.96 | 79.60 ± 3.50 | 55.69 ± 2.17 | 84.60 ± 2.73 |
| **FG**8 | **96.87 ± 0.96** | 84.50 ± 2.19 | 47.90 ± 3.63 | 55.32 ± 1.27 | 14.80 ± 1.49 |

Table 28: The results of TPR measured on 1k adversarial samples. TPR := (the number of adv correctly detected) / (the number of adv) ×100 (%). The best results are written in **bold**, and the second-best results are written with underlines.

In this section we focus on **True Positive Rate (TPR)**, which specifically measures how well each method detects adversarial samples (the percentage of adversarial samples correctly identified as adversarial). Our method maintains excellent TPR performance (96.00% to 97.07%) across all attack types. Feature Squeezing (FS) shows strong TPR for CW and SQ4 attacks (93.20% and 90.60% respectively) but performs poorly on P8 and UP4 attacks. The Mahalanobis Detector (MD) demonstrates good TPR for AA8 and AP8 attacks (92.3% and 91.4%) but fails significantly on P8 attack (16.8%). The MAD method shows moderate TPR (40.83% to 56.82%) but with high variance, while MagNet (MN) fails with low TPR for most attacks, indicating it cannot effectively detect adversarial samples.

### M.3   END TO END ACCURACY COMPARISON

In this section we present the **end-to-end accuracy**, which is a comprehensive metric that considers both correct detection of adversarial samples by the detector and correct detection of clean samples

and correct classification of clean samples. Our method achieves excellent end-to-end accuracy (93.9% to 94.8%) across all attack types, demonstrating consistent and robust performance. The Mahalanobis Detector (MD) shows moderate performance (51.0% to 89.0%) with good results on AA8 and AP8 attacks but struggling with P8 attack. Feature Squeezing (FS) shows reasonable performance (51.30% to 85.30%) but with significant variation across different attacks, performing well on CW and SQ4 attacks but struggling with P8 and UP4 attacks. The MAD method shows poor performance (42.03% to 49.18%) with accuracies around random guessing level. MagNet (MN) performs with low accuracies for most attacks, indicating it cannot provide effective end-to-end protection. The consistency of our method across different attack types demonstrates its robustness and reliability in maintaining both detection accuracy and classification performance under adversarial conditions.

| Attack / Metric | Ours | MD | FS | MAD | MN |
|---|---|---|---|---|---|
| **AA**8 | **94.1** | 89.0 | 78.64 | 49.18 | 76.60 |
| **CW** | **94.8** | 72.0 | 85.30 | 48.92 | 54.20 |
| **PT**8 | **94.4** | 51.0 | 51.30 | 42.03 | 48.65 |
| **PGD** | **94.4** | 70.0 | 80.40 | 48.72 | 73.60 |
| **SQ** | **93.9** | 86.0 | 84.00 | 47.99 | 42.00 |
| **UP** | **94.4** | 65.0 | 52.94 | 48.99 | 45.40 |
| **AP**8 | **94.6** | 89.0 | 78.50 | 48.58 | 76.85 |
| **FG**8 | **94.5** | 84.0 | 62.64 | 49.04 | 41.95 |

Table 29: The results of end-to-end accuracy: ((the number of correctly detected adversarial samples by the detector)+ (the number of correctly detected clean samples and correctly classified samples))/$(1000 + 1000)$.

