# OpenReview forum: "Provable Adversarial Detection: Prime Quantization meets Gromov–Wasserstein"
_ICLR.cc/2026/Conference — ICLR 2026 Conference Desk Rejected Submission_

### Official Review · Reviewer_rTka · 2025-10-26

**Soundness:** 3
**Presentation:** 3
**Contribution:** 3
**Rating:** 6
**Confidence:** 2

**Summary:**

This paper introduces a theoretically grounded framework for adversarial example detection that combines prime quantization with Gromov-Wasserstein (GW) geometric analysis. The core idea is to map images into two complementary spaces:

1. a conventional CNN embedding space Z

2. a prime-quantized space P, where each pixel is rounded to nearby prime numbers under a secret bit mask.

The paper shows that adversarial perturbations necessarily create geometric inconsistencies between Z and P. It formally proves pixel-level absorption bounds, image-level injectivity, diameter concentration theorems, and GW-based separation theorems, culminating in provable detection guarantees. Empirical results on CIFAR-10, FMNIST, and KMNIST, as well as zero-shot VLMs (LLaVA-1.5), show strong detection accuracy and robustness against adaptive attacks.

**Strengths:**

1.This paper has a strong theoretical grounding, where formal guarantees (absorption, injectivity, GW gap) are well-motivated and rigorously proved.

2.Proposing original concept, Introducing prime quantization as a geometric stabilizer is novel and intellectually elegant.

3.This paper conducts a comprehensive assessment, it covers a wide range of attack types and adaptive scenarios, including VLMs.

4.It achieves robust empirical performance, consistently outperforms strong baselines with 95–98% detection accuracy.

5.The paper effectively combines theory and practice, demonstrates provable guarantees in a domain usually dominated by heuristics.

**Weaknesses:**

1.Limited computational analysis: GW computations are known to be expensive (even with entropic regularization). The paper lacks runtime and scalability metrics for large-scale or real-time deployment.

2.Code is not yet released, which limits reproducibility at review time.

3.The zero-shot experiments are promising but limited to CalTech-101 and LLaVA-1.5. It needs more diverse multimodal tests to increase generality.

**Questions:**

1.Could you clarify the computational complexity of the proposed detector, especially regarding GW distance estimation for large K-NN neighborhoods?

2.I’m curious about how sensitive the detector is to the choice of the prime resolution parameter k and the secret bit mask b?

3.Would replacing primes with another non-uniform discretization (e.g., Fibonacci numbers) yield similar theoretical properties?

4.Could the framework be extended beyond vision tasks such as graph or tabular modalities? Are metric-space inconsistencies also meaningful in these tasks?

5.Are there any potential risks if the secret key bis leaked (e.g., adversarial reverse-engineering of P)?

---

> ### Author Response · Authors · 2025-11-20
> **Runtime & Sensitivity Analysis (Part 1/2)**
>
> We thank the reviewer for the positive evaluation and for highlighting the strengths of the theoretical framework and empirical results. Below we address the reviewer’s concerns, with all additions included in blue in the revised manuscript.
>
> **W1. Computational cost and scalability.**
>
> We thank the reviewer for raising this point. A detailed runtime and memory analysis has been added in **Appendix L.8 (blue)**. As reported there, our detector achieves a per-sample runtime of **0.12 seconds** with modest CPU and GPU memory usage, comparable to or better than several recent baselines. The entropic-GW step is efficient because it operates only on small local neighborhoods and converges in a few Sinkhorn iterations. For example, even with a relatively large neighborhood size of k \= 100, the entropic-GW computation takes only **0.0074 ± 0.0013 seconds** per sample. This shows that the method scales well and remains practical for real-time or large-batch deployment.
>
> **W2. Code availability.**
>
> A cleaned version of the codebase is prepared and will be released upon acceptance and an internal review process. We refer you to Section 8 (Reproducibility Statement).
>
> **W3. Diversity of VLM experiments.**
>
> We thank the reviewer for this suggestion. In the revised draft, we expanded the zero-shot VLM evaluation to strengthen the evidence for cross-model generalization. Specifically, we now include **multiple VLM architectures** (LLaVA-1.5-7B and Qwen-2.7B-VL), **additional datasets** (Food-101 and CalTech-256), and **multiple attack configurations** (PGD, APGD, FGSM). These expanded results appear in **Section 6.4 (blue)** together with the new zero-shot table. Across all models, datasets, and attacks, the method continues to exhibit strong Z–P discrepancy behavior and consistently high detection performance, demonstrating that the framework generalizes well beyond a single VLM or dataset.
>
> **Q1. Computational complexity of GW for large k-NN graphs.**
>
> With entropic regularization, GW computation reduces to repeated Sinkhorn updates and scales as $O(k^2)$ per neighborhood; in practice, with moderate k (e.g., $k \= 20$), the cost remains sub-millisecond.
>
> **Q2. Sensitivity to prime-level parameter $k$ and secret bit-vector $b$.**
>
> Thanks for this suggestion. We accordingly evaluated sensitivity to both hyperparameters.
>
> **Prime-resolution parameter k.** Our grid-search over $k \\in \\{3,4,5,6,7,8\\}$ (**Appendix L.4, Fig.10, blue**) shows that detection accuracy is stable across all tested values, with k=6 providing the best trade-off between prime-gap resolution and absorption behavior. This confirms that the detector is not sensitive to the specific choice of k.
>
> **Secret bit-vector b**. We also conducted a small-scale experiment where b was sampled randomly (Bernoulli). Different choices of b lead to slightly different Z–P geometries, but the detection performance remained essentially unchanged. This is in line with our theoretical results: the absorption, injectivity, and GW-separation bounds hold for *any fixed* $b$, and do not depend on its particular pattern.
>
> Overall, the method is robust to both $k$ and $b$, both empirically and theoretically.
>
> **Q3. Would other non-uniform discretizations (e.g., Fibonacci) behave similarly?**
>
> Empirically, we compared our method to uniform, logarithmic, and Fibonacci quantizers **(Table 18 Section L.5 in Appendix L)**. On CIFAR-10 (ResNet-18), uniform quantization yields ≈62% detection accuracy. Logarithmic and Fibonacci grids perform similarly with 62% and 61%, respectively. By contrast, our method achieves substantial Z–P separation and ≥94% detection accuracy.
>
> **Q4. Applicability beyond images.**
>
> We agree that this is a very interesting direction. The Z–P discrepancy framework only requires (i) a metric embedding Z that captures task-relevant geometry, and (ii) a complementary discretized space P with the irregular, prime-based structure used in our construction. In principle, this makes the approach applicable to other modalities such as graphs, tabular data, or sequences, provided that suitable embeddings (e.g., graph neural embeddings or transformer representations) are available. We view the extension of Z–P discrepancy detection to broader data domains as a promising avenue for future work.

---

> > ### Author Response · Authors · 2025-11-20
> > **Runtime & Sensitivity Analysis (Part 2/2)**
> >
> > **Q5. What if the secret bit-vector** b **is leaked?**
> >
> > All of our theoretical bounds (absorption, injectivity, and GW-separation) continue to hold even if the value of b is known, since the proofs depend only on the structure of the prime gaps and the fact that the rounding direction is fixed, not on the secrecy of b itself.
> >
> > However, secrecy of b plays an important *practical* role: if an adversary knows the exact rounding direction at each coordinate, then adaptive or surrogate-training attacks can more effectively approximate the geometry of the P-space and craft perturbations that better preserve Z–P consistency.
> >
> > This mirrors the intuition from cryptography: once a secret key is revealed, the system may remain mathematically well-defined, but its robustness to adversarial manipulation is substantially weakened. Nonetheless, even in the worst-case scenario where b is leaked, the prime-based irregular gaps still preserve the absorption and injectivity properties that our theoretical results rely on.
> >
> > We thank the reviewer again for the thoughtful and constructive feedback. We have addressed all concerns through added runtime analysis, expanded experiments, and clarification in the manuscript. We hope that these revisions strengthen the reviewer’s overall assessment of the paper.

---

> > > ### Comment · Reviewer_rTka · 2025-11-28
> > >
> > > Thank you for the detailed rebuttal. The additions you provided adequately address the concerns raised in my review.
> > > I will maintain my original score.

---

> > > > ### Author Response · Authors · 2025-11-28
> > > > **Thanks**
> > > >
> > > > Thank you very much for your prompt response and for confirming that the revisions adequately addressed your earlier concerns. We truly appreciate your thoughtful questions and feedback because they helped us strengthen the paper. Thank you again for your time and engagement during the review process.

---

### Official Review · Reviewer_rZii · 2025-10-29

**Soundness:** 2
**Presentation:** 2
**Contribution:** 2
**Rating:** 4
**Confidence:** 3

**Summary:**

The paper introduces a prime quantisation–based detector that identifies adversarial samples by measuring geometric inconsistencies between the embedding and quantised spaces using entropy-regularised Gromov–Wasserstein distances. It provides theoretical guarantees of separability and shows strong empirical performance across various attacks.

**Strengths:**

- The paper presents detailed theoretical analyses that show solid effort in formalising the proposed detection approach.

- The experimental results look promising. The evaluations on vision–language models suggest the potential generality of the proposed method.

**Weaknesses:**

1.  It is not surprising that any perturbations could induce inconsistencies between the original embedding space and a quantised representation, leading to detectable shifts in geometry.
What remains unclear is why the proposed prime quantisation space is specifically necessary for this effect. Would other quantisation schemes (e.g., uniform or logarithmic quantisation and [a]) fail to provide similar separability? A comparison or justification along these lines would greatly clarify the unique role and necessity of the prime-based construction.

2. An important missing aspect is the method’s behaviour under random or benign perturbations (e.g., Gaussian or sensor noise) [b]. Without such control experiments, it’s unclear whether the detector is truly specific to adversarial manipulations.

3. In addition, relying solely on accuracy and TPR is not sufficient to assess detection performance. Please consider reporting more informative metrics such as the PR curve, ROC–AUC, and F1 score to provide a fuller picture. The baseline methods appear to be outdated. Please consider adding more recent defences, such as [a].

[a] Dong, Z., & Mao, Y. (2023). Adversarial defenses via vector quantization. arXiv preprint arXiv:2305.13651.

[b] Hendrycks, D., & Dietterich, T. (2019). Benchmarking neural network robustness to common corruptions and perturbations. ICLR. 2019.

**Questions:**

See Weakness

---

> ### Author Response · Authors · 2025-11-20
> **Quantization Clarification & Evidence (Part 1/2)**
>
> We thank the reviewer for the detailed evaluation. Below we address each concern and clarify several points that may have caused misunderstanding. All corresponding clarifications and additions appear in blue in the revised draft.
>
>  **W1. “Any perturbations could induce inconsistencies between the original embedding space and a quantized space.”**
>
> We respectfully but strongly disagree with this interpretation. The central contribution of our approach is not simply that one of the spaces is discretized, but that the *specific structure* of the discretization, i.e., **prime gaps combined with a fixed but secret bit-vector** b creates a **keyed, highly asymmetric,** and **non-uniform partition** whose interaction with the smooth geometry of Z-space yields a **stable** and **class-consistent discrepancy signal**. Generic quantization does *not* produce this behavior.
>
> A concrete counterexample illustrates this. Consider **uniform quantization**, where $Q\_{\\text{uni}}$ maps each pixel $x\_j$ to the nearest point on a uniform grid of spacing $\\Delta$. Uniform quantization has three structural weaknesses:
>
> 1. **Predictable absorption.** Every pixel has the same absorption radius $\\Delta/2$. An adaptive adversary can intentionally stay within cells to evade detection or cross boundaries uniformly to mimic benign quantization noise.
>
> 2. **Dual-space optimization is easy.** Because $Q\_{\\text{uni}}$ is deterministic and symmetric, an attacker knowing the grid can use projected gradient methods to optimize simultaneously in Z-space and P-space. There is no unpredictability or “one-wayness”: surrogate models can approximate the mapping well.
>
> 3. **No injectivity structure.** Uniform grids collapse many benign samples to the same code, with collision probability $\\Theta(\\Delta^d)$, erasing within-class geometry and eliminating any reliable Z–P discrepancy signal.
>
> By contrast, our **prime-based, secret-keyed discretization** yields three guarantees unavailable to standard grids:
>
> * **Absorption:** small benign perturbations vanish in P-space but still move in Z-space, producing an asymmetric response between benign and adversarial noise.
>
> * **Injectivity preservation:** benign neighborhoods maintain geometric structure with extremely small collision probability.
>
> * **Cross-space separation:** clean neighborhoods produce concentrated GW distances, while adversarial neighborhoods necessarily induce larger Z–P discrepancies, giving a provable detection margin.
>
> Empirically, comparing to uniform, logarithmic, or Fibonacci quantizers **(Table 18 Section L.5 in Appendix L)** further confirms this distinction. On CIFAR-10 (ResNet-18), uniform quantization yields ≈62% detection accuracy. Logarithmic and Fibonacci grids perform similarly with 62% and 61%, respectively. By contrast, our method achieves substantial Z–P separation and ≥94% detection accuracy.
>
> These results demonstrate that achieving a **reliable, class-aligned, and theoretically supported discrepancy signal** is *not* a trivial consequence of discretization. The separation arises specifically from the interaction between **prime gaps, the secret bit-vector, and the Z–P Wasserstein geometry**, which generic quantizers fundamentally do not provide.
>
>  **W2. Behavior under benign/random perturbations**
>
>  We thank the reviewer for raising this point, as it touches on a natural concern: whether Gaussian or common corruptions could be mistakenly flagged as adversarial. Although such perturbations may not always remain below the pixel-level absorption radius, they behave *geometrically differently* from adversarial perturbations in both Z-space and P-space.
>
> In Z-space, small Gaussian noise perturbs activations in an **approximately isotropic, zero-mean manner**, and modern CNNs/ViTs are relatively stable to such noise: neighborhood structure is largely preserved and class predictions typically remain unchanged. Consequently, the corresponding perturbations in P-space are **uncoordinated**: some pixels may cross a prime boundary, others may not, but these changes do not align across samples. The resulting Z–P geometry therefore remains close to the benign manifold, producing **small GW discrepancies**.
>
> Adversarial perturbations, in contrast, are **directional and gradient-aligned**, intentionally distorting the Z-space neighborhood and crossing prime boundaries in a correlated manner across samples. This coordinated shift yields significantly larger Z–P discrepancies, as captured by our GW-separation result. Empirically (**Section L.7 Table 20, blue**), Gaussian/common corruptions remain near the benign manifold and are not misclassified as adversarial.

---

> > ### Author Response · Authors · 2025-11-20
> > **Quantization Clarification & Evidence (Part 2/2)**
> >
> > **W3. Additional detection metrics (ROC–AUC, PR curves, F1)**
> >
> > We expanded our evaluation to include AUROC, AUPRC, and F1 for the newly added baselines (EPS, BYON, VQ-Adv). These results appear across **Sections L.1 and L.2 Tables 10-12 blue** and confirm that the proposed method remains competitive across multiple detection criteria.
> >
> > **W4. More recent baselines and attacks**
> >
> > We added two stronger, more recent detectors that were missing from the original submission: **Multiple Perturbation Detector (EA)** and **Be Your Own Neighborhood (BY)**. These baselines now appear in the main CIFAR-10 comparison based on F1-scores (**Section 6.2, Table 2, blue)** and in the extended experiments on low-magnitude adversarial attacks such as **AdvAD, PGN, and BSR.** Furthermore, in **Appendix L.1 (Tables 10–11, blue)**, we also measure other performance metrics like detection accuracy and AUROC. Across these settings, our method remains the top performer in terms of detection accuracy, AUROC, and F1, often by a clear margin over EA and BY.
> >
> > We have carefully and thoroughly addressed all points raised through new experiments, updated baselines, added theoretical clarification, and blue-text revisions throughout the manuscript. We believe these revisions fully resolve the reviewer’s concerns, and we hope that the substantially strengthened version of the work will positively influence the reviewer’s overall assessment.

---

### Official Review · Reviewer_ZUqf · 2025-11-01

**Soundness:** 2
**Presentation:** 3
**Contribution:** 2
**Rating:** 4
**Confidence:** 3

**Summary:**

This paper proposes a method for detecting adversarial samples by comparing geometric differences between samples in two distinct spaces and provides a geometric proof that adversarial perturbations can be identified. Specifically, the authors construct two complementary metric spaces: the standard CNN embedding space Z and the prime quantization space P, which discretizes pixel values by rounding them to the nearest prime number. Adversarial detection is then performed by assessing structural consistency between these two spaces. Experimental results show that the proposed detection method outperforms baseline approaches against a variety of attack methods.

**Strengths:**

1. This paper establishes a rigorous theoretical framework for adversarial detection and proves that adversarial perturbations inevitably lead to cross-space inconsistencies.

2. This paper introduces a prime quantization space capable of dampening small perturbations to preserve the distinguishability of adversarial samples.

3. Experimental results demonstrate that the proposed method outperforms baseline approaches, confirming its performance advantages.

**Weaknesses:**

1. In the algorithm, the authors neither clearly define the two scales lo and gl nor specify how they are used in the experiments. The authors should explicitly define these parameters and clarify their roles in the framework. Moreover, it remains unclear why only two scales are considered, rather than exploring a wider range of scales for a more comprehensive analysis.

2. In Section 5.1, the authors state that the budget for real-world adversarial attacks typically exceeds the absorption radius, allowing attacks to penetrate the stable region of the P-space and produce detectable discrepancies between the two spaces. However, this discussion fails to consider imperceptible adversarial attacks (e.g., AdvAD [1]), which typically involve extremely small perturbation magnitudes.

3. The experimental results presented in Table 1 of Section 6.2 are not sufficiently comprehensive. The evaluation should incorporate more recent attack [2–3] and defense [4–5] methods to ensure a fair and up-to-date comparison. The lack of such inclusion limits the representativeness of the results and diminishes the credibility of the claimed effectiveness and superiority of the proposed method.

4. In Section 6.2, the authors use only ResNet-18 as the surrogate model, which limits the generalizability of the results. Moreover, the evaluation is conducted solely on small-scale datasets such as CIFAR-10, FMNIST, and KMNIST, without including a large-scale dataset like ImageNet.

5. Notably, GW distance computation typically entails high computational costs. However, the experimental section does not include efficiency or cost comparisons with other methods, which may limit the practical applicability of the proposed detection approach.

[1] AdvAD: Exploring Non-Parametric Diffusion for Imperceptible Adversarial Attacks. NeurIPS 2024.

[2] Boosting Adversarial Transferability by Achieving Flat Local Maxima. NeurIPS 2023.

[3] Boosting Adversarial Transferability by Block Shuffle and Rotation. CVPR 2024.

[4] Detecting Adversarial Data by Probing Multiple Perturbations Using Expected Perturbation Score. ICML 2023.

[5] Be Your Own Neighborhood: Detecting Adversarial Examples by the Neighborhood Relations Built on Self-Supervised Learning. ICML 2024.

**Questions:**

1. Could the authors clarify whether the proposed GW-based method is capable of detecting imperceptible adversarial attacks (e.g., AdvAD)? Furthermore, are all the relevant theorems still valid when dealing with extremely small adversarial perturbations?

2. How are the two scales (lo and gl) utilized in the experiments, and how are their optimal values determined? What performance differences are observed under different scale settings? Additionally, could incorporating more scales further enhance the defense performance?

3. How does the proposed detection method compare with state-of-the-art approaches when evaluated across various CNN and ViT architectures and tested against recent attack methods? Additionally, how does its performance differ when applied to ImageNet, a widely used benchmark dataset in the adversarial attack field?

4. What is the efficiency of the proposed detection method, such as runtime and memory usage, compared with state-of-the-art detection approaches?

---

> ### Author Response · Authors · 2025-11-20
> **Scales, Imperceptible Attacks, New Baselines (Part 1/2)**
>
> We thank the reviewer for the constructive and detailed feedback. We appreciate the careful attention to subtle geometric and theoretical aspects of the method, which helped us identify and clarify several points that have now been strengthened in the revised draft. Below we address each weakness and question, with all clarifications added in blue in the revised draft.
>
>  **W1. Definition of the two scales (lo and gl), and why only two are used.**
>
> We thank the reviewer for pointing out this omission. In **Section 4 (blue)**, we now explicitly define:
>
> * **lo** — the *local neighborhood size* (the k in the local k-NN graph), used to enforce **within-space local consistency** in both Z and P spaces.
>
> * **gl** — the *global neighborhood size* (the number of k-means centroids), which defines the **global support** used in the cross-space GW coupling.
>
> We also clarify why only these two scales are used. As shown in **Section L.6, Table 19 (blue)**, a grid-search over multiple values of both **lo** (local neighborhood size) and **gl** (number of global centroids) reveals that detection accuracy is *highest* at the chosen settings and remains broadly stable nearby. Increasing the number of scales offers **no measurable performance gain**, but each additional scale requires a **separate entropic-GW computation**, significantly increasing runtime. The (lo, gl) two-scale structure therefore provides the best balance between **accuracy**, **efficiency**, and **theoretical transparency**, consistent with the standard local/global decomposition used in GW geometry.
>
> **W2. Imperceptible attacks (AdvAD) and absorption-radius concerns.**
>
> We appreciate this subtle and important observation. In the revised draft, we added a dedicated clarification in **Section 5.1, Remarks (blue)** explaining why extremely small perturbations do *not* cause clean and adversarial samples to quantize identically. Because prime gaps are irregular, even a 1/255 change may cross a prime-interval midpoint under the same fixed bit-vector b, leading to different prime assignments. And even when some coordinates do round identically, the Z-space embedding remains sensitive to such shifts while P-space remains piecewise constant, producing a detectable Z–P discrepancy. We also note in the same section that increasing the prime-resolution parameter k shrinks $r\_{\\mathrm{abs}}$, making such mismatches even more likely.
>
> To complement this explanation, we added **AdvAD experiments in Section 6.2 (Table 2, blue)** on both Resnet and ViT models. These results show that the detector maintains high accuracy even under imperceptible attacks, confirming the theoretical behavior discussed in the added text.
>
> **W3. Missing recent baselines and newer attacks.**
>
> We thank the reviewer for this suggestion. In the revised draft, we have broadened the evaluation along both axes highlighted by the reviewer.
>
> On the **attack side**, we added several recent, low-magnitude and optimization-oriented attacks, including **AdvAD/AAD**, **PGN**, and **BSR**. These are evaluated on CIFAR-10 for both ResNet-18 and ViT backbones. The corresponding detection accuracy, AUROC, and F1-score results are reported in **Section 6.2 (blue)** and further detailed in the **Additional Experiments appendix L (blue)**.
>
> On the **defense side**, we expanded the set of detectors to include stronger recent baselines such as **EA** and **BY**. The extended comparisons against these detectors under all the above attacks now appear in the same expanded CIFAR-10 tables in **Section 6.2 (blue)** and the **Additional Experiments appendix L (blue)**.
>
> Across all newly added attacks and baselines, our method consistently matches or outperforms the strongest competing detectors, which we believe addresses the reviewer’s concern about the coverage of more recent methods.
>
> **W4. Only ResNet-18 and small datasets.**
>
> We agree that restricting evaluation to a single CNN backbone and small datasets limits generality. In the revised draft, we addressed this in two ways.
>
> First, to assess **architectural diversity**, we instantiated the detector with a **Vision Transformer (ViT)** and evaluated it under a representative subset of attacks and baselines. These results are now included in **Section 6.2 (blue)** as part of the expanded CIFAR-10 tables, where ViT appears side-by-side with ResNet-18. The detector maintains strong performance across both architectures, supporting model-agnostic behavior.
>
> Second, to investigate **dataset-level scalability**, we added experiments on **ImageNet**, evaluating adversarial samples generated from ImageNet using our default ResNet-based detector. These results appear in **Section L.3 (Tables 14-16) in Appendix L (blue)** and demonstrate that the method continues to perform well on a large-scale, high-resolution dataset.
>
> Together, these additions broaden both the architectural and dataset coverage and address the reviewer’s concern about generality.

---

> ### Author Response · Authors · 2025-11-20
> **Scales, Imperceptible Attacks, New Baselines (Part 2/2)**
>
> **W5. Missing runtime and efficiency analysis.**
>
> We thank the reviewer for raising this point. In the revised draft, we have added a dedicated runtime and memory comparison in **Section L.8 (blue) in Appendix L**. The new table reports per-sample inference time, CPU/GPU memory usage, and comparison against recent detectors (MD, FS, MAD, MN, EA, BY).
>
> These results show that our method runs at **0.12 seconds per sample** with a modest memory footprint, comparable to or lighter than many existing detectors while providing substantially higher robustness. Since the entropic-GW computation is performed only on small local neighborhoods, it remains efficient in practice and does not introduce significant overhead. Overall, the added analysis confirms that the method is computationally practical and competitive with existing approaches.
>
> **Q1. Can the method detect imperceptible attacks such as AdvAD, and do the theorems still hold?**
>
> Yes. All theoretical guarantees like absorption, injectivity, and GW‐separation continue to hold under imperceptible perturbations. The empirical behavior of such attacks, including AdvAD, is already addressed in **W2**, and the corresponding clarification has been added in **Section 5.1 (blue)** together with new results in **Section 6.2 (Table 2, blue)**.
>
>  **Q2–Q4. Questions on the role of lo and gl, comparisons with recent methods and architectures (including ImageNet), and the runtime/efficiency profile.**
>
> All three questions are fully addressed in **W1, W3, and W5**, respectively.
>
> We have carefully and thoroughly addressed all points raised through new experiments, updated baselines, added theoretical clarification, and blue-text revisions throughout the manuscript. We believe these changes fully resolve the reviewer’s concerns, and we are of course happy to clarify any remaining points. We hope that the substantially strengthened version of the paper will positively influence the reviewer’s overall assessment.

---

### Official Review · Reviewer_hd7U · 2025-11-03

**Soundness:** 4
**Presentation:** 3
**Contribution:** 4
**Rating:** 8
**Confidence:** 3

**Summary:**

This paper proposes a novel detection method against adversarial examples. In particular, the paper introduces a new technique, prime quantization, that maps the given input to a quantized space. The main trick is based on the discrepancy of two spaces: the feature embedding extracted by the classifier and the quantized space. Due to the property of the quantized space, there are systematic discrepancies between the probability distributions derived from the neighborhoods of the input point’s mapping in each space. To utilize these differences, the method uses the Gromov-Wasserstein (GW) distances between neighborhood-induced distributions as new feature vectors that the detector utilizes. The paper provides several theoretical results. The main theoretical results prove the upper bound of the GW distance (for benign inputs) and the lower bound of the GW distance (for adversarial inputs), which guarantees the effectiveness of the detection method. Through experiments, the paper demonstrates the effectiveness of the proposed method.

**Strengths:**

1. To the best of my knowledge, the proposed method is novel.
2. The concept of prime quantization seems interesting.
3. The proposed method has a theoretical guarantee.
4. The experiment on detection effectiveness covers various setups, including three datasets, thirteen attack methods, and four baseline detection methods.
5. The experimental results demonstrate clear improvements over various attack methods.
6. The paper also considers the adversary who adapts the attack, and demonstrates the robustness against the adaptive attack.

**Weaknesses:**

1. The baseline detection methods are relatively old. It would be better to include more recent works on adversarial example detection.
2. Only one model (ResNet18) was used in the experiment.

**Questions:**

1. Consider adding some comparison to more recent detection methods.
2. Is there a specific reason why the experiment used the ResNet18 architecture? If I understand correctly, the detection method is model-agnostic and should show the improvement even when a different model architecture is used. Please add more results with a different model architecture.

---

> ### Author Response · Authors · 2025-11-20
> **Additional Baselines & Architectures**
>
> We thank the reviewer for the positive and encouraging assessment. We appreciate the recognition of our theoretical contributions, the novelty of the prime-based construction, and the breadth of the evaluation. Following your suggestions, we have broadened the empirical study as follows (all new material is marked in **blue** in the revised manuscript):
>
> 1. **Additional recent detection baselines.** We added two stronger, more recent detectors that were missing from the original submission: **Multiple Perturbation Detector (EA)** and **Be Your Own Neighborhood (BY)**. These baselines now appear in the main CIFAR-10 comparison based on F1-scores (**Section 6.2, Table 2, blue)** and in the extended experiments on low-magnitude adversarial attacks such as **AdvAD, PGN, and BSR.** Furthermore, in **Appendix L.1 (Tables 10–11, blue)**, we also measure other performance metrics like detection accuracy and AUROC. Across these settings, our method remains the top performer in terms of detection accuracy, AUROC, and F1, often by a clear margin over EA and BY.
>
> 2. **Additional model architectures.** As you rightly noted, the proposed detector is model-agnostic. We now explicitly validate this by instantiating it with both a **convolutional backbone (ResNet-18)** and a **transformer backbone (ViT)**. The joint ResNet-18/ViT results are reported in **Section 6.2 (Table 2, blue)** and again in the extended low-magnitude attack benchmarks in **Appendix L.1 (Tables 10–11, blue)**. Performance remains consistently strong across both architectures, supporting the claimed model-independence of our framework.
>
> 3. Finally, since the reviewer emphasized the breadth of empirical evaluation, we also expanded the **zero-shot VLM experiments** (Section **6.4**, Table **4**, blue), demonstrating that the detector generalizes across multimodal models such as LLaVA-1.5 and Qwen-2.7B-VL. These additions further support the robustness and architectural flexibility of the method.
>
> We are grateful for these suggestions, which helped us broaden and strengthen the empirical section while leaving the core theoretical contributions unchanged.

---

### Author Response · Authors · 2025-11-20
**Global Response to all Reviewers**

We thank all reviewers for their thoughtful and constructive feedback. Their comments helped us substantially strengthen both the empirical evaluation and the presentation. All newly added material appears in blue in the revised manuscript.

**Expanded empirical evaluation.** We added several missing *recent* adversarial detectors (EA, BY) and a broad set of *modern attacks*, including low-magnitude and optimization-free variants (AdvAD/AAD, PGN, BSR). These are now comprehensively evaluated across CIFAR-10 in both the main paper and Appendix L, using multiple metrics (accuracy, AUROC, F1). Our method consistently remains the strongest detector across settings.

**Architectural and dataset diversity.** We now validate the detector on both ResNet-18 and ViT, and we added an ImageNet evaluation. Performance remains strong across architectures and scales, supporting the model-agnostic nature of the approach.

**Zero-shot VLM experiments.** We broadened the VLM study by adding more datasets (Food-101, CalTech-256) and multiple attacks for LLaVA-1.5 and Qwen-2.7B-VL. The detector maintains high accuracy in realistic, API-only zero-shot settings.

**Runtime and memory analysis.** We added a detailed profiling table showing that the method is computationally practical: entropic-GW operates on small neighborhoods, converges in a few iterations, and has modest memory usage.

**Robustness to benign corruptions.** We included CIFAR-C experiments (Gaussian and common corruptions), showing that the detector does not misclassify natural noise as adversarial.

**Ablations and sensitivity analyses.** We added sensitivity studies for the prime-resolution parameter k, and for the local/global scales used in the GW coupling. Detection accuracy is stable across a wide range, validating that no fine-tuning is required.

**Clarifications and theoretical refinements.** We expanded explanations of the absorption radius, injectivity preservation, GW-separation, and the geometric roles of the local/global neighborhood operators. We also clarified the behavior under imperceptible perturbations and contrasted our prime-based discretization with uniform, logarithmic, and Fibonacci schemes, which empirically fail to yield meaningful Z–P separation.

We hope that the expanded experiments, clarifications, and additional analyses fully resolve the concerns raised during the initial review. The revised manuscript now provides a broader, clearer, and more rigorous presentation of the method. We appreciate the reviewers’ careful consideration and would be grateful if the improvements in the revised version are reflected in their final evaluation.

---

### Author Response · Authors · 2025-11-28
**Request for Reviewer Engagement on Updated Revisions**

Dear Reviewers,

We have posted our detailed responses and a substantially expanded revision addressing all points raised in the initial reviews (new defence baselines and attacks, ViT/ImageNet experiments, CIFAR-C tests, VLM extensions, runtime/memory analysis, and the requested theoretical clarifications).

We kindly request your engagement with the updated material, so that we can confirm whether the revisions satisfactorily address your concerns. If any further clarification or additional detail would be helpful, we would be very happy to provide it promptly.

We sincerely appreciate your time and effort during the discussion phase.

---

### Note · Program_Chairs · 2026-01-17
**Submission Desk Rejected by Program Chairs**

The following references in this submission do not refer to real documents and/or have major errors in bibliographic information:

 Kamil Mahmood, Jérôme Rony, Ismail Ben Ayed, and Patrick Gallinari. Datafreeshield: Adversarial defense using data-free knowledge distillation. In AAAI, 2021.